# Earth System Model Evaluation Tool (ESMValTool) v2.0 – extended set of large-scale diagnostics for quasi-operational and comprehensive evaluation of Earth system models in CMIP

Veronika Eyring[1,2], Lisa Bock[1], Axel Lauer[1], Mattia Righi[1], Manuel Schlund[1], Bouwe Andela[3], Enrico Arnone[4,5], Omar Bellprat[6], Björn Brötz[1], Louis-Philippe Caron[6], Nuno Carvalhais[7,8], Irene Cionni[9], Nicola Cortesi[6], Bas Crezee[10], Edouard L. Davin[10], Paolo Davini[4], Kevin Debeire[1], Lee de Mora[11], Clara Deser[12], David Docquier[13], Paul Earnshaw[14], Carsten Ehbrecht[15], Bettina K. Gier[2,1], Nube Gonzalez-Reviriego[6], Paul Goodman[16], Stefan Hagemann[17], Steven Hardiman[14], Birgit Hassler[1], Alasdair Hunter[6], Christopher Kadow[15,18], Stephan Kindermann[15], Sujan Koirala[7], Nikolay Koldunov[19,20], Quentin Lejeune[10,21], Valerio Lembo[22], Tomas Lovato[23], Valerio Lucarini[22,24,25], François Massonnet[26], Benjamin Müller[27], Amarjiit Pandde[16], Núria Pérez-Zanón[6], Adam Phillips[12], Valeriu Predoi[28], Joellen Russell[16], Alistair Sellar[14], Federico Serva[29], Tobias Stacke[17,30], Ranjini Swaminathan[31], Verónica Torralba[6], Javier Vegas-Regidor[6], Jost von Hardenberg[4,32], Katja Weigel[2,1], and Klaus Zimmermann[13]

[1]Deutsches Zentrum für Luft- und Raumfahrt (DLR), Institut für Physik der Atmosphäre, Oberpfaffenhofen, Germany
[2]University of Bremen, Institute of Environmental Physics (IUP), Bremen, Germany
[3]Netherlands eScience Center, Science Park 140, 1098 XG Amsterdam, the Netherlands
[4]Institute of Atmospheric Sciences and Climate, Consiglio Nazionale delle Ricerche (ISAC-CNR), Italy
[5]Department of Physics, University of Torino, Italy
[6]Barcelona Supercomputing Center (BSC), Barcelona, Spain
[7]Department of Biogeochemical Integration, Max Planck Institute for Biogeochemistry, Jena, Germany.
[8]Departamento de Ciências e Engenharia do Ambiente, DCEA, Faculdade de Ciências e Tecnologia, FCT, Universidade Nova de Lisboa, 2829-516 Caparica, Portugal
[9]Agenzia nazionale per le nuove tecnologie, l'energia e lo sviluppo economico sostenibile (ENEA), Rome, Italy
[10]ETH Zurich, Institute for Atmospheric and Climate Science, Zurich, Switzerland
[11]Plymouth Marine Laboratory (PML), Plymouth, UK
[12]National Center for Atmospheric Research (NCAR), Boulder, CO, USA
[13]Rossby Centre, Swedish Meteorological and Hydrological Institute (SMHI), Sweden
[14]Met Office, Exeter, UK
[15]Deutsches Klimarechenzentrum, Hamburg, Germany
[16]Department of Geosciences, University of Arizona, Tucson, AZ, USA
[17]Institute of Coastal Research, Helmholtz-Zentrum Geesthacht (HZG), Geesthacht, Germany
[18]Freie Universität Berlin (FUB), Berlin, Germany
[19]MARUM, Center for Marine Environmental Sciences, Bremen, Germany
[20]Alfred-Wegener-Institut Helmholtz-Zentrum für Polar- und Meeresforschung, Bremerhaven, Germany
[21]Climate Analytics, Berlin, Germany
[22]CEN, University of Hamburg, Meteorological Institute, Hamburg, Germany
[23]Fondazione Centro Euro-Mediterraneo sui Cambiamenti Climatici (CMCC), Bologna, Italy
[24]Department of Mathematics and Statistics, University of Reading, Department of Mathematics and Statistics, Reading, UK
[25]Centre for the Mathematics of Planet Earth, University of Reading, Centre for the Mathematics of Planet Earth Department of Mathematics and Statistics, Reading, UK
[26]Georges Lemaître Centre for Earth and Climate Research, Earth and Life Institute, Université catholique de Louvain, Louvain-la-Neuve, Belgium
[27]Ludwig Maximilians Universität (LMU), Department of Geography, Munich, Germany
[28]NCAS Computational Modelling Services (CMS), University of Reading, Reading, UK
[29]Institute of Marine Sciences, Consiglio Nazionale delle Ricerche (ISMAR-CNR), Italy
[30]Max Planck Institute for Meteorology (MPI-M), Hamburg, Germany
[31]Department of Meteorology, University of Reading, Reading, UK
[32]Department of Environment, Land and Infrastructure Engineering, Politecnico di Torino, Turin, Italy

Correspondence to: Veronika Eyring (veronika.eyring@dlr.de)

**Abstract.** The Earth System Model Evaluation Tool (ESMValTool) is a community diagnostics and performance metrics tool designed to improve comprehensive and routine evaluation of Earth System Models (ESMs) participating in the Coupled Model Intercomparison Project (CMIP). It has undergone rapid development since the first release in 2016 and is now a well-tested tool that provides end-to-end provenance tracking to ensure reproducibility. It consists of an easy-to-install, well documented Python package providing the core functionalities (ESMValCore) that performs common pre-processing operations and a diagnostic part that includes tailored diagnostics and performance metrics for specific scientific applications. Here we describe large-scale diagnostics of the second major release of the tool that supports the evaluation of ESMs participating in CMIP Phase 6 (CMIP6). ESMValTool v2.0 includes a large collection of diagnostics and performance metrics for atmospheric, oceanic, and terrestrial variables for the mean state, trends, and variability. ESMValTool v2.0 also successfully reproduces figures from the evaluation and projections chapters of the Intergovernmental Panel on Climate Change (IPCC) Fifth Assessment Report (AR5) and incorporates updates from targeted analysis packages, such as the NCAR Climate Variability Diagnostics Package for the evaluation of modes of variability the Thermodynamic Diagnostic Tool (TheDiaTo) to evaluate the energetics of the climate system, as well as parts of AutoAssess that contains a mix of top-down performance metrics. The tool has been fully integrated into the Earth System Grid Federation (ESGF) infrastructure at the Deutsches Klima Rechenzentrum (DKRZ) to provide evaluation results from CMIP6 model simulations shortly after the output is published to the CMIP archive. A result browser has been implemented that enables advanced monitoring of the evaluation results by a broad user community at much faster timescales than what was possible in CMIP5.

## 1. Introduction

The Intergovernmental Panel on Climate Change (IPCC) Fifth Assessment Report (AR5) concluded that the warming of the climate system is unequivocal and that the human influence on the climate system is clear (IPCC, 2013). Observed increases of greenhouse gases, warming of the atmosphere and ocean, sea ice decline, and sea level rise, in combination with climate model projections of a likely temperature increase between 2.1 and 4.7°C for a doubling of atmospheric $CO_2$ concentration from pre-industrial (1980) levels, make it an international priority to improve our understanding of the climate system and to reduce greenhouse gas emissions. This is reflected for example in the Paris Agreement of the United Nations Framework Convention on Climate Change (UNFCCC) 21[st] session of the Conference of the Parties (COP21, UNFCCC (2015)).

Simulations with climate and Earth System Models (ESMs) performed by the major climate modelling centres around the world under common protocols are coordinated as part of the World Climate Research Programme (WCRP) Coupled Model Intercomparison Project (CMIP) since the early 90s (Eyring et al., 2016a; Meehl et al., 2000; Meehl et al., 2007; Taylor et al., 2012). CMIP simulations provide a fundamental source for IPCC Assessment Reports and for improving understanding of past, present and future climate change. Standardization of model output in a common format (Juckes et al., 2019) and publication of the CMIP model output on the Earth System Grid Federation (ESGF) facilitates multi-model evaluation and analysis (Balaji et al., 2018; Eyring et al., 2016a; Taylor et al., 2012). This effort is additionally supported by observations for Model Intercomparison Project (obs4MIPs) which provides the community with access to CMIP-like datasets (in terms of variables definitions, temporal and spatial coordinates, time frequencies and coverages) of satellite data

(Ferraro et al., 2015; Teixeira et al., 2014; Waliser et al., 2019). The availability of observations and models in the same format strongly facilitates model evaluation and analysis.

CMIP is now in its 6th phase (CMIP6, Eyring et al. (2016a)) and is confronted with a number of new challenges. More centres are running more versions of more models of increasing complexity. An ongoing demand to resolve more processes requires increasingly higher model resolutions. Accordingly, the data volume of 2 PB in CMIP5 is expected to grow by a factor of 10-20 for CMIP6, resulting in a CMIP6 database of between 20 and 40 PB, depending on model resolution and the number of modelling centres ultimately contributing to the project

(Balaji et al., 2018). Archiving, documenting, subsetting, supporting, distributing, and analysing the huge CMIP6 output together with observations challenges the capacity and creativity of the largest data centres and fastest data networks. In addition, the growing dependency on CMIP products by a broad research community and by national and international climate assessments, as well as the increasing desire for operational analysis in support of mitigation and adaptation, means that systems should be set in place that allow for an efficient and

comprehensive analysis of the large volume of data from models and observations.

To help achieving this, the Earth System Model Evaluation Tool (ESMValTool) is developed. A first version that was tested on CMIP5 models was released in 2016 (Eyring et al., 2016c). With the release of ESMValTool version 2.0 (v2.0), for the first time in CMIP an evaluation tool is now available that provides evaluation results from CMIP6 simulations as soon as the model output is published to the ESGF (https://cmip-

esmvaltool.dkrz.de/). This is realized through text files that we refer to as recipes, each calling a certain set of diagnostics and performance metrics to reproduce analyses that have demonstrated to be of importance in ESM evaluation in previous peer-reviewed papers or assessment reports. The ESMValTool is developed as a community diagnostics and performance metrics tool that allows for routine comparison of single or multiple models, either against predecessor versions or against observations. It is developed as a community-effort

currently involving more than 40 institutes with a rapidly growing developer and user community. Given the level of detailed evaluation diagnostics included in ESMValTool v2.0, several diagnostics are of interest only to the climate modelling community whereas others, including but not limited to those on global mean temperature or precipitation, will also be valuable for the wider scientific user community. The tool allows for full tractability and provenance of all figures and outputs produced. This includes preservation of the netCDF metadata of the

input files including the global attributes. These metadata are also written to the products (netCDF and plots) using the Python package W3C-PROV. Details can be found in the ESMValTool v2.0 technical overview description paper by Righi et al. (2020).

The release of ESMValTool v2.0 is documented in four companion papers: Righi et al. (2020) provide the technical overview of ESMValTool v2.0 and show a schematic representation of the *ESMValCore*, a Python

package that provides the core functionalities, and the *Diagnostic Part* (see their Figure 1). This paper describes recipes of the *Diagnostic Part* for the evaluation of large-scale diagnostics. Recipes for extreme events and in support of regional model evaluation are described by Weigel et al. (2020) and recipes for emergent constraints and model weighting by Lauer et al. (2020). In the present paper, the use of the tool is demonstrated by showing example figures for each recipe for either all or a subset of CMIP5 models. Section 2 describes the type of

modelling and observational data currently supported by ESMValTool v2.0. In Section 3 an overview of the recipes for large-scale diagnostics provided with the ESMValTool v2.0 release is given along with their diagnostics and performance metrics as well as the variables and observations used. Section 4 describes the

workflow of routine analysis of CMIP model output alongside the ESGF and the ESMValTool result browser. Section 5 closes with a summary and an outlook.

## 2. Models and observations

The open-source release of ESMValTool v2.0 that accompanies this paper is intended to work with CMIP5 and CMIP6 model output, and partly also with CMIP3 (although the availability of data for the latter is significantly lower, resulting in a limited number of recipes and diagnostics that can be applied with such data), but the tool is compatible with any arbitrary model output, provided that it is in CF-compliant netCDF format (CF = Climate and Forecast, http://cfconventions.org/) and that the variables and metadata are following the CMOR (Climate Model Output Rewriter, https://pcmdi.github.io/cmor-site/media/pdf/cmor_users_guide.pdf) tables and definitions (see e.g. https://github.com/PCMDI/cmip6-cmor-tables/tree/master/Tables for CMIP6). As in ESMValTool v1.0, for the evaluation of the models with observations, we make use of the large observational effort to deliver long-term, high quality observations from international efforts such as obs4MIPs (Ferraro et al., 2015; Teixeira et al., 2014; Waliser et al., 2019) or observations from the ESA Climate Change Initiative (CCI), Lauer et al. (2017)). In addition, observations from other sources and reanalyses data are used in several diagnostics (see Table 3 in Righi et al. (2020)). The processing of observational data for usage in ESMValTool v2.0 is described in Righi et al. (2020). The observations used by individual recipes and diagnostics are described in Section 3 and listed in Table 1. With the broad evaluation of the CMIP models, the ESMValTool substantially supports one of CMIP's main goals, which is the comparison of the models with observations (Eyring et al., 2016a; Eyring et al., 2019).

## 3. Overview of recipes included in ESMValTool v2.0

In this section, all recipes for large-scale diagnostics that have been newly added in v2.0 since the first release of the ESMValTool in 2016 (see Table 1 in Eyring et al. (2016c) for an overview of namelists (now called recipes) included in v1.0) are described. In each subsection, we first scientifically motivate the inclusion of the recipe by reviewing the main systematic biases in current ESMs and their importance and implications. We then give an overview of the recipes that can be used to evaluate such biases along with the diagnostics and performance metrics included, and the required variables and corresponding observations that are used in ESMValTool v2.0. For each recipe we provide 1-2 example figures that are applied to either all or a subset of the CMIP5 models. An assessment of CMIP5 or CMIP6 models is however not the focus of this paper. Rather, we attempt to illustrate how the recipes contained within ESMValTool v2.0 can facilitate the development and evaluation of climate models in the targeted areas. Therefore, the results of each figure are only briefly described. Table 1 provides a summary of all recipes included in ESMValTool v2.0 along with a short description, information on the quantities and ESMValTool variable names for which the recipe is tested, the corresponding diagnostic scripts and observations. All recipes are included in the ESMValTool repository on GitHub (see Righi et al., 2020 for details) and can be found in the directory https://github.com/ESMValGroup/ESMValTool/tree/master/esmvaltool/recipes.

We describe recipes separately for integrative measures of model performance (Section 3.1) and for the evaluation of processes in the atmosphere (Section 3.2), ocean and cryosphere (Section 3.3), land (Section 3.4),

and biogeochemistry (Section 3.5). Recipes that reproduce chapters from the evaluation chapter of the IPCC
       Fifth Assessment Report (Flato et al., 2013) are described within these sections.

### 3.1 Integrative Measures of Model Performance

### 3.1.1 Performance metrics for essential climate variables for the atmosphere, ocean, sea ice and land

       Performance metrics are quantitative measures of agreement between a simulated and observed quantity.
Various statistical measures can be used to quantify differences between individual models or generations of
       models and observations. Atmospheric performance metrics were already included in
       *namelist_perfmetrics_CMIP5.nml* of ESMValTool v1.0. This recipe has now been extended to include
       additional atmospheric variables as well as new variables from the ocean, sea ice and land. Similar to Figure 9.7
       of Flato et al. (2013), Figure 1 shows the relative space-time root mean square deviation (RMSD) for the CMIP5
historical simulations (1980-2005) against a reference observation and, where available, an alternative
       observational data set [*recipe_perfmetrics_CMIP5.yml*]. Performance varies across CMIP5 models and
       variables, with some models comparing better with observations for one variable and another model performing
       better for a different variable. Except for global average temperatures at 200 hPa (ta_Glob-200) where most but
       not all models have a systematic bias, the multi-model mean outperforms any individual model. Additional
variables can be easily added if observations are available by providing a custom CMOR table and a Python
       script to do the calculations in case of derived variables, see further details in Section 4.1.1 of Eyring et al.
       (2016c). In addition to the performance metrics displayed in Figure 1, several other quantitative measures of
       model performance are included in some of the recipes and are described throughout the respective sections of
       this paper.

### 3.1.2. Centered pattern correlations for different CMIP ensembles

       Another example of a performance metric is the pattern correlation between the observed and simulated
       climatological annual mean spatial patterns. Following Figure 9.6 of the IPCC AR5 (Chapter 9, Flato et al.
       (2013)), a diagnostic for computing and plotting centered pattern correlations for different models and CMIP
       ensembles has been implemented (Figure 2) and added to *recipe_flato13ipcc.yml*. The variables are first
regridded to a 4° x 5° longitude by latitude grid to avoid favouring a specific model resolution. Regridding is
       done by the Iris package which offers different regridding schemes (see
       https://esmvaltool.readthedocs.io/projects/esmvalcore/en/latest/recipe/preprocessor.html#horizontal-regridding).
       The figure shows both a large model spread as well as a large spread in the correlation depending on the
       variable, signifying that some aspects of the simulated climate agree better with observations than others. The
centered pattern correlations, which measure the similarity of two patterns after removing the global mean, are
       computed against a reference observation. Should the input models be from different CMIP ensembles, they are
       grouped by ensemble and each ensemble is plotted side by side for each variable with a different colour. If an
       alternate model is given, it is shown as a solid green circle. The axis ratio of the plot reacts dynamically to the
       number of variables (n_var) and ensembles (n_ensemble) after it surpasses a combined number of
n_var*n_ensemble = 16, and the y-axis range is calculated to encompass all values. The centered pattern
       correlation is good to see both the spread in models within a single variable, as well as a quick overview of how
       well other variables and aspects of the climate on a large scale are reproduced with respect to observations.

Furthermore when using several ensembles, the progress made by each ensemble on a variable basis can be seen at a quick glance.

### 3.1.3 Single model performance index

Most model performance metrics only display the skill for a specific model and a specific variable at a time, not making an overall index for a model. This works well when only a few variables or models are considered, but can result in an overload of information for a multitude of variables and models. Following Reichler and Kim (2008), a Single Model Performance Index (SMPI) has been implemented in *recipe_smpi.yml*. The SMPI (called "I2") is based on the comparison of several different climate variables (atmospheric, surface and oceanic) between climate model simulations and observations or reanalyses, and evaluates the time-mean state of climate. For I2 to be determined, the differences between the climatological mean of each model variable and observations at each of the available data grid points are calculated, and scaled to the interannual variance from the validating observations. This interannual variability is determined by performing a bootstrapping method (random selection with replacement) for the creation of a large synthetic ensemble of observational climatologies. The results are then scaled to the average error from a reference ensemble of models, and in a final step the mean over all climate variables and one model is calculated. Figure 3 shows the I2 values for each model (orange circles) and the multi-model mean (black circle), with the diameter of each circle representing the range of I2 values encompassed by the 5th and 95th percentiles of the bootstrap ensemble. The SMPI allows for a quick estimation on which models are performing the best on average across the sampled variables (see Table 1) and in this case shows that the common practice of taking the multi-model mean as best overall model is valid. The I2 values vary around one, with values greater than one for underperforming models, and values less than one for more accurate models. This diagnostic requires that all models have input for all of the variables considered, as this is the basis to have a meaningful comparison of the resulting I2 values.

### 3.1.4 Auto-Assess

While highly condensed metrics are useful for comparing a large number of models, for the purpose of model development it is important to retain granularity on which aspects of model performance have changed, and why. For this reason, many modelling centres have their own suite of metrics which they use to compare candidate model versions against a predecessor. AutoAssess is such a system, developed by the UK Met Office and used in the development of the HadGEM3 and UKESM1 models. The output of AutoAssess contains a mix of top-down metrics evaluating key model output variables (e.g. temperature and precipitation) and bottom-up metrics which assess the realism of model processes and emergent behaviour such as cloud variability and El Niño–Southern Oscillation (ENSO). The output of AutoAssess includes around 300 individual metrics. To facilitate interpretation of the results, these are grouped into 11 thematic areas, ranging from the broad-scale such as global tropic circulation and stratospheric mean state and variability, to the region- and process-specific, such as monsoon regions and the hydrological cycle.

It is planned that all the metrics currently in AutoAssess will be implemented in ESMValTool. At this time, a single assessment area (group of metrics) has been included as a technical demonstration: that for the stratosphere. These metrics have been implemented in a set of recipes named *recipe_autoassess_*.yml*. They include metrics of the Quasi-Biennial Oscillation (QBO) as a measure of tropical variability in the stratosphere. Zonal mean zonal wind at 30 hPa is used to define metrics for the period and amplitude of the QBO. Figure 4

displays the downward propagation of the QBO for a single model using zonal mean zonal wind averaged between 5°S and 5°N. Zonal wind anomalies propagate downward from the upper stratosphere. The figure shows that the period of the QBO in the chosen model is about 6 years, significantly longer than the observed period of ~2.3 years. Metrics are also defined for the tropical tropopause cold point (100 hPa, 10°S – 10°N) temperature, and stratospheric water vapour concentrations at entry point (70 hPa, 10°S – 10°N). The cold point temperature is important in determining the entry point humidity, which in turn is important for the accurate simulation of stratospheric chemistry and radiative balance (Hardiman et al., 2015). Other metrics characterise the realism of the stratospheric easterly jet and polar night jet.

### 3.2 Diagnostics for the evaluation of processes in the atmosphere

#### 3.2.1 Multi-model mean bias for temperature and precipitation

Near-surface air temperature (tas) and precipitation (pr) of ESM simulations are the two variables most commonly requested by users. Often, diagnostics for tas and pr are shown for the multi-model mean of an ensemble. Both of these variables are the end result of numerous interacting processes in the models, making it challenging to understand and improve biases in these quantities. For example, near surface air temperature biases depend on the models' representation of radiation, convection, clouds, land characteristics, surface fluxes, as well as atmospheric circulation and turbulent transport (Flato et al., 2013), each with their own potential biases that may either augment or oppose one another.

The diagnostic that calculates the multi-model mean bias compared to a reference data set is part of *recipe_flato13ipcc.yml* and reproduces Figures 9.2 and 9.4 of Flato et al. (2013). We extended the *namelist_flato13ipcc.xml* of ESMValTool v1.0 by adding the mean root mean square error of the seasonal cycle with respect to the reference dataset. The multi-model mean near-surface temperature agrees with ERA-Interim mostly within ±2°C (Figure 5). Larger biases can be seen in regions with sharp gradients in temperature, for example in areas with high topography such as the Himalaya, the sea ice edge in the North Atlantic, and over the coastal upwelling regions in the subtropical oceans. Biases in the simulated multi-model mean precipitation compared to Global Precipitation Climatology Project (GPCP, Adler et al. (2003)) data include too low precipitation along the Equator in the western Pacific and too high precipitation amounts in the tropics south of the Equator (Figure 6). Figure 7 shows observed and simulated time series of the anomalies in annual and global mean surface temperature. The model datasets are subsampled by the HadCRUT4 observational data mask (Morice et al., 2012) and pre-processed as described by Jones et al. (2013). Overall, the models represent the annual global-mean surface temperature increase over the historical period quite well, including the more rapid warming in the second half of the 20[th] century and the cooling immediately following large volcanic eruptions. The figure reproduces Figure 9.8 of Flato et al. (2013) and is part of *recipe_flato13ipcc.yml*.

#### 3.2.2 Precipitation quantile bias

Precipitation is a dominant component of the hydrological cycle, and as such a main driver of the climate system and human development. The reliability of climate projections and water resources strategies therefore depends on how well precipitation can be simulated by the models. While CMIP5 models can reproduce the main patterns of mean precipitation (e.g., compared to observational data from GPCP (Adler et al., 2003)), they often show shortages and biases under particular conditions. Comparison of precipitation from CMIP5 models and observations shows a general good agreement for mean values at large scale (Kumar et al., 2013; Liu et al.,

2012a). Models have however a poor representation of frontal, convective, and mesoscale processes, resulting in substantial biases at regional scale (Mehran et al., 2014): models tend to overestimate precipitation over complex topography and underestimate it especially over arid or some subcontinental regions as for example northern Eurasia, eastern Russia, and central Australia. Biases are typically stronger at high quantiles of precipitation, making the study of precipitation quantile biases an effective diagnostic for addressing the quality of simulated precipitation.

The *recipe_quantilebias.yml* implements the calculation of the quantile bias to allow for the evaluation of precipitation biases based on a user defined quantile in models as compared to a reference dataset following Mehran et al. (2014). The quantile bias is defined as the ratio of monthly precipitation amounts in each simulation to that of the reference dataset above a specified threshold $t$ (e.g., the $75^{th}$ percentile of all the local monthly values). An example is displayed in Figure 8, where gridded observations from the GPCP project were adopted. A quantile bias equal to 1 indicates no bias in the simulations, whereas a value above (below) 1 corresponds to a model's overestimation (underestimation) of the precipitation amount above the specified threshold $t$, with respect to that of the reference dataset. An overestimation over Africa for models in the right column and an underestimation crossing central Asia from Siberia to the Arabic peninsula is visible, promptly identifying the best performances or outliers. For example, the HadGEM2-ES model here shows a smaller bias compared to the other models in this subset. The recipe allows the evaluation of the precipitation bias based on a user-defined quantile in models as compared to the reference dataset.

### 3.2.3 Atmospheric dynamics

*3.2.3.1 Stratosphere-troposphere coupling*

The current generation of climate models include the representation of stratospheric processes, as the vertical coupling with the troposphere is important for the representation of weather and climate at the surface (Baldwin and Dunkerton, 2001). Stratosphere-resolving models are able to internally generate realistic annular modes of variability in the extratropical atmosphere (Charlton-Perez et al., 2013) which are however too persistent in the troposphere and delayed in the stratosphere compared to reanalysis (Gerber et al., 2010), leading to biases in the simulated impacts on surface conditions.

The recipe *recipe_zmnam.yml* can be used to evaluate the representation of the Northern Annular Mode (NAM, Wallace (2000)) in climate simulations, using reanalysis datasets as reference. The calculation is based on the "zonal mean algorithm" of Baldwin and Thompson (2009), and is an alternative to pressure based or height-dependent methods. This approach provides a robust description of the stratosphere-troposphere coupling on daily timescales, requiring less subjective choices and a reduced amount of input data. Starting from daily mean geopotential height on pressure levels, the leading Empirical Orthogonal Functions (EOFs) / principal components are computed from linearly detrended zonal mean daily anomalies, with the principal component representing the zonal mean NAM index. Missing values, which may occur near the surface level, are filled with a bilinear interpolation procedure. The regression of the monthly mean geopotential height onto this monthly averaged index represents the NAM pattern for each selected pressure level. The outputs of the procedure are the time series (Figure 9, left) and the histogram (not shown) of the zonal-mean NAM index, and the regression maps for selected pressure levels (Figure 9, right). The well-known annular pattern, with opposite anomalies between polar and mid-latitudes, can be seen in the regression plot. The user can select the specific datasets (climate model simulation and/or reanalysis) to be evaluated, and a subset of pressure levels of interest.

*3.2.3.2 Atmospheric blocking indices*

Atmospheric blocking is a recurrent mid-latitude weather pattern identified by a large-amplitude, quasi-stationary, long-lasting, high-pressure anomaly that ''blocks'' the westerly flow forcing the jet stream to split or meander (Rex, 1950). It is typically initiated by the breaking of a Rossby wave in a region at the exit of the storm track, where it amplifies the underlying stationary ridge (Tibaldi and Molteni, 1990). Blocking occurs more frequently in the Northern Hemisphere cold season, with larger frequencies observed over the Euro-Atlantic and North Pacific sectors. Its lifetime oscillates from a few days up to several weeks (Davini et al., 2012). Atmospheric blocking still represents an open issue for the climate modelling community since state-of-the-art weather and climate models show limited skill in reproducing it (Davini and D'Andrea, 2016; Masato et al., 2013). Models are indeed characterized by large negative bias over the Euro-Atlantic sector, a region where blocking is often at the origin of extreme events, leading to cold spells in winter and heat waves in summer (Coumou and Rahmstorf, 2012; Sillmann et al., 2011).

Several objective blocking indices have been developed aimed at identifying different aspects of the phenomenon (see Barriopedro et al. (2010) for details). The recipe *recipe_miles_block.yml* integrates diagnostics from the Mid-Latitude Evaluation System (MiLES) v0.51 (Davini, 2018) tool in order to calculate two different blocking indices based on the reversal of the meridional gradient of daily 500 hPa geopotential height. The first one is a 1-d index, namely the Tibaldi and Molteni (1990) blocking index, here adapted to work with 2.5° x 2.5° grids. Blocking is defined when the reversal of the meridional gradient of geopotential height at 60°N is detected, i.e. when easterly winds are found in the mid-latitudes. The second one is the atmospheric blocking index following Davini et al. (2012). It is a 2-d extension of Tibaldi and Molteni (1990) covering latitudes from 30°N up to 75°N. The recipe computes both the Instantaneous Blocking frequencies and the Blocking Events frequency (which includes both spatial and 5-day-minimum temporal constraints). It reports also two intensity indices, namely the Meridional Gradient Index and the Blocking Intensity index, and it evaluates the wave breaking characteristic associated with blocking (cyclonic or anticyclonic) through the Rossby wave orientation index. A supplementary Instantaneous Blocking index (named "ExtraBlock") including an extra condition to filter out low-latitude blocking events is also provided. The recipe compares multiple datasets against a reference one (default is ERA-Interim) and provides output (in NetCDF4 compressed Zip format) as well as figures for the climatology of each diagnostic. An example output is shown in Figure 10. The MPI-ESM-MR model shows the well-known underestimation of atmospheric blocking – typical of many climate models – over Central Europe, where blocking frequencies are about the half when compared to reanalysis. Slight overestimation of low latitude blocking and North Pacific blocking can also be seen, while Greenland blocking frequencies show negligible bias.

**3.2.4 Thermodynamics of the climate system**

The climate system can be seen as a forced and dissipative non-equilibrium thermodynamic system (Lucarini et al., 2014), converting potential into mechanical energy, and generating entropy via a variety of irreversible processes The atmospheric and oceanic circulation are caused by the inhomogeneous absorption of solar radiation, and, all in all, they act in such a way to reduce the temperature gradients across the climate system. At steady-state, assuming stationarity, the long term global energy input and output should balance. Previous studies have shown that this is essentially not the case, and most of the models are affected by non-negligible energy drift (Lucarini et al., 2011; Mauritsen et al., 2012). This severely impacts the prediction capability of

state-of-the-art models, given that most of the energy imbalance is known to be taken up by oceans (Exarchou et al., 2015). Global energy biases are also associated with inconsistent thermodynamic treatment of processes taking place in the atmosphere, as the dissipation of kinetic energy (Lucarini et al., 2011) and the water mass balance inside the hydrological cycle (Liepert and Previdi, 2012; Wild and Liepert, 2010). Climate models

feature substantial disagreements in the peak intensity of the meridional heat transport, both in the ocean and in the atmospheric parts, whereas the position of the peaks of the (atmospheric) transport blocking are consistently captured (Lucarini and Pascale, 2014). In the atmosphere, these issues are related to inconsistencies in the models' ability to reproduce the mid-latitude atmospheric variability (Di Biagio et al., 2014; Lucarini et al., 2007) and intensity of the Lorenz Energy Cycle (Marques et al., 2011). Energy and water mass budgets, as well

as the treatment of the hydrological cycle and atmospheric dynamics, all affect the material entropy production in the climate system, i.e. the entropy production related to irreversible processes in the system. It is possible to estimate the entropy production either via an indirect method, based on the radiative heat convergence in the atmosphere (the ocean accounts only for a minimal part of the entropy production), or via a direct method, based on the explicit computation of entropy production due to all irreversible processes (Goody, 2000). Differences in

the two methods emerge when considering coarse-grained data in space and/or in time (Lucarini and Pascale, 2014), as subgrid-scale processes have long been known to be a critical issue, when attempting to provide an accurate climate entropy budget (Gassmann and Herzog, 2015; Kleidon and Lorenz, 2004; Kunz et al., 2008). When possible (energy budgets, water mass and latent energy budgets, components of the material entropy production with the indirect method) horizontal maps for the average of annual means are provided. For the

Lorenz Energy Cycle, a flux diagram (Ulbrich and Speth, 1991), showing all the storage, conversion, source and sink terms for every year, is provided. The diagram in Figure 11 shows the baroclinic conversion of the available potential energy (APE) to kinetic energy (KE), and ultimately its dissipation through frictional heating (Lorenz, 1955; Lucarini et al., 2014). When a multi-model ensemble is provided, global metrics are related in scatter plots, where each dot is a member of the ensemble, and the multi-model mean, together with uncertainty range,

is displayed. An output log file contains all the information about the time-averaged global mean values, including all components of the material entropy production budget. For the meridional heat transports, annual mean meridional sections are shown in Figure 12 (Lembo et al., 2017; Lucarini and Pascale, 2014; Trenberth et al., 2001). The model spread has roughly the same magnitude in the atmospheric and oceanic transports, but its relevance is much larger for the oceanic transports. The model spread is also crucial in the magnitude and sign of

the atmospheric heat transports across the Equator, given its implications for the atmospheric general circulation. The diagnostic tool is run through the recipe *recipe_thermodyn_diagtool.yml*, where the user can also specify the options on which modules should be run.

### 3.2.5 Natural modes of climate variability and weather regimes

*3.2.5.1. NCAR Climate Variability Diagnostic Package*

Natural modes of climate variability co-exist with externally-forced climate change, and have large impacts on climate, especially at regional and decadal scales. These modes of variability are due to processes intrinsic to the coupled climate system, and exhibit limited predictability. As such, they complicate model evaluation as the observational record is often not long enough to reliably assess the variability, and confound assessments of anthropogenic influences on climate (Bengtsson and Hodges, 2019; Deser et al., 2012; Deser et al., 2014; Deser

et al., 2017; Kay et al., 2015; Suárez-Gutiérrez et al., 2017). Despite their importance, systematic evaluation of

these modes in Earth system models remains a challenge due to the wide range of phenomena to consider, the length of record needed to adequately characterize them, and uncertainties in the short observational data sets (Deser et al., 2010; Frankignoul et al., 2017; Simpson et al., 2018). While the temporal sequences of internal variability in models do not necessarily need to match those in the single realization of nature, their statistical

properties (e.g., time scale, autocorrelation, spectral characteristics, and spatial patterns) need to be realistically simulated for credible climate projections.

In order to assess natural modes of climate variability in models, the NCAR Climate Variability Diagnostics Package (CVDP, Phillips et al. (2014)) has been implemented into the ESMValTool. The CVDP has been developed as a standalone tool. To allow for easy updating of the CVDP once a new version is released, the

structure of the CVDP is kept in its original form and a single recipe *recipe_CVDP.yml* has been written to enable the CVDP to be run directly within ESMValTool. The CVDP facilitates evaluation of the major modes of climate variability, including ENSO (Deser et al., 2010), the Pacific Decadal Oscillation (PDO, (Deser et al., 2010; Mantua et al., 1997)), the Atlantic Multi-decadal Oscillation (AMO, Trenberth and Shea (2006)), the Atlantic Meridional Overturning Circulation (AMOC, Danabasoglu et al. (2012)), and atmospheric

teleconnection patterns such as the Northern and Southern Annular Modes (NAM and SAM; (Hurrell and Deser, 2009; Thompson and Wallace, 2000)), North Atlantic Oscillation (NAO, Hurrell and Deser (2009)), and Pacific North and South American (PNA and PSA, Thompson and Wallace (2000)), patterns. For details on the actual calculation of these modes in CVDP we refer to the original CVDP package and explanations available at http://www.cesm.ucar.edu/working_groups/CVC/cvdp/.

Depending on the climate mode analysed, the CVDP package uses the following variables: precipitation (pr), sea level pressure (psl), near-surface air temperature (tas), skin temperature (ts), snow depth (snd), sea ice concentration (siconc), and basin-average ocean meridional overturning mass stream function (msftmz). The models are evaluated against a wide range of observations and reanalysis data, for example Berkeley Earth System Temperature (BEST) for near-surface air temperature, Extended Reconstructed Sea Surface Temperature

v5 (ERSSTv5) for skin temperature, and ERA-20C extended with ERA-Interim for sea level pressure. Additional observations or reanalysis can be added by the user for these variables. The ESMValTool v2.0 recipe runs on all CMIP5 models. As an example, Figure 13 shows the representation of ENSO teleconnections during the peak phase (December-February). Models produce a wide range of ENSO amplitudes and teleconnections. Note that even when based on over 100 years of record, the ENSO composites are subject to uncertainty due to

sampling variability (Deser et al., 2017). Figure 14 shows the representation of the AMO as simulated by 41 CMIP5 models and observations during the historical period. The pattern of SSTA* associated with the AMO is generally realistically simulated by models within the North Atlantic basin, although its amplitude varies. However, outside of the North Atlantic, the models show a wide range of spatial patterns and polarities of the AMO.

*3.2.5.2 Weather regimes*

Weather Regimes (WRs) refer to recurrent large-scale atmospheric circulation structures that allow the characterization of complex atmospheric dynamics in a particular region (Michelangeli et al., 1995; Vautard, 1990). The identification of WRs reduces the continuum of atmospheric circulation to a few recurrent and quasi-stationary (persistent) patterns. WRs have been extensively used to investigate atmospheric variability at the

mid-latitudes, as they are associated with extreme weather events such as heat waves or droughts (Yiou et al., 2008). For example, there is a growing recognition of their significance especially over the Euro-Atlantic sector

during the winter season, where four robust weather regimes have been identified - namely the NAO+, NAO-, Atlantic Ridge and Scandinavian Blocking (Cassou et al., 2005). These WRs can also be used as a diagnostic to investigate the performance of state-of-the-art climate forecast systems: difficulties in reproducing the Atlantic

Ridge and the Scandinavian blocking have been often reported (Dawson et al., 2012; Ferranti et al., 2015). Forecast systems which are not able to reproduce the observed spatial patterns and frequency of occurrence of WRs may have difficulties in reproducing climate variability and its long-term changes (Hannachi et al., 2017). Hence, the assessment of WRs can help improve our understanding of predictability on intra-seasonal to inter-annual time scales. In addition, the use of WRs to evaluate the impact of the atmospheric circulation on essential

climate variables and sectoral climatic indices is of great interest to the climate services communities (Grams et al., 2017). The diagnostic can be applied to model simulations under future scenarios as well. However, caution must be applied since large changes in the average climate, due to large radiative forcing, might affect the results and lead to somehow misleading conclusions. In such cases further analysis will be needed to assess to what extent the response to climate change projects on the regimes patterns identified by the tool in the historical and

future periods and to verify how future anomalies project onto the regime patterns identified in the historical period.

The recipe *recipe_modes_of_variability.yml* takes daily or monthly data from a particular region, season (or month) and period as input, and then applies k-mean clustering or hierarchical clustering either directly to the spatial data or after computing the EOFs. This recipe can be run for both a reference/observational dataset and

climate projections simultaneously, and the root mean square error is then calculated between the mean anomalies obtained for the clusters from the reference and projection data sets. The user can specify the number of clusters to be computed. The recipe output consist of netCDF files of the time series of the cluster occurrences, the mean anomaly corresponding to each cluster at each location and the corresponding *p*-value, for both the observed and projected WR and the RMSE between them. The recipe also creates three plots: the

observed/reference modes of variability (Figure 15), the reassigned modes of variability for the future projection (Figure 16) and a table displaying the RMSE values between reference and projected modes of variability (Figure 17). Low RMSE values along the diagonal show that the modes of variability simulated by the future projection (Figure 16) match the reference modes of variability (Figure 15). The recipe *recipe_miles_regimes.yml* integrates the diagnostics from the Mid-Latitude Evaluation System (MiLES) v0.51

tool (Davini, 2018) in order to calculate the four relevant North Atlantic weather regimes. This is done by analysing the 500 hPa geopotential height over the North Atlantic (80°W – 40°E, 30°N - 87.5°N). Once a 5-day smoothed daily seasonal cycle is removed, the EOFs which explain at least the 80% of the variance are extracted in order to reduce the phase-space dimensions. A k-means clustering using Hartigan-Wong algorithm with k=4 is then applied providing the final weather regimes identification. The recipe compares multiple datasets against a

reference one (default is ERA-Interim) producing multiple figures which show the pattern of each regime and its difference against the reference dataset. Weather regimes patterns and timeseries are provided in NetCDF4 compressed Zip format. Considering the limited physical significance of Euro-Atlantic weather regimes in other seasons, only winter is currently supported. An example output is shown in Figure 18. The Atlantic ridge regime, which is usually badly simulated by climate models, is reproduced with the right frequency of occupancy and

pattern in MPI-ESM-MR when compared to ERA-Interim reanalysis.

### 3.2.5.3 Empirical Orthogonal Functions

EOF analysis is a powerful method to decompose spatiotemporal data using an orthogonal basis of spatial patterns. In weather sciences, EOFs have been extensively used to identify the most important modes of climate variability and their associated teleconnection patterns: for instance, the North Atlantic Oscillation (NAO, (Ambaum, 2010; Wallace and Gutzler, 1981)) and the Arctic Oscillation (AO, Thompson and Wallace (2000)) are usually defined with EOFs. Biases in the representation of the NAO or the AO have been found to be typical in many CMIP5 models (Davini and Cagnazzo, 2013).

The recipe *recipe_miles_eof.yml* integrates diagnostics from the MiLES v0.51 tool (Davini, 2018) in order to extract the first EOFs over a user-defined domain. Three default patterns are supported, namely the "NAO" (North Atlantic Oscillation, over the 90°W – 40°E, 20°N – 85°N box), the "PNA" (Pacific North America pattern, over the 140°W – 80°E, 20°N – 85°N box) and the "AO" (Arctic Oscillation, over the 20°N – 85°N box). The computation is based on Singular-Value Decomposition (SVD) applied to the anomalies of the monthly 500 hPa geopotential height. The recipe compares multiples datasets against a reference one (default is ERA-Interim) producing multiple figures which show the linear regressions of the Principal Component (PC) of each EOF on the monthly 500hPa geopotential and its differences against the reference dataset. By default the first four EOFs are stored and plotted. As an example, Figure 19 shows that the NAO is well represented by the MPI-ESM-LR model (that is used here for illustration), although the variance explained is underestimated and the northern center of action, which is found close to Iceland in reanalysis, is westward displaced over Greenland.

### 3.2.5.4 Indices from differences between area averages

In addition to indices and modes of variability obtained from EOF and clustering analyses, users may wish to compute their own indices based on area-weighted averages or difference in area-weighted averages. For example, the Niño 3.4 index is defined as the sea surface temperature (SST) anomalies averaged over [170–120°W, 5°N - 5°S]. Similarly, the NAO index can be defined as the standardized difference between weighted area-average mean sea level pressure of the domain bounded by [0–80° W, 30–50° N] and [0–80° W 60–80°N].

The functions for computing indices based on area averages in *recipe_combined_indices.yml* have been adapted to allow users to compute indices for the Niño 3, Niño 3.4, Niño 4, NAO and the Southern Oscillation Index (SOI) defined region(s), with the option of selecting different variables (e.g. temperature of the ocean surface (tos, commonly named sea surface temperature) or pressure at sea level (psl, sea level pressure)) with the option to compute standardized variables, applying running means and select different seasons by selecting the start and end months. The output of this recipe is a netCDF file containing a time series of the computed indices and a time series of the evolution of the index for individual models and the multi-model mean (see Figure 20).

## 3.3 Diagnostics for the evaluation of processes in the ocean and cryosphere

### 3.3.1 Physical ocean

The global ocean is a core component of the Earth system. A significant bias in the physical ocean can impact the performance of the entire model. Several diagnostics exist in ESMValTool v2.0 to evaluate the broad behaviour of models of the global ocean. Figures 21 to 26 show several diagnostics of the ability of the CMIP5 models to simulate the global ocean. All available CF-compliant CMIP5 models are compared, however each

figure shown in this section may include a different set of models, as not all CMIP5 models produced all the required datasets in a CF-compliant format. To minimise noise, these figures are shown with a 6-year moving window average.

The volume weighted global average temperature anomaly of the ocean is shown in Figure 21 and displays the change in the mean temperature of the ocean relative to the start of the historical simulation. The temperature

anomaly is calculated against the years 1850-1900. Nearly all CMIP5 models show an increase in the mean temperature of the ocean over the historical period. This figure was produced using the recipe *recipe_ocean_scalar_fields.yml*. The AMOC is an indication of the strength of the overturning circulation in the Atlantic Ocean and is shown in Figure 22. It transfers heat from tropical waters to the Northern Atlantic ocean. The AMOC has an observed strength of 17.2 Sv (McCarthy et al., 2015). In the example shown in Figure 22, all

CMIP5 models show some interannual variability in the AMOC behaviour, but the decline in the multi-model mean over the historical period is not statistically significant. Previous modelling studies (Cheng et al., 2013; Gregory et al., 2005) have predicted a decline in the strength of the AMOC over the 20th century. The Drake Passage current is a measure of the strength of the Antarctic Circumpolar Current (ACC). This is the strongest current in the global ocean and runs clockwise around Antarctica. The ACC was recently measured through the

Drake Passage at 173.3±10.7 Sv (Donohue et al., 2016). Four of the CMIP5 models fall within this range (Figure 23). Figures 22 and 23 were produced using the recipe *recipe_ocean_amocs.yml*. The global total flux of $CO_2$ from the atmosphere into the ocean for several CMIP5 models is shown in Figure 24. This figure shows the absorption of atmospheric carbon by the ocean. At the start of the historic period, most of the models shown here have been spun up, meaning that the air to sea flux of $CO_2$ should be close to zero. As the $CO_2$ concentration in

the atmosphere increases over the course of the historical simulation, the flux of carbon from the air into the sea also increases. The CMIP5 models shown in Figure 24 agree very closely on the behaviour of the air to sea flux of $CO_2$ over the historical period with all models showing an increase from close to zero, and rising up to approximately 2 Pg of Carbon per year (C $yr^{-1}$) by the start of the 21st century. The global total integrated primary production from phytoplankton is shown in Figure 25. Marine phytoplankton is responsible for 56±7 Pg

C $yr^{-1}$ of primary production (Buitenhuis et al., 2013), which is of similar magnitude to that of land plants (Field et al., 1998). In all cases, we do not expect to observe a significant change in primary production over the course of the historical period. However, the differences in the magnitude of the total integrated primary production inform us about the level of activity of the marine ecosystem. All CMIP5 models in Figure 25 show little inter-annual variability in the integrated marine primary production, and there is no clear trend in the multi-model

mean. Figure 24 and 25 were both produced with the recipe *recipe_ocean_scalar_fields.yml*. The combination of these five key time series figures allows a coarse scale evaluation of the ocean circulation and biogeochemistry. The global volume weighted temperature shows the effect of a warming ocean, while the change in the Drake Passage and the AMOC show significant global changes in circulation. The integrated primary production shows changes in marine productivity and the air sea flux of $CO_2$ shows the absorption of anthropogenic atmospheric

carbon by the ocean.

In addition, a diagnostic from Chapter 9 of IPCC AR5 for the ocean is added (Flato et al., 2013) which is included in *recipe_flato13ipcc.yml*. Figure 26 shows an analysis of the SST that documents the performance of models compared to one standard observational dataset, namely the SST part of the Hadley Centre Sea Ice and Sea Surface Temperature (HadISST) (Rayner et al., 2003) dataset. The SST plays an important role in climate

simulations because it is the main oceanic driver of the atmosphere. As such, a good model performance for SST

has long been a hallmark of accurate climate projections. In this figure we reproduce Figure 9.14 of Flato et al. (2013). It shows both zonal mean and equatorial (averaged over 5°S to 5°N) SST. For the zonal mean it shows (a) the error compared to observations for the individual models, (c) the multi-model mean with the standard deviation. For the equatorial average it shows (b) the individual model errors and (d) the multi-model mean of the temperatures together with the observational dataset. In this way a good overview of both the error and the absolute temperatures can be provided for the individual model level. Figure 26 shows the overall good agreement of the CMIP5 models among themselves as well as compared to observations, but also highlights the global areas with largest uncertainty and biggest room for improvement. This is an important benchmark for the upcoming CMIP6 ensemble.

### 3.3.2 Southern ocean

The Southern ocean is central to the global climate and the global carbon cycle, and to the climate's response to increasing levels of atmospheric greenhouse gases, as it ventilates a large fraction of the global ocean volume. Roemmich et al. (2015) concluded that the Southern Ocean was responsible for 67-98% of the total oceanic heat uptake; the oceanic increase in heat accounts for 93% of the radiative imbalance at the top of the atmosphere. Global coupled climate models and Earth system models, however, vary widely in their simulations of the Southern Ocean and its role in and response to anthropogenic forcing. Due to the region's complex water-mass structure and dynamics, Southern Ocean carbon and heat uptake depend on a combination of winds, eddies, mixing, buoyancy fluxes, and topography. Russell et al. (2018) laid out a series of diagnostic, observational-based metrics that highlight biases in critical components of the Southern Hemisphere climate system, especially those related to the uptake of heat and carbon by the ocean. These components include the surface fluxes (including wind and heat and carbon), the frontal structure, the circulation and transport within the ocean, the carbon system (in the ESMs) and the sea ice simulation. Each component is associated with one or more model diagnostics, and with relevant observational data sets that can be used for the model evaluation. Russell et al. (2018) noted that biases in the strength and position of the surface westerlies over the Southern Ocean were indicative of biases in several other variables. The strength, extent, and latitudinal position of the Southern Hemisphere surface westerlies are crucial to the simulation of the circulation, vertical exchange and overturning, and heat and carbon fluxes over the Southern Ocean. The net transfer of wind energy to the ocean depends critically on the strength and latitudinal structure of the winds. Equatorward-shifted winds are less aligned with the latitudes of the Drake Passage and are situated over shallower isopycnal surfaces, making them less effective at both driving the ACC and bringing dense deep water up to the surface.

Figure 27 shows the annually-averaged, zonally-averaged zonal wind stress over the Southern Ocean from a sample of the CMIP5 climate simulations and the equivalent quantity from the Climate Forecast System Reanalysis (Saha et al., 2013). While most model metrics indicate that simulations generally bracket the observed quantity, this metric indicates that ALL of the models have an equatorward bias relative to the observations, an indication of a deeper modelling issue. Although Russell et al. (2018) only included six of the simulations submitted as part of CMIP5, the recipe *recipe_russell18jgr.yml* will recreate all of the metrics of this study for all CMIP5 simulations. Each metric assesses a simulated variable, or a climatically-relevant quantity calculated from one or more simulated variables (e.g. heat content is calculated from the simulated ocean temperature, thetao, while the meridional heat transport depends on both the temperature, thetao, and the meridional velocity, vo) relative to the observations. The recipe focuses on factors affecting the simulated heat

and carbon uptake by the Southern Ocean. Figure 28 shows the relationship between the latitudinal width of the surface westerly winds over the Southern Ocean with the net heat uptake south of 30°S – the correlation (-0.8) is significant above the 98% level.

### 3.3.3 Arctic ocean

The Arctic ocean is one of the areas of the Earth where the effects of climate change are especially visible today. Two most prominent processes are Arctic atmospheric temperature warming amplification (Serreze and Barry, 2011) and decrease of the sea ice area and thickness (see Section 3.3.2). Both receive good coverage in the literature and are already well-studied. Much less attention is paid to the interior of the Arctic Ocean itself. In order to increase our confidence in projections of the Arctic climate future proper representation of the Arctic

Ocean hydrography is necessary.

The vertical structure of temperature and salinity (T and S) in the ocean model is a key diagnostic that is used for ocean model evaluation. Realistic temperature and salinity distributions mean that the models properly represent dynamic and thermodynamic processes in the ocean. Different ocean basins have different hydrological regimes so it is important to perform analysis of vertical TS distribution for different basins separately. The basic

diagnostics in this sense are the mean vertical profiles of temperature and salinity over some basin averaged for a relatively long period of time. Figure 29 shows the mean (1970-2005) vertical ocean potential temperature distribution in the Eurasian Basin of the Arctic Ocean as produced with *recipe_arctic_ocean.yml*. It shows that CMIP5 models tend to overestimate temperature in the interior of the Arctic Ocean and have too deep Atlantic water depth. In addition to individual vertical profiles for every model, we also show the mean over all

participating models and similar profile from climatological data (PHC3, Steele et al. (2001)). The characteristics of vertical TS distribution can change with time, and consequently the vertical TS distribution is an important indicator of the behaviour of the coupled ocean-sea ice-atmosphere system in the North Atlantic and Arctic Oceans. One way to evaluate these changes is by using Hovmoller diagrams. We have created Hovmoller diagrams for two main Arctic Ocean basins – Eurasian and Amerasian (as defined in Holloway et al. (2007)),

with T and S spatially averaged on a monthly basis for every vertical level. This diagnostic allows the temporal evolution of vertical ocean potential temperature distribution to be assessed. The T-S diagrams allow the analysis of water masses and their potential for mixing. The lines of constant density for specific ranges of temperature and salinity are shown on the background of the T-S diagram. The dots on the diagram are individual grid points from specified region at all model levels within user specified depth range. The depths are colour coded.

Examples of the mean (1970-2005) T-S diagram for Eurasian Basin of the Arctic Ocean shown in Figure 30 refer to *recipe_arctic_ocean.yml*. Most models cannot properly represent Arctic Ocean water masses and either have wrong values for temperature and salinity or miss specific water masses completely.

The spatial distribution of basic oceanographic variables characterises the properties and spreading of ocean water masses. For the coupled models, capturing the spatial distribution of oceanographic variables is especially

important in order to correctly represent the ocean-ice-atmosphere interface. We have implemented plots with spatial maps of temperature, salinity and current speeds at original model levels. For temperature and salinity, we have also implemented spatial maps of model biases from the observed climatology with respect to PHC3 climatology. For the model biases, values from the original model levels are linearly interpolated to the climatology (PHC3) levels and then spatially interpolated from the model grid to the regular PHC3 climatology

grid. Resulting fields show model performance in simulating spatial distribution of temperature and salinity.

Vertical transects through arbitrary sections are important for analysis of vertical distribution of ocean water properties. Therefore, diagnostics that allow for the definition of an arbitrary ocean section by providing set of points on the ocean surface are also implemented. For each point, a vertical profile of temperature or salinity on the original model levels is interpolated. All profiles are then connected to form a transect. The great-circle distance between the points is calculated and used as along-track distance. One of the main use cases for transects is to create vertical sections across ocean passages. Transects that follow the pathway of the Atlantic water according to Ilıcak et al. (2016) are also included. Atlantic water is a key water mass of the Arctic Ocean and its proper representation is one of the main challenges in Arctic Ocean modelling. A diagnostic that calculates the temperature of the Atlantic water core for every model as the maximum potential temperature between 200 and 1000-meter depth in the Eurasian Basin is included. The depth of the Atlantic water core is calculated as the model level depth where the maximum temperature is found in Eurasian Basin (Atlantic water core temperature). In order to evaluate the spatial distribution of Atlantic water in different climate models we also provide diagnostics with maps of the spatial distribution of water temperature at the depth of Atlantic water core in *recipe_arctic_ocean.yml*.

### 3.3.4 Sea Ice

Sea ice is a critical component of the climate system, which considerably influences the ocean and atmosphere through different processes and feedbacks (Goosse et al., 2018). In the Arctic, sea ice has been dramatically retreating (Stroeve and Notz, 2018) and thinning (Kwok, 2018) in the past decades (Meredith et al., 2019). In the Antarctic, the sea ice cover has exhibited no significant change over the period of satellite observations, although this is the result of regional compensations and large interannual variability (Meredith et al., 2019). Climate models constitute a useful tool to make projections of the future changes in sea ice (Massonnet et al., 2012). However, the different climate models largely disagree on the magnitude of sea ice changes, even for the same forcing (Stroeve et al., 2012). One reason could be the different treatment of thermodynamic and dynamic processes and feedbacks related to sea ice.

In order to better understand and reduce model errors, two recipes related to sea ice have been implemented into ESMValTool v2.0. The first recipe, *recipe_seaice_feedback.yml*, is related to the negative sea ice growth–thickness feedback (Massonnet et al., 2018b). In this recipe, one process-based diagnostic named the Ice Formation Efficiency (IFE) is computed based on monthly mean sea ice volume estimated north of 80°N. The diagnostic intends to evaluate the strength of the negative sea ice thickness/growth feedback, which causes late-summer negative anomalies in sea ice area and volume to be partially recovered during the next growing season (Notz and Bitz, 2017). To estimate the strength of that feedback, anomalies of the annual minimum of sea ice volume north of 80°N are first estimated. Then, the increase in sea ice volume until the next annual maximum is computed for each year. The IFE is defined as the regression of this ice volume production onto the baseline summer volume anomaly (Figure 31). All CMIP5 models, without exception, simulate negative IFE over the historical period, implying that all these models display a basic mechanism of ice volume recovery when large negative anomalies occur in late summer. However, the strength of the IFE is simulated very differently by the models (Massonnet et al., 2018a). The IFE is closely associated with the annual mean sea ice volume north of 80°N. Also, the strength of the IFE is directly connected to the long-term variability, providing prospects for the application of emergent constraints. However, the shortness of observational records of sea ice thickness and their large uncertainty preclude rigorous applications of such constraints. The analyses nevertheless allow (1) to

pin down that the spread in CMIP5 sea ice volume projections is inherently linked to the way they represent the strength of sea ice feedbacks, and so their mean state, and (2) to provide guidance for the development of future observing systems in the Arctic, by stressing the need for more reliable estimates of sea ice thickness in the central Arctic basin (Ponsoni et al., 2019).

The second recipe, *recipe_sea_ice_drift.yml*, allows to quantify the relationships between Arctic sea ice drift speed, concentration and thickness (Docquier et al., 2017). A decrease in concentration or thickness, as observed in recent decades in the Arctic Ocean (Kwok, 2018; Stroeve and Notz, 2018), leads to reduced sea ice strength and internal stress, and thus larger sea ice drift speed (Rampal et al., 2011). Olason and Notz (2014) investigate the relationships between Arctic sea ice drift speed, concentration and thickness using satellite and buoy

observations. They show that both seasonal and recent long-term changes in sea ice drift are primarily correlated to changes in sea ice concentration and thickness. Our recipe allows quantifying these relationships in climate models. In this recipe, four process-based metrics are computed based on the multi-year monthly mean sea ice drift speed, concentration and thickness, averaged over the Central Arctic. The first metric is the ratio between the modelled drift-concentration slope and the observed drift-concentration slope. The second metric is similar to

the first one, except that sea ice thickness is involved instead of sea ice concentration. The third metric is the normalised distance between the model and observations in the drift-concentration space. The fourth metric is similar to the third one, except that sea ice thickness is involved instead of sea ice concentration. Sea ice concentration from the European Organisation for the Exploitation of Meteorological Satellites Ocean and Sea Ice Satellite Application Facility (Lavergne et al., 2019)), sea ice thickness from the Pan-Arctic Ice-Ocean

Modeling and Assimilation System reanalysis (PIOMAS, Zhang and Rothrock (2003)) and sea ice drift from the International Arctic Buoy Programme (IABP, Tschudi et al. (2016)) are used as reference products to compute these metrics (Figure 32). Results in this example show that the GFDL-ESM2G model can reproduce the sea ice drift speed - concentration/thickness relationships compared to observations, with higher drift speed with lower concentration/thickness, despite the too thin ice in the model, while the MPI-ESM-LR model cannot reproduce

this result.

**3.4 Diagnostics for the evaluation of land processes**

**3.4.1 Land Cover**

Land cover (LC) is either prescribed in the CMIP models or simulated using a Dynamic Global Vegetation Model (DGVM). Within the recent decade, numerous studies focused on the quantification of the impact of land

cover change on climate (see Mahmood et al. (2014) and references therein for a comprehensive review). There is a growing body of evidence that vegetation, especially tree cover, significantly affects the terrestrial water cycle, energy balance (Alkama and Cescatti, 2016; Duveiller et al., 2018b) and carbon cycle (Achard et al., 2014). However, understanding the impact of LC change on climate remains controversial and is still work in progress (Bonan, 2008; Ellison et al., 2012; Mahmood et al., 2014; Sheil and Murdiyarso, 2009). In order to

judge the LC related ESM results, an independent assessment of the accuracy of the simulated spatial distributions of major land cover types is desirable to evaluate the DGVM accuracy for present climate conditions (Lauer et al., 2017).

Recently in the frame of the European Space Agency (ESA) Climate Change Initiative (CCI), a new global LC dataset has been published (Defourny et al., 2014; Defourny et al., 2016) that can be used to evaluate or

prescribe vegetation distributions for climate modelling. Effects of LC uncertainty in the ESA CCI LC dataset on

land surface fluxes and climate are described by Hartley et al. (2017) and Georgievski and Hagemann (2018), respectively. Satellite derived LC classes cannot directly be used for the evaluation of ESM vegetation due to the different concepts of vegetation representation in DGVMs, which are typically based on the concept of plant functional types (PFTs) that are supposed to represent groups of LC with similar functional behaviour. Thus, an important first step is to map the ESA CCI LC classes to PFTs as described by Poulter et al. (2015). As the PFTs in ESMs differ, the current LC diagnostic analyses only major LC types (bare soil, crops, grass, shrubs, trees), which is similar to the approach chosen by Brovkin et al. (2013) and Lauer et al. (2017). The corresponding evaluation metric was implemented into the ESMValTool in *recipe_landcover.yml*. It evaluates areas, mean fractions and biases compared to ESA CCI LC data over the land area of four major regions (global, tropics, northern and southern extra-tropics). Currently the evaluation is using ESA CCI LC data for the epoch 2008-2012 that have been generated with the ESA CCI LC user tool at 0.5 degree resolution. Consequently, model data are interpolated to the same resolution. For the calculation of mean fractions per major region, a land area of these regions needs to be specified and is currently taken from ESA CCI land cover. Example plots of accumulated area and biases in major LC types for different models are shown in Figure 33.

**3.4.2 Albedo changes associated to land cover transitions**

Land Cover Changes (LCC) can modify climate by altering land surface properties such as surface albedo, surface roughness and evaporative fraction. In particular, historical deforestation since the preindustrial era led to an increase in surface albedo corresponding to a global radiative forcing of -0.15 +/- 0.10 $Wm^{-2}$ (Myhre et al., 2013). There are however large uncertainties, even concerning the sign of the effect, regarding the impacts of LCC on near-surface temperature due to persistent model disagreement (Davin et al., 2020; de Noblet-Ducoudré et al., 2012; Lejeune et al., 2017; Pitman et al., 2009). These disagreements arise from uncertainties in 1) the interplay between radiative (albedo) and non-radiative processes (surface roughness and evaporative fraction), 2) the role of local versus large scale processes and feedbacks (Winckler et al., 2017), and 3) the magnitude of change in given surface properties (e.g. albedo). Concerning the latter, Myhre et al. (2005) and Kvalevåg et al. (2010) suggest that the albedo change between natural vegetation and croplands is usually overestimated in climate simulations compared to satellite-derived observational evidence. In addition to this potential bias compared to observational data, there is a substantial spread in the model parametrizations for the albedo response to land-cover perturbations. Boisier et al. (2012) identified that as being responsible for half of the dispersion in the albedo response to LCC since preindustrial times among models participating in the LUCID project, whereas the remaining uncertainty was found to result from differences in the prescribed land cover. A more systematic evaluation of model performance in simulating LUC-induced changes in albedo based on latest available observations is therefore essential in order to reduce these uncertainties.

A satellite-based dataset providing potential effect of a range of land cover transitions on the full surface energy balance (including albedo), at global scale, 1°-resolution, and monthly timescale is now available (Duveiller et al., 2018a). The potential albedo changes associated to vegetation transitions were extracted by a statistical treatment combining the ESA CCI LC data (see 3.4.1 for references) and the mean of the white-sky and black-sky albedo values of the NASA MCD43C3 albedo product for the 2008-2012 period (see Schaaf et al. (2002) for information on the retrieval algorithm). Because land cover-specific albedo values are not a standard output of climate models, in order to retrieve them a diagnostic was developed by Lejeune et al. (2020) which has been implemented into the ESMValTool v2.0 in *recipe_albedolandcover.yml*. This approach determines the

coefficients of multiple linear regressions between the albedo values and the tree, shrub, short vegetation (crops and grasses) and bare soil fractions of each grid cell within spatially moving windows encompassing 5° x 5° model grid cells. These four LC classes correspond to the "IGBPgen" classification in Duveiller et al. (2018a). The recipe provides the option to run the algorithm on an interpolated grid or on the native model grid. The latter option was used in the example provided in Figure 34. Solving these regressions provides the albedo values for trees, shrubs and short vegetation (crops and grasses) from which the albedo changes associated with transitions between these three land cover types are derived. The diagnostic is applied to monthly data, and based on the value of the snow area fraction (snc) distinguishes between snow-free (snc<0.1) and snow-covered (snc>0.9) grid cells for each month. It can calculate albedo estimates for each of these two cases and each of the three land cover types, given that some criteria are fulfilled: the regressions are only conducted in the areas with a minimum number of 15 grid cells (either snow-free or snow-covered), taking only into account the grid cells where the sum of the area fractions occupied by the three considered land cover types exceeds 90%. The algorithm eventually plots global maps of the albedo changes associated with the corresponding LC transitions for each model in their original resolution, next to the satellite-derived estimates from Duveiller et al. (2018a). The diagnostic shows data according to the IGBPgen classification, which entails only four LC classes that can be directly compared to model PFTs. An example plot is shown in Figure 34 for the July albedo change associated with a transition from trees to short vegetation types (crops and grasses). Almost only snow-free areas are visible for this month, while grey areas indicate where the spatial coexistence of the two LC classes was not high enough for the regression technique to be performed, where the regression results did not pass the required quality checks, or grid cells which could not be categorised either as snow-free or as snow-covered (Duveiller et al., 2018a). In the example shown here, July albedo difference between trees and crops or grasses is about at least twice as high in the MPI-ESM-LR model as in the observations, strongly suggesting that the simulated summer albedo increase from historical land cover changes is overestimated in this model. The results reveal that the July albedo difference between trees and crops or grasses is about at least twice as high in the MPI-ESM-LR model as in the observations, strongly suggesting that the simulated summer albedo increase from historical LCC is overestimated in this model.

**3.5 Diagnostics for the evaluation of biogeochemical processes**

**3.5.1 Terrestrial biogeochemistry**

With $CO_2$ being the most important anthropogenic greenhouse gas, it is vital for ESMs to have a realistic representation of the carbon cycle. Atmospheric concentration of $CO_2$ can be inferred from the difference between anthropogenic emissions and the land and ocean carbon sinks simulated by the models. These sinks are affected by atmospheric $CO_2$ and climate change, thus introducing feedbacks between the climate system and the carbon cycle (Arora et al., 2013; Friedlingstein et al., 2006). Quantification of these feedbacks to estimate the evolution of these carbon sinks and thus the atmospheric $CO_2$ concentration and the resulting climate change is paramount (Cox et al., 2013; Friedlingstein et al., 2014; Wenzel et al., 2014; Wenzel et al., 2016). The Anav et al. (2013) paper evaluated CMIP5 models in three different time scales: long-term trends, interannual variability and seasonal cycles for the main climatic variables controlling both the spatial and temporal characteristics of the carbon cycle, i.e. surface land temperature (tas), precipitation over land (pr), sea surface temperature (tos), land-atmosphere (nbp) and ocean-atmosphere fluxes (fgco$_2$), gross primary production (gpp), leaf area index (lai), and carbon content in soil and vegetation (cSoil, cVeg). Models are able to simulate key characteristics of the main

climatic variables and their seasonal evolution, but deficiencies in the simulation of specific variables, especially in the land carbon cycle with a general overestimation of photosynthesis and leaf area index, as well as an underestimation of the primary production in the ocean, exist.

The analysis from the Anav et al. (2013) can be reproduced with *recipe_anav13jclim.yml*. In addition to the diagnostics already implemented in ESMValTool v1.0 and ported to v2.0, new diagnostics for the timeseries anomalies of tas, pr, tos, as well as timeseries for nbp and fgco$_2$ have been added, reproducing Figures 1, 2, 3, 5, and 13 of Anav et al. (2013), with the latter two also forming Figure 26 of Flato et al. (2013). In ESMValTool v2.0, observational estimates of gpp are included from the latest data release of the FLUXCOM project (Jung et al., 2019) which integrates FLUXNET measurements, satellite remote sensing and climate data with machine learning to provide improved global products of land-atmosphere fluxes for evaluation. The routines needed to make carbon and energy fluxes from the FLUXCOM project CMOR-compliant to facilitate process based model evaluation is also made available as part of ESMValTool v2.0. As an example of the newly added plots, Figure 35 shows the timeseries for the land-atmosphere carbon flux nbp, similar to Figure 5 of Anav et al. (2013). Shading indicates the confidence interval of the CMIP5 ensemble standard deviation, derived from assuming a *t*-distribution centered on the ensemble mean (inner curve), while the gray shading shows the overall range of variability of the models. As positive values correspond to a carbon uptake of the land, the plot shows a slight increase in the land carbon uptake over the whole period.

### 3.5.2 Ecosystem Turnover Times of Carbon

The exchange of carbon between the land biosphere and atmosphere represents a key feedback mechanism that will determine the effect of global changes on the carbon cycle and vice-versa (Heimann and Reichstein, 2008). Despite significant implications, the uncertainties in simulated land carbon stocks that integrates the land-atmosphere carbon exchange are large, and, therefore, represent a major challenge for ESMs (Friedlingstein et al., 2014; Friend et al., 2014). One of the major factors leading to these uncertainties is the turnover time of carbon, the time period that a carbon atom on average spends in land ecosystems, from assimilation through photosynthesis to its release back into the atmosphere. This emergent ecosystem property, calculated, for example, as a ratio of long-term average total carbon stock to gross primary productivity, has been extensively used to evaluate ESM simulations (Carvalhais et al., 2014; Koven et al., 2015; Koven et al., 2017; Todd-Brown et al., 2013). Despite the large range of observational uncertainties and sources, ESM simulations consistently exhibit a robust correlation with the observation ensembles, but with a substantial underestimation bias.

Carvalhais et al. (2014) evaluated the biases in ecosystem carbon turnover time in CMIP5 models, their associations with climate variables, and then quantified multimodel biases and agreements. The *recipe_carvalhais2014nat.yml* reproduces the analysis of Carvalhais et al. (2014). It requires the simulations of total vegetation carbon content (cVeg), total soil carbon content (cSoil), gross primary productivity (gpp), as well as precipitation (pr), and near surface air temperature (tas). As an example, an evaluation of the zonal means of turnover time in CMIP5 models is shown in Figure 36. The models follow the gradient of increasing turnover times of carbon from tropics to higher latitudes, much related to temperature decreases, as observed in observations. However, for most of the latitudinal bands, with the exception of one model, most simulations reveal turnover times that are faster than the observations. Most CMIP5 models (and multi-model ensemble) have a much shorter turnover time than the observation-based estimate across the whole latitudinal range. Even though different estimates of observation-based carbon fluxes and stocks can vary significantly, a recent study

by Fan et al. (2020), their Figure 5a, shows that the zonal distributions of observation-based estimates of turnover time is robust against the differences in observations.. The spread among the models is also large and can vary by one order of magnitude. This results not only in a large bias in turnover time, but also a considerable disagreement among the models. In fact, the majority of CMIP5 models simulate turnover time more than four times shorter than the observation-based estimate in most regions (Figure 37). A generalized underestimation of turnover times of carbon is apparently dominant in water limited regions. In most of these regions most models show estimates outside of the observational uncertainties (stippling). These results challenge the combined effects of water and temperature limitations on turnover times of carbon and suggest the need for improvement on the description of the water cycle in terrestrial ecosystems. In arid and semi-arid regions model agreement is also low with 2 or fewer (out of 10) models within the observational uncertainty. In addition, the recipe also produces the full factorial model-model-observation comparison matrix that can be used to evaluate individual models. It further provides a quantitative measure of turnover times across different biomes, as well as its relationship with precipitation and temperature.

### 3.5.3 Marine biogeochemistry

ESMValTool v2.0 now includes a wide set of metrics to assess marine biogeochemistry performances of ESMs, contained in *recipe_ocean_bgc.yml*. This recipe allows a direct comparison of the models against observational data for temperature (thetao), salinity (so), oxygen (o2), nitrate (no3), phosphate (po4) and silicate (si) from World Ocean Atlas 2013 (WOA, Garcia et al. (2013)), $CO_2$ air-sea fluxes (fgco$_2$) estimated by Landschuetzer et al. (2016), chlorophyll-a (chl) fields from ESACCI-OC (Volpe et al., 2019) and primary production expressed as carbon (intpp) produced by Oregon State University using MODIS data (Behrenfeld and Falkowski, 1997).

We first demonstrate the recipe using the nitrate concentration in the HadGEM2-ES model in the r1i1p1 ensemble member of the historical experiment in the years 2001-2005. However, this recipe can be expanded to include any other ESM with a marine biogeochemical component, or any other field with a suitable observational dataset. The analysis produced by the recipe is a point to point comparison of the model against the observational dataset, similar to the method described in (De Mora et al., 2013). Figures 38 and 39 show the results of a comparison the surface dissolved nitrate concentration in the HadGEM2-ES model compared against the World Ocean Atlas nitrate. To produce these two figures, the surface layer is extracted, an average over the time dimension is produced, then the model are observational data are re-gridded to a common grid. Figure 38 includes four panels; the model and observations in the top two panes, then the difference and the quotient in the lower two panes. It highlights that the HadGEM2-ES model is proficient at reproducing the surface nitrate concentration in the Atlantic ocean, and in mid latitudes, but may struggle to reproduce observations at high latitudes. Figure 39 uses the same preprocessed data as Figure 38, with the model data plotted along the x axis and the observational data along the y-axis. A linear regression line of best fit is shown as a black line. A dashed line indicates the 1:1 line. The results of a linear regression are shown in the top left corner of the figure, where $\hat{\beta}_0$ is the intercept, $\beta_1$ is the slope, R is the correlation, P is the P value, and N is the number of data point pairs. As both the fitted slope and the correlation coefficient are near one, the HadGEM2-ES simulation excelled at reproducing the observed values of the surface nitrate concentration. When viewed together, Figures 38 and 39 show the biases between the model and the observations in the surface layer relative to each other, both in terms of their spatially-independent distribution in Figure 38 and their spatially-dependent distribution in Figure 39. Figure 40 shows the global average depth profile of the dissolved nitrate concentration in the HadGEM2-ES

model and against the World Ocean Atlas dataset. The colour scale indicates the annual average, although in this specific case there is little observed inter-annual variability so the annual averages are closely overlaid. Nevertheless, this class of figure can be useful to evaluate biases between model and observations over the entire depth profile of the ocean and can also be used to identify long term changes in the vertical structure of the ocean models. This figure shows that while the model and the observations both have a similar overall depth structure, the model is not able to produce the observed maximum nitrate concentration at approximately 1000 m depth and overestimates the nitrate concentration deeper in the water column. A multiple panel comparison of satellite derived observations for marine primary production against 16 CMIP5 models over the period 1995-2004 is shown in Figure 41. Both observation and model data are regridded to a regular 1° X 1° horizontal grid and differences are then computed. Systematic biases characterize all models mainly in the equatorial Pacific and Antarctic regions, in some cases with opposite sign, and coastal ocean productivity is generally underestimated with major deviations in the equatorial zone.

**3.5.4 Stratospheric temperature and trace species influencing stratospheric ozone chemistry**

The *recipe_eyring06jgr.yml* has been ported in ESMValTool v2.0 from the CCMVal-Diag tool described by Gettelman et al. (2012) to evaluate coupled chemistry-climate model (CCM) based on a set of core processes relevant for stratospheric ozone concentrations, centered around four main categories (radiation, dynamics, transport, and stratospheric chemistry). Each process is associated with one or more model diagnostics, and with relevant observational data sets that can be used for the model evaluation (Eyring et al., 2006; Eyring et al., 2005).

Since most of the chemical reactions determining ozone distribution in the stratosphere depend on temperature, *recipe_eyring06jgr.yml* allows the comparison of modelled stratospheric temperature with observations in terms of climatological mean, variability and trends (Figure 42). High-latitude temperatures in winter and spring are particularly important for correctly modelling polar ozone depletion induced by polar stratospheric clouds. In the middle stratosphere there are large variations between the analyses and most models, with no clear bias direction, whereas the temperature bias in the troposphere between analyses and models is somewhat smaller, but is negative around 200 hPa in most models. The upper stratosphere is only available for a few models, and while for most of the seasons shown the agreement is relatively good, the spread between analyses and models is very large for the Antarctic polar regions in JJA. The *recipe_eyring06jgr.yml* evaluates the main features of the atmospheric transport by examining the distribution of long-lived traces (such as methane or $N_2O$), the vertical propagation of the annual cycle of water vapour ("tape recorder") and the mean age of air. Due to its important role in driving stratospheric ozone depletion, especially in the polar regions, this recipe includes the vertical distribution and temporal evolution of modelled chlorine ($Cl_y$). It also assesses the capability of the models to simulate realistic ozone vertical distributions (Figure 43) and total ozone annual cycle. Ozone is clearly overestimated by most models, compared to the observations, in the Northern high latitudes between 50 hPa and 10 hPa, which becomes also apparent in the climatological zonal mean at 50 hPa. Southern high latitudes are slightly better represented in the models at 50 hPa with a more general spread around the observations, but at lower pressure levels an overestimation of ozone compared to the observations becomes apparent in some models.

**4. Routine evaluation of CMIP6 models**

**4.1 Running the ESMValTool alongside the ESGF**

An important goal for CMIP6 was to establish a system that allows for routine model evaluation alongside the ESGF directly after the model output is published to the CMIP archive (Eyring et al., 2016a; Eyring et al., 2019; Eyring et al., 2016b). With the release of ESMValTool v2.0, this was reached through ae semi-automatic execution of the ESMValTool at DKRZ on CMIP6 data published to the ESGF. This is supported by the following components: 1) a locally hosted CMIP6 replica data pool, 2) an automatic CMIP6 data replication process, embracing ESMValTool data needs as replication priorities, and 3) a query mechanism to inform the ESMValTool on the availability on new data in the data pool. Based on these components both regularly scheduled ESMValTool executions as well as executions triggered by the availability of new data can be realised. At the moment, the automatic regular execution is implemented. The replica pool is hosted as part of the parallel Lustre HPC file system at DKRZ and associated to a dedicated data project which is supervised by a panel deciding on CMIP6 data storage priorities. However, rapid data replication from ESGF to the local replica tool remains an issue that requires further work, see also the discussion in Eyring et al. (2016b).

ESMValTool data needs are managed in a GitHub repository and automatically integrated into the Synda tool (http://prodiguer.github.io/synda/) based CMIP6 replication pipeline at DKRZ. The content of the data pool is regularly indexed thus providing a high performance query mechanism on locally available data. This index is used to automatically update several recipes with all available CMIP6 models. If new model output has been published to the ESGF, an ESMValTool execution is triggered and new plots are created. The results produced by the ESMValTool are automatically copied to a result cache which is used by the result browser (see next section).

**4.2 ESMValTool result browser at DKRZ**

The ESMValTool result browser has been set up at http://cmip-esmvaltool.dkrz.de/. The ESMValTool results are visualized with the Freie University Evaluation system (FREVA). FREVA provides an efficient and comprehensive access to the evaluation results and datasets. The application system is developed as an easy to use low-end application minimizing technical requirements for users and tool developers. Initially this website shows CMIP5 results that are already published. Newly produced results for CMIP6 are initially water-marked and are only made available without water-mark once quality control has taken place and related papers have been written. This strategy has been supported, encouraged, and approved by the WCRP Working Group of Coupled Modelling (WGCM). The result browser includes a search function that allows to sort by ESMValTool recipes, projects, CMIP6 realms, scientific themes, domain, plot type, applied statistics, references, variables, datasets (including models, multi-model mean and median, and observations), and (k) results. Each figure includes a caption, that is displayed alongside with the figure, and the corresponding metadata. These metadata include the ESMValTool configuration used to perform the analysis and draw the plot, software versions, date of production, input data, program output, notes, and results. In order to get a quick overview, a summary of the ESMValTool configuration used to create a given plot is also available. This summary includes the recipe name, variables and models used as well as the name of the diagnostic script run and the exact version of the ESMValTool (corresponds to the release tag on GitHub) used as basic information to reproduce a plot. Full provenance information providing all details on the figure creation such as version of the input files and

preprocessing steps applied is stored in the metadata of the figure file itself and can be retrieved by downloading the figure and reading the Exif header of the image file.

**5. Summary and Outlook**

The Earth System Model Evaluation Tool (ESMValTool) is a community diagnostics and performance metrics tool specifically targeted to facilitate and enhance comprehensive evaluation of Earth System Models (ESMs) participating in the Coupled Model Intercomparison Project (CMIP). Since the first ESMValTool release in 2016 (v1.0, Eyring et al. (2016c)), substantial technical improvements have been made by a continuously growing

developer community and additional diagnostics have been added. The tool is now developed by more than 40 institutions as open source code on a Github repository (https://github.com/ESMValGroup).

This paper is part of a series of publications that describe the release of ESMValTool version 2.0 (v2.0). One of the main structural changes compared to v1.0 is the separation of the tool into *ESMValCore* and a *Diagnostic Part*. *ESMValCore* is an easy-to-install, well documented Python package that provides the core functionalities

to perform common pre-processing operations and writes the output from models and observations to netCDF files (Righi et al., 2020). These preprocessed output files are then read by the *Diagnostic Part* that includes tailored diagnostics and performance metrics for specific scientific applications that are called by *recipes*. These recipes reproduce sets of diagnostics or performance metrics that have demonstrated their importance in ESM evaluation in the peer-reviewed literature.

This paper describes recipes for the evaluation of large-scale diagnostics in ESMValTool v2.0. It focuses on those diagnostics that were not part of the first major release of the tool (Eyring et al., 2016c) and includes (1) integrative measures of model performance, as well as diagnostics for the evaluation of processes in (2) the atmosphere, (3) ocean and cryosphere, (4) land and (5) biogeochemistry. Recipes for extreme events and in support of regional model evaluation are described by Weigel et al. (2020) and recipes for emergent constraints

and model weighting by Lauer et al. (2020).

Compared to ESMValTool v1.0, the integrative measures of model performance have been expanded with additional atmospheric variables as well as new variables from the ocean, sea ice and land (extending Figure 9.7 of Flato et al. (2013)). In addition, the centered pattern correlation that allows the quantification of progress between different ensembles of CMIP models for multiple variables (extending Figure 9.6 of Flato et al. (2013))

and the single model performance index proposed by Reichler and Kim (2008) that allows an overall assessment of model performance have been added. For the purpose of model development it is important to look at many different metrics. AutoAssess that is developed by the UK Met Office therefore includes a mix of top-down metrics evaluating key model output variables and bottom-up process-oriented metrics. AutoAssess includes 11 thematic areas which will all be implemented in ESMValTool, but in v2.0 as a technical demonstration only the

area for the stratosphere was implemented.

For the evaluation of processes in the atmosphere, the recipe to calculate multi-model averages (e.g., for surface temperature and precipitation) now not only includes absolute values, but also the mean root mean square error of the seasonal cycle compared to observations. The time series of the anomalies in annual and global mean surface temperature with the models being subsampled as in the observations from HadCRUT4 is also included.

In addition, a recipe for the evaluation of the precipitation quantile bias has been added. For atmospheric dynamics recipes to evaluate stratosphere-troposphere coupling and atmospheric blocking indices have been

included. A new diagnostic tool for the evaluation of the water, energy and entropy budgets in climate models (TheDiaTo (v1.0), Lembo et al. (2019)) has been newly implemented, while the NCAR Climate Variability Diagnostic Package (Phillips et al., 2014), already available in v1.0, has been updated in ESMValTool v2.0 to its latest version. In addition, several other diagnostics to evaluate modes of variability as well as weather regimes calculated by the MiLES package (Davini, 2018) have been added.

To evaluate the broad behaviour of models for the global ocean, several diagnostics have been newly implemented, including diagnostics to evaluate the volume weighted global average temperature anomaly, the AMOC, the Drake Passage current, the global total flux of $CO_2$ from the atmosphere into the ocean, and the global total integrated primary production from phytoplankton. A recipe to evaluate specifically the Southern ocean following Russell et al. (2018) has been included and for the Arctic ocean vertical ocean distributions (e.g. temperature and salinity) for different Artic ocean basins and a transect that follows the pathway of the Atlantic water can now be calculated. For sea ice, a recipe related to the evaluation of the negative sea ice growth–thickness feedback which includes the Ice Formation Efficiency (IFE) aa a process-based diagnostic (Massonnet et al., 2018b) and a recipe that can quantify the relationships between Arctic sea ice drift speed, concentration and thickness (Docquier et al., 2017) have been added.

For the evaluation of land processes, satellite derived land cover classes cannot directly be used for ESM vegetation evaluation because Dynamic Global Vegetation Models (DGVMs) use different concepts for vegetation representation, typically based on plant functional types. A recipe has therefore been added that maps the ESA CCI land cover classes to plant functional types as described by Poulter et al. (2015). It includes major land cover types (bare soil, crops, grass, shrubs, trees) similar to the evaluation study by Lauer et al. (2017). In addition, a recipe has been added that can be used to evaluate albedo changes associated to land cover transitions using the ESA CCI dataset of Duveiller et al. (2018a).

For the terrestrial biosphere, a recipe that allows the evaluation of the main climatic variables controlling both the spatial and temporal characteristics of the carbon cycle on three different time scales (long-term trends, interannual variability and seasonal cycles) has been added following Anav et al. (2013). These key variables include surface land temperature, precipitation over land, sea surface temperatures, land-atmosphere and ocean-atmosphere fluxes, gross primary production, leaf area index, and carbon content in soil and vegetation. To evaluate the simulated land carbon stocks that integrates the land-atmosphere carbon exchange, a recipe to evaluate biases in ecosystem carbon turnover time, the time period that a carbon atom on average spends in land ecosystems, from assimilation through photosynthesis to its release back into the atmosphere (Carvalhais et al., 2014) has been added. For marine biogeochemistry, v2.0 now includes a recipe that allows a direct comparison of the models against observational data for several variables including temperature, salinity, oxygen, nitrate, phosphate, silicate, $CO_2$ air-sea fluxes, chlorophyll-a and primary production. The point to point comparison of the model against the observational dataset is similar to De Mora et al. (2013). To evaluate stratospheric dynamics and chemistry a recipe based on a set of core processes relevant for stratospheric ozone concentrations, centered around four main categories (radiation, dynamics, transport, and stratospheric chemistry) has been added (Eyring et al., 2006). Overall these recipes together with those already included in v1.0 allow a broad characterization of the models for key variables (such as temperature and precipitation) on the large-scale, but v2.0 also includes several process-oriented diagnostics.

With this release, for the first time in CMIP it is now possible to evaluate the models as soon as the output is published to the Earth System Grid Federation (ESGF) in a quasi-operational manner. To achieve this, the

ESMValTool has been fully integrated into the ESGF structure at the Deutsches Klima Rechenzentrum (DKRZ). The data from the ESGF are first copied to a local replica and the ESMValTool is then automatically executed alongside the ESGF as soon as new output arrives. An ESMValTool result browser has been set up that makes the evaluation results available to the wider community (http://cmip-esmvaltool.dkrz.de/).

Another major advancement of ESMValTool v2.0 is that it provides full provenance and traceability (see Section 5.2. in Righi et al. (2020) for details). Provenance information for example includes technical information such as global attributes of all input netCDF files, preprocessor settings, diagnostic script settings, and software version numbers but also diagnostic script name and recipe authors, funding projects, references for citation purposes, as well as tags for categorizing the result plots into various scientific topics (like chemistry, dynamics, sea ice, etc.) realms (land, atmosphere, ocean, etc.) or statistics applied (RMSE, anomaly, trend, climatology, etc.). This not only facilitates the sorting of the results in the ESMValTool result browser but also qualifies the tool for the use in studies or assessments where provenance and traceability is particularly important. The current approach to provenance and tags (i.e. what is reported) can be adjusted to international provenance standards as they become available.

These recent ESMValTool developments and their coupling to the ESGF results can now be exploited by global and regional ESM developers as well as by the data analysis and user communities, to better understand the large CMIP ensemble and to support data exploitation. In particular with the addition of provenance, the tool can also provide a valuable source to produce figures in national and international assessment reports (such as the IPCC climate assessments) to enhance the quality control, reproducibility and traceability of the figures included.

The ESMValTool development community will further enhance the capabilities of the tool with the goal to take – together with other activities - climate model evaluation to the next level (Eyring et al., 2019). Targeted technical enhancements will for example include the development of quick-look capabilities that allow to monitor the simulations while they are running to help identifying errors in the simulations early on, a further extension to the application to regional models so that a consistent evaluation between global and regional models can be provided, and distributed computing functionalities. In addition, the tool will be expanded with additional process-oriented diagnostics in various projects to further enhance comprehensive evaluation and analysis of the CMIP models.

**6. Code and data availability**

ESMValTool v2.0 is released under the Apache License, VERSION 2.0. The latest release of ESMValTool v2.0 is publicly available on Zenodo at https://doi.org/10.5281/zenodo.3401363. The source code of the ESMValCore package, which is installed as a dependency of the ESMValTool v2.0, is also publicly available on Zenodo at https://doi.org/10.5281/zenodo.3387139. ESMValTool and ESMValCore are developed on the GitHub repositories available at https://github.com/ESMValGroup.

CMIP5 data are available freely and publicly from the Earth System Grid Federation. Observations used in the evaluation are detailed in the various sections of the manuscript and listed in Table 1. They are not distributed with the ESMValTool, that is restricted to the code as open source software.

*Author contribution.* VE coordinated the ESMValTool v2.0 diagnostic effort and led the writing of the paper. LB, AL, MR, and MS coordinated the diagnostic implementation in ESMValTool v2.0. CE and SK helped with

the coupling of ESMValTool v2.0 and CK with the visualization of the results at the ESMValTool result browser. All other co-authors contributed individual diagnostics to this release. All authors contributed to the text.

*Competing interests.* The authors declare that they have no conflict of interest

*Acknowledgements.* We dedicate this paper to our great friend and colleague, Alexander Loew, who lost his life in a tragic traffic accident. Our thoughts are with his family and his department. The diagnostic development of
1090 ESMValTool v2.0 for this paper was supported by different projects with different scientific focus, in particular by (1) European Union's Horizon 2020 Framework Programme for Research and Innovation "Coordinated Research in Earth Systems and Climate: Experiments, kNowledge, Dissemination and Outreach (CRESCENDO)" project under Grant Agreement No. 641816, (2) Copernicus Climate Change Service (C3S) "Metrics and Access to Global Indices for Climate Projections (C3S-MAGIC)" project, (3) European Union's
Horizon 2020 Framework Programme for Research and Innovation "Advanced Prediction in Polar regions and beyond: Modelling, observing system design and LInkages associated with a Changing Arctic climate (APPLICATE)" project under Grant Agreement No. 727862, (4) European Union's Horizon 2020 Framework Programme for Research and Innovation "PRocess-based climate sIMulation: AdVances in high-resolution modelling and European climate Risk Assessment (PRIMAVERA)" project under Grant Agreement No. 641727,
(5) Federal Ministry of Education and Research (BMBF) CMIP6-DICAD project, (6) ESA Climate Change Initiative Climate Model User Group (ESA CCI CMUG), (7) Helmholtz Society project "Advanced Earth System Model Evaluation for CMIP (EVal4CMIP)", (8) project S1 (Diagnosis and Metrics in Climate Models) of the Collaborative Research Centre TRR 181 "Energy Transfer in Atmosphere and Ocean" funded by the Deutsche Forschungsgemeinschaft (DFG, German Research Foundation) Project No 274762653 and (9) the
National Environmental Research Council (NERC) National Capability Science Multi-Centre (NCSMC) funding for the U.K. Earth System Modelling project (Grant NE/N018036/1). In addition, we received technical support on the ESMValTool v2.0 development from the European Union's Horizon 2020 Framework Programme for Research and Innovation "Infrastructure for the European Network for Earth System Modelling (IS-ENES3)" project under Grant Agreement No 824084. We acknowledge the World Climate Research Program's (WCRP's)
Working Group on Coupled Modelling (WGCM), which is responsible for CMIP, and we thank the climate modelling groups listed in Table 2 for producing and making available their model output. We thank Mariano Mertens (DLR, Germany) for his helpful comments on a previous version and Michaela Langer (DLR, Germany) for her help with editing the manuscript. The computational resources of the Deutsches Klima RechenZentrum (DKRZ, Hamburg, Germany) were essential for developing and testing this new version and are
kindly acknowledged.

**Tables**

**Table 1.** Overview of standard recipes implemented in ESMValTool v2.0 along with the section they are described, a brief description, the diagnostic scripts included, as well as the variables and observational datasets used . For further details we refer to the GitHub repository.

| Recipe name | Chap ter | Descriptio n | Diagnostic scripts | Variables | Observational datasets |
|---|---|---|---|---|---|
| **Section 3.1: Integrative Measures of Model Performance** | | | | | |
| *recipe_perfmetrics_CM IP5.yml* | 3.1.2. 1 | Recipe for plotting the performanc e metrics for the CMIP5 datasets, including the standard ECVs as in Flato et al. (2013), and some additional variables (e.g., ozone, sea ice, aerosol) | perfmetrics/main.ncl  perfmetrics/collect.ncl | ta ua va zg tas | ERA-Interim (Tier 3, Dee et al. (2011))  NCEP (Tier 2, Kalnay et al. (1996)) |
| | | | | hus | AIRS (Tier 1, Aumann et al. (2003))  ERA-Interim (Tier 3, Dee et al. (2011)) |
| | | | | ts | ESACCI-SST (Tier 2, Merchant (2014))  HadISST (Tier 2, Rayner et al. (2003)) |
| | | | | pr | GPCP-SG (Tier 1, Adler et al. (2003)) |
| | | | | clt | ESACCI-CLOUD (Tier 2, Stengel et al. (2016))  PATMOS-X (Tier 2, Heidinger et al. (2014)) |
| | | | | rlut rsut lwcre swcre | CERES-EBAF (Tier 2, Loeb et al. (2018)) |
| | | | | od550aer od870aer abs550aer d550lt1aer | ESACCI-AEROSOL (Tier 2, Popp et al. (2016)) |
| | | | | toz | ESACCI-OZONE (Tier 2, Loyola et al. (2009)),  NIWA-BS (Tier 3, Bodeker et al. (2005)) |
| | | | | sm | ESACCI-SOILMOISTURE (Tier 2, Liu et al. (2012b)) |
| | | | | et | LandFlux-EVAL (Tier 3, Mueller et al. (2013)) |
| | | | | fgco2 | JMA-TRANSCOM (Tier 3, Maki et al. (2017))  Landschuetzer2016 (Tier 2, Landschuetzer et al. (2016)) |
| | | | | nbp | JMA-TRANSCOM (Tier 3, Maki et al. (2017)) |

| | | | | lai | LAI3g (Tier 3, Zhu et al. (2013)) |
|---|---|---|---|---|---|
| | | | | gpp | FLUXCOM (Tier 3, Jung et al. (2019)), Jung et al 2019), MTE (Tier 3, Jung et al. (2011)) |
| | | | | Rlus Rlds Rsus rsds | CERES-EBAF (Tier 2, Loeb et al. (2018)) |
| *recipe_smpi.yml* | 3.1.2.3 | Recipe for computing Single Model Performance Index. Follows Reichler and Kim (2008) | perfmetrics/main.ncl perfmetrics/collect.ncl | ta va ua hus tas psl hfds tauu tauv | ERA-Interim (Tier 3, Dee et al. (2011)) |
| | | | | pr | GPCP-SG (Tier 1, Adler et al. (2003)) |
| | | | | tos sic | HadISST (Tier 2, Rayner et al. (2003)) |
| *recipe_autoassess_*.yml* | 3.1.2.4 | Recipe for mix of top-down metrics evaluating key model output variables and bottom-up metrics | autoassess/autoassess_area_base.py autoassess/plot_autoassess_metrics.py autoassess/autoassess_radiation_rms.py | rtnt rsnt swcre lwcre rsns rlns rsut rlut rsutcs | CERES-EBAF (Tier 2, Loeb et al. (2018)) |
| | | | | rlutcs rldscs | J RA-55 (Tier 1, Onogi et al. (2007)) |
| | | | | prw | SSMI-MERIS (Tier 1, Schröder (2012)) |
| | | | | pr | GPCP-SG (Tier 1, Adler et al. (2003)) |
| | | | | rtnt rsnt swcre lwcre rsns rlns rsut rlut rsutcs | CERES-EBAF (Tier 2, Loeb et al. (2018)) CERES-SYN1deg (Tier 3, Wielicki et al. (1996)) |
| | | | | rlutcs rldscs | JRA-55 (Tier 1, ana4mips, ) CERES-SYN1deg (Tier 3, Wielicki et al. (1996)) |
| | | | | prw | SSMI-MERIS Tier 1, obs4mips, ) SSMI (Tier 1, obs4mips, ) |

| | | | | cllmtisccp clltkisccp clmmtisccp clmtkisccp clhmtisccp clhtkisccp | ISCCP (Tier 1, Rossow and Schiffer (1991)) |
|---|---|---|---|---|---|
| | | | | ta ua hus | ERA-Interim (Tier 3, Dee et al. (2011)) |

| **Section 3.2: Detection of systematic biases in the physical climate: atmosphere** | | | | | |
|---|---|---|---|---|---|
| *recipe_flato13ipcc.yml* | 3.1.2 3.2.1 3.3.1 | Reproducing selected figures from IPCC AR5, chap. 9 (Flato et al., 2013) 9.2, 9.4, 9.5, 9.6, 9.8, 9.14. | clouds/clouds_bias.ncl  clouds/clouds_ipcc.ncl  ipcc_ar5/tsline.ncl  ipcc_ar5/ch09_fig09_06.ncl  ipcc_ar5/ch09_fig09_06_collect.ncl  ipcc_ar5/ch09_fig09_14.py | tas | ERA-Interim (Tier 3, Dee et al. (2011))  HadCRUT4 (Tier 2, Morice et al. (2012)) |
| | | | | tos | HadISST (Tier 2, Rayner et al. (2003)) |
| | | | | swcre lwcre netcre rlut | CERES-EBAF (Tier 2, Loeb et al. (2018)) |
| | | | | pr | GPCP-SG (Tier 1, Adler et al. (2003)) |
| *recipe_quantilebias.yml* | 3.2.2 | Recipe for calculation of precipitation quantile bias | quantilebias/quantilebias.R | pr | GPCP-SG (Tier 1, Adler et al. (2003)) |
| *recipe_zmnam.yml* | 3.2.3.1 | Recipe for zonal mean Northern Annular Mode. The diagnostic computes the index and the spatial pattern to assess the simulation of the stratosphere-troposphere coupling in the boreal hemisphere | zmnam/zmnam.py | zg | --- |
| *recipe_miles_block.yml* | 3.2.3.2 | Recipe for computing 1-d and 2-d atmospheric blocking indices and diagnostics | miles/miles_block.R | zg | ERA-Interim (Tier 3, Dee et al. (2011)) |

| recipe_thermodyn_diagtool.yml | 3.2.4 | Recipe for the computation of various aspects associated with the thermodynamics of the climate system, such as energy and water mass budgets, meridional enthalpy transports, the Lorenz Energy Cycle and the material entropy production. | thermodyn_diagtool/thermodyn_diagnostics.py | hfls hfss pr ps prsn rlds rlus rlut rsds rsus rsdt rsut ts hus tas uas vas ta ua va wap | --- |
|---|---|---|---|---|---|
| recipe_CVDP.yml | 3.2.5.1 | Recipe for executing the NCAR CVDP package in the ESMValTool framework. | cvdp/cvdp_wrapper.py | pr | GPCP-SG (Tier 1, Adler et al. (2003)) |
| | | | | psl | ERA-Interim (Tier 3, Dee et al. (2011)) |
| | | | | tas | Berkeley Earth (Tier 1, Rohde and Groom (2013)) |
| | | | | ts | ERSSTv5 (Tier 1, Huang et al. 2017) |
| recipe_modes_of_variability.yml | 3.2.5.2 | Recipe to compute the RMSE between the observed and modelled patterns of variability obtained through classification and their relative bias (percentage) in the frequency of occurrence and the persistence of each mode. | magic_bsc/weather_regime.r | zg | --- |

| | | | | | |
|---|---|---|---|---|---|
| *recipe_miles_regimes.yml* | 3.2.5.2 | Recipe for computing Euro-Atlantic weather regimes based on k-means clustering | miles/miles_regimes.R | zg | ERA-Interim (Tier 3, Dee et al. (2011)) |
| *recipe_miles_eof.yml* | 3.2.5.3 | Recipe for computing the Northern Hemisphere EOFs | miles/miles_eof.R | zg | ERA-Interim (Tier 3, Dee et al. (2011)) |
| *recipe_combined_indices.yml* | 3.2.5.4 | Recipe for computing seasonal means or running averages, combining indices from multiple models and computing area averages | magic_bsc/combined_indices.r | psl | --- |
| **Section 3.3: Detection of systematic biases in the physical climate: ocean and cryosphere** | | | | | |
| *recipe_ocean_scalar_fields.yml* | 3.3.1 | Recipe to reproduce time series figures of scalar quantities in the ocean. | ocean/diagnostic_timeseries.py | gtintpp gtfgco2 amoc mfo thetaoga soga zostoga | --- |
| *recipe_ocean_amoc.yml* | 3.3.1 | Recipe to reproduce time series figures of the AMOC, the Drake passage current and the stream function | ocean/diagnostic_timeseries.py ocean/diagnostic_transects.py | amoc mfo msftmyz | --- |
| *recipe_russell18jgr.yml* | 3.3.2 | Recipe to reproduce figure from Russell et al. (2018) | russell18jgr/russell18jgr-polar.ncl russell18jgr/russell18jgr-fig*.ncl | tauu tauuo thetao so uo vo sic pH fgco2 | --- |

| | | | | | |
|---|---|---|---|---|---|
| *recipe_arctic_ocean.yml* | 3.3.3 | Recipe for evaluation of ocean components of climate models in the Arctic Ocean | arctic_ocean/arctic_ocean.py | thetao(K) so (0.001) | PHC (Tier 2, Steele et al. (2001)) |
| *recipe_seaice_feedback.yml* | 3.3.4 | Recipe to evaluate the negative ice growth-thickness feedback | seaice_feedback/negative_seaice_feedback.py | sithick | ICESat (Tier2, Kwok et al. (2009)) |
| *recipe_sea_ice_drift.yml* | 3.3.4 | Recipe for sea ice drift - strength evaluation | seaice_drift/seaice_drift.py | siconc | OSI-450-nh (Tier 2, Lavergne et al. (2019)) |
| | | | | sivol | PIOMAS (Tier 2, Zhang and Rothrock (2003)) |
| | | | | sispeed | IABP (Tier 2, Tschudi et al. (2016)) |
| *recipe_SeaIce.yml* | 3.3.4 | Recipe for plotting sea ice diagnostics at the Arctic and Antarctic | seaice/SeaIce_ancyc.ncl<br><br>seaice/SeaIce_tsline.ncl<br><br>seaice/SeaIce_polcon.ncl<br><br>seaice/SeaIce_polcon_diff.ncl | sic | HadISST (Tier 2, Rayner et al. (2003)) |

**Section 3.4: Detection of systematic biases in the physical climate: land**

| | | | | | |
|---|---|---|---|---|---|
| *recipe_landcover.yml* | 3.4.1 | Recipe for plotting the accumulated area, average fraction and bias of landcover classes in comparison to ESA_CCI_LC data for the full globe and large scale regions. | landcover/landcover.py | baresoilFrac<br>grassFrac<br>treeFrac<br>shrubFrac<br>cropFrac | ESACCI-LANDCOVER (Tier 2, Defourny et al. (2016)) |
| *recipe_albedolandcover.yml* | 3.4.2 | Recipe for evaluate land cover-specific albedo values. | landcover/albedolandcover.py | alb | Duveiller 2018 (Tier 2, (Duveiller et al., 2018a) |

**Section 3.5: Detection of biogeochemical biases**

| | | | | | |
|---|---|---|---|---|---|
| *recipe_anav13jclim.yml* | 3.5.1 | Recipe to reproduce most of the figures of Anav et al. | carbon_cycle/mvi.ncl<br><br>carbon_cycle/main.ncl<br><br>carbon_cycle/two_variables.ncl | tas<br>pr | CRU (Tier 3, Harris et al. (2014)) |
| | | | | lai | LAI3g (Tier 3, Zhu et al. (2013)) |

| | | | | | |
|---|---|---|---|---|---|
| | | (2013) | perfmetrics/main.ncl<br><br>perfmetrics/collect.ncl | fgco2<br>nbp | JMA-TRANSCOM (Tier 2, Maki et al. (2017))<br><br>GCP (Tier 2, Le Quere et al. (2018)) |
| | | | | tos | HadISST (Tier 2, Rayner et al. (2003)) |
| | | | | gpp | MTE (Tier 2, (Jung et al., 2011)) |
| | | | | cSoil | HWSD (Tier 2, Wieder (2014)) |
| | | | | cVeg | NDP (Tier 2, Gibbs (2006)) |
| *recipe_carvalhais2014 nat.yml* | 3.5.2 | Recipe to evaluated the biases in ecosystem carbon turnover time. | regrid_areaweighted.py<br><br>compare_tau_modelVobs_matri x.py<br><br>compare_tau_modelVobs_climat ebins.py<br><br>compare_zonal_tau.py<br><br>compare_zonal_correlations_tau Vclimate.py | tau (non-CMOR variable, that is derived as the ratio of total ecosystem carbon stock and gross primary productivit y) | Carvalhais et al. (2014) |
| *recipe_ocean_bgc.yml* | 3.5.3 | Recipe to evaluate the marine biogeoche mistry models of CMIP5. There are also some physical evaluation metrics. | ocean/diagnostic_timeseries.py<br>ocean/diagnostic_profiles.py<br>ocean/diagnostic_maps.py<br>ocean/diagnostic_model_vs_obs. py<br>ocean/diagnostic_transects.py<br>ocean/diagnostic_maps_multimo del.py | thetao<br>so<br>no3<br>o2<br>si | WOA (Tier 2, (Locarnini, 2013))<br><br><br>WOA (Tier 2, Garcia et al. (2013)) |
| | | | | intpp | Eppley-VGPM-MODIS (Tier 2, Behrenfeld and Falkowski, 1997) |
| | | | | chl | ESACCI-OC (Tier 2, Volpe et al. (2019)Volpe et al., 2019) |
| | | | | fgco2 | Landschuetzer2016 (Tier 2, Landschuetzer et al. (2016)) |
| | | | | dfe<br>talk<br>mfo | |
| *recipe_eyring06jgr.yml* | 3.5.4 | Recipe to reproduce stratospheri c dynamics and chemistry figures from Eyring et al. (2006) | eyring06jgr/eyring06jgr_fig*.ncl | ta<br>ua | ERA-Interim (Tier 3, Dee et al. (2011)) |
| | | | | vmro3<br>vmrh2o | HALOE (Tier 2, Russell et al. (1993), Grooß and Russell Iii (2005)) |
| | | | | toz | NIWA-BS (Tier 3, Bodeker et al. (2005)) |

**Table 2.** Overview of CMIP5 models used in the figures shown in this paper alongside with a reference.

| | Modelling Center | Model | Reference |
|---|---|---|---|
| 1 | Centre for Australian Weather and Climate Research, Australia | **ACCESS1-0** | Dix et al. (2013) |
| | | **ACCESS1-3** | Dix et al. (2013) |
| 2 | Beijing Climate Center, China Meteorological Administration, China | **BCC-CSM1.1** | Wu (2012) |
| | | **BCC-CSM1.1-M[#]** | Wu (2012) |
| 3 | College of Global Change and Earth System Science, Beijing Normal University, China | **BNU-ESM** | |
| 4 | Canadian Centre for Climate Modelling and Analysis, Canada | **CanAM4** | von Salzen et al. (2013) |
| | | **CanCM4** | von Salzen et al. (2013) |
| | | **CanESM2** | Arora et al. (2011) |
| 5 | National Centre for Atmospheric Research, USA | **CCSM4** | Gent et al. (2011); Meehl et al. (2012) |
| | Community Earth System Model Contributors | **CESM1(BGC)** | Gent et al. (2011); Meehl et al. (2012) |
| | | **CESM1(CAM5)** | Gent et al. (2011); Meehl et al. (2012) |
| | | **CESM1(FASTCHEM)** | Gent et al. (2011); Meehl et al. (2012) |
| | | **CESM1(WACCM)** | Calvo et al. (2012); Gent et al. (2011); Marsh et al. (2013) |
| 6 | Centro Euro-Mediterraneo per I Cambiamenti Climatici, Italy | **CMCC-CM** | Fogli et al. (2009) |
| | | **CMCC-CMS** | Fogli et al. (2009) |
| 7 | Centre National de Recherches Meteorologiques, France | **CNRM-CM5** | Voldoire et al. (2012) |
| | | **CNRM-CM5-2** | Voldoire et al. (2012) |
| 8 | Commonwealth Scientific and Industrial Research Organization in collaboration with Queensland Climate Change Centre of Excellence, Australia | **CSIRO-Mk3-6-0** | Rotstayn et al. (2012) |
| 9 | EC-EARTH consortium, Europe | **EC-EARTH[#]** | Hazeleger et al. (2012) |
| 10 | LASG, Institute of Atmospheric Physics, Chinese Academy of Sciences and CESS,Tsinghua University, China | **FGOALS-g2** | Li et al. (2013) |
| 11 | LASG, Institute of Atmospheric Physics, Chinese Academy of Sciences, China | **FGOALS-s2** | Bao et al. (2013) |
| 12 | The First Institute of Oceanography, SOA, China | **FIO-ESM** | Zhou et al. (2014) |
| 13 | NOAA Geophysical Fluid Dynamics Laboratory, USA | **GFDL-CM2p1** | Qiao et al. (2004); Song et al. (2012) |
| | | **GFDL-CM3** | Donner et al. (2011) |
| | | **GFDL-ESM2G** | Dunne et al. (2012) |
| | | **GFDL-ESM2M** | Dunne et al. (2012) |
| 14 | NASA Goddard Institute for Space Studies, USA | **GISS-E2-H** | Schmidt et al. (2006) |
| | | **GISS-E2-R** | Schmidt et al. (2006) |
| 15 | Met Office Hadley Centre, UK | **HadCM3** | Gordon et al. (2000) |
| | | **HadGEM2-CC** | Martin et al. (2011) |
| | | **HadGEM2-ES** | Collins et al. (2011) |
| 16 | National Institute of Meteorological Research, Korea Meteorological Administration, Korea | **HadGEM2-AO[#]** | Martin et al. (2011) |
| 17 | Russian Institute for Numerical Mathematics, Russia | **INM-CM4** | Volodin et al. (2010) |
| 18 | Institut Pierre Simon Laplace, France | **IPSL-CM5A-LR** | Dufresne et al. (2013) |
| | | **IPSL-CM5A-MR** | Dufresne et al. (2013) |
| | | **IPSL-CM5B-LR** | Dufresne et al. |

| | | | (2013) |
|---|---|---|---|
| 19 | Japan Agency for Marine-Earth Science and Technology, Atmosphere and Ocean Research Institute (The University of Tokyo), and National Institute for Environmental Studies, Japan | **MIROC-ESM** | Watanabe et al. (2011) |
| | | **MIROC-ESM-CHEM** | Watanabe et al. (2011) |
| | | **MIROC4h** | Sakamoto et al. (2012) |
| | | **MIROC5** | Watanabe et al. (2010) |
| 20 | Max Planck Institute for Meteorology, Germany | **MPI-ESM-LR** | Giorgetta et al. (2013) |
| | | **MPI-ESM-MR** | Giorgetta et al. (2013) |
| | | **MPI-ESM-P** | Giorgetta et al. (2013) |
| 21 | Meteorological Research Institute, Japan | **MRI-CGCM3** | Yukimoto et al. (2012) |
| 22 | Norwegian Climate Centre, Norway | **NorESM1-M** | Bentsen et al. (2013); Iversen et al. (2012) |
| | | **NorESM1-ME** | Bentsen et al. (2013); Iversen et al. (2012) |

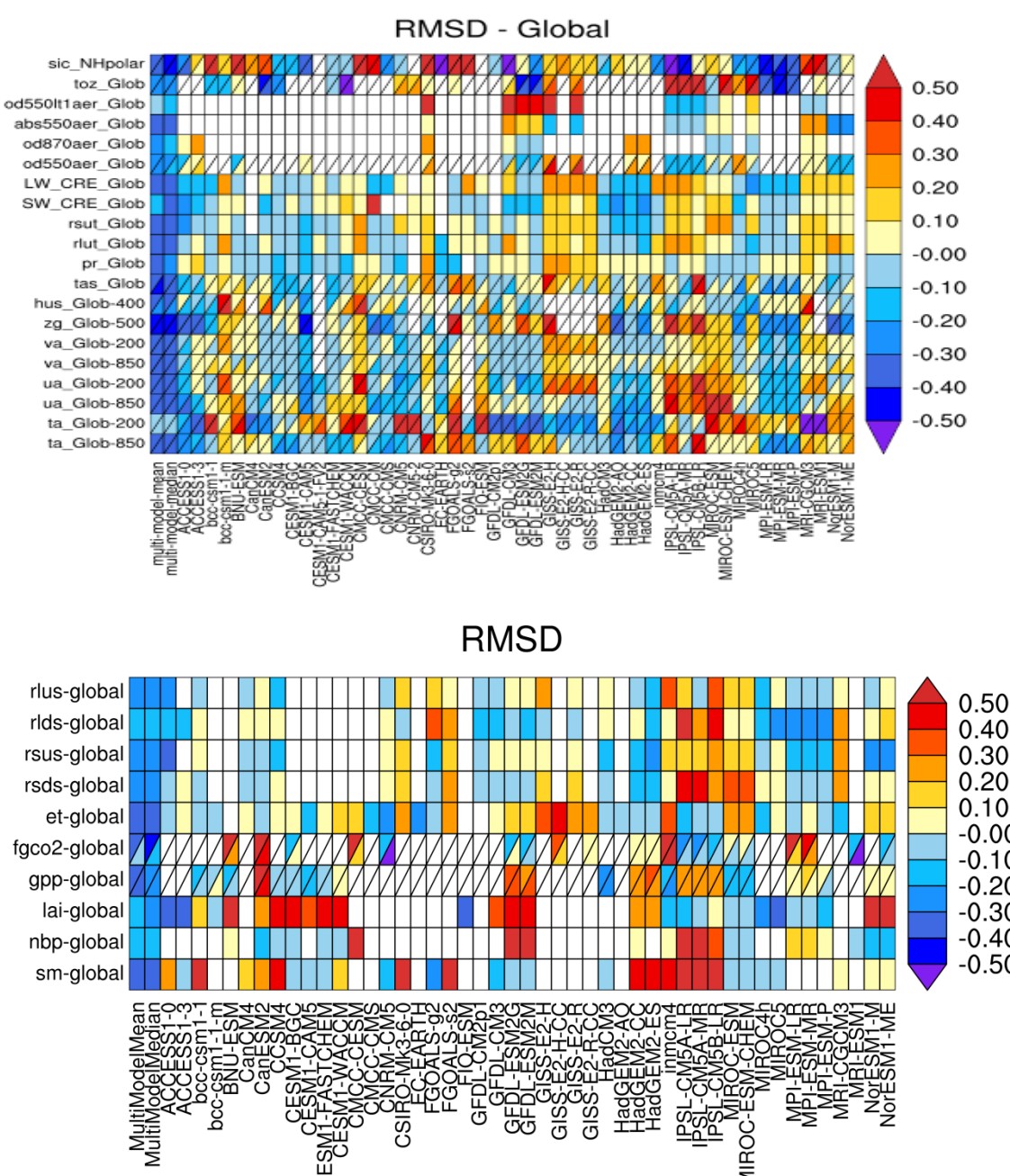

**Figure 1. Relative space-time root-mean-square deviation (RMSD) calculated from the climatological seasonal cycle of the CMIP5 simulations. The years averaged depend on the years with observational data available. A relative performance is displayed, with blue shading indicating better and red shading indicating worse performance than the median of all model results. Note that the colors would change if models are added or removed. A diagonal split of a grid square shows the relative error with respect to the reference data set (lower right triangle) and the alternative data set (upper left triangle). White boxes are used when data are not available for a given model and variable. The performance metrics are shown separately for atmosphere, ocean and sea ice (upper panel), and land (lower panel). Extended from Figure 9.7 of IPCC WG I AR5 Chapter 9 (Flato et al., 2013) and produced with** *recipe_perfmetrics_CMIP5.yml.***, see details in Section 3.1.1.**

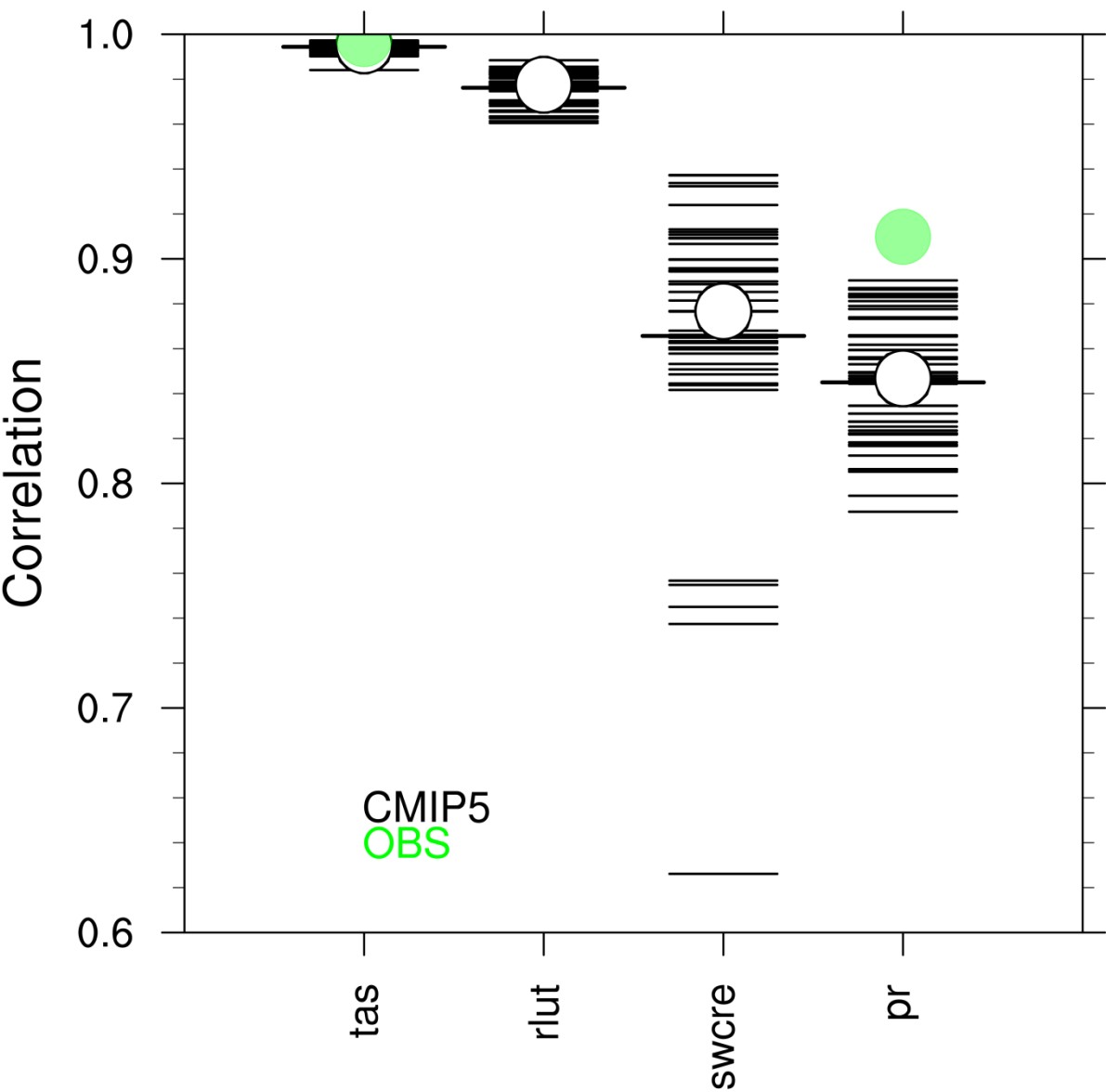

**Figure 2. Centred pattern correlations for the annual mean climatology over the period 1980-1999 between models and observations. Results for individual CMIP5 models are shown (thin dashes), as well as the ensemble average (longer thick dash) and median (open circle). The correlations are computed between the models and the reference dataset. When an alternate observational dataset is present, its correlation to the reference dataset is also shown (solid green circles). Similar to Figure 9.6 of IPCC WG I AR5 Chapter 9 (Flato et al., 2013) and produced with** *recipe_flato13ipcc.yml*, **see details in Section 3.1.2.**



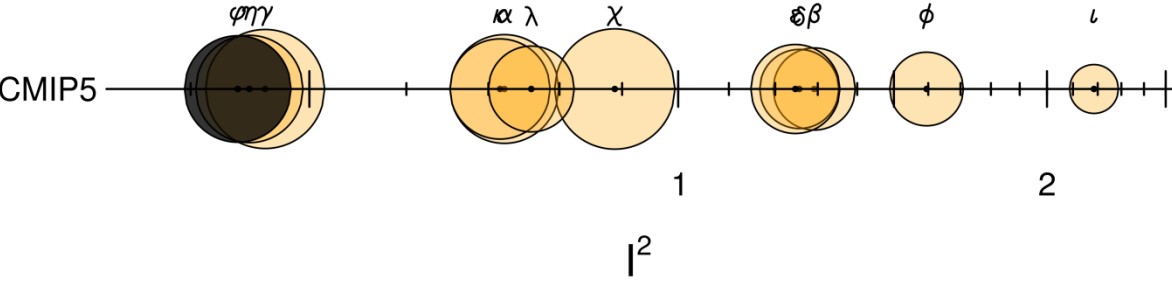

α: CNRM-CM5
β: CSIRO-Mk3-6-0
χ: GFDL-ESM2G
δ: MIROC-ESM
ε: MIROC-ESM-CHEM
φ: MIROC5
γ: MPI-ESM-LR

η: MPI-ESM-MR
ι: MRI-CGCM3
φ: MultiModelMean
κ: NorESM1-M
λ: NorESM1-ME

**Figure 3. Single Model Performance Index I2 for individual models (orange circles). The size of each circle represents the 95% confidence interval of the bootstrap ensemble. The black circle indicates the I2 of the CMIP5 multi-model mean. The I2 values vary around one, with underperforming models having a value greater than one, while values below one represent more accurate models. Similar to Reichler and Kim (2008) Figure 1 and produced with *recipe_smpi.yml*, see details in Section 3.1.3.**


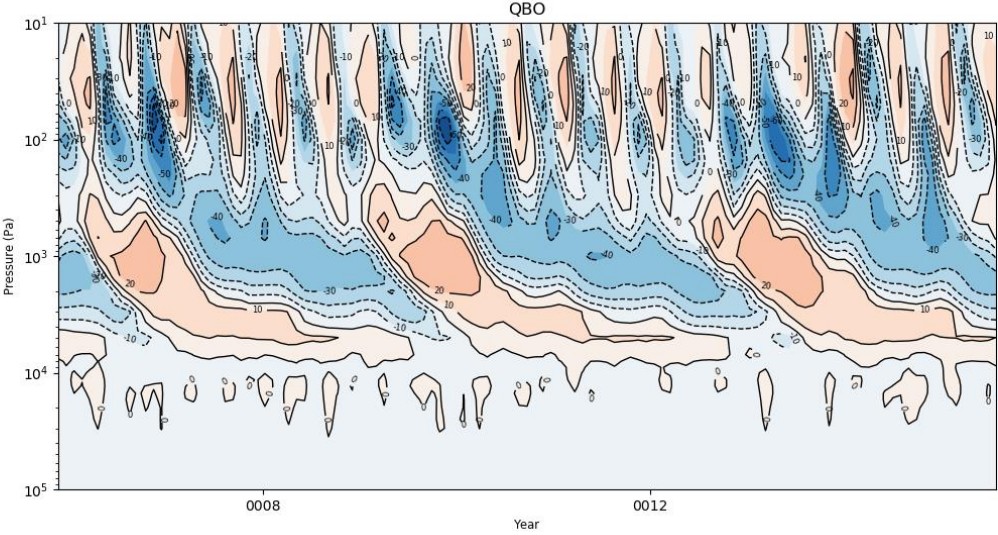

**Figure 4. AutoAssess diagnostic for the Quasi-Biennial Oscillation (QBO) showing the time-height plot of zonal mean zonal wind averaged between 5°S and 5°N for UKESM1-0-LL over the period 1995-2014 in m/s. Produced with recipe_*autoassess_\*.yml*., see details in Section 3.1.4.**

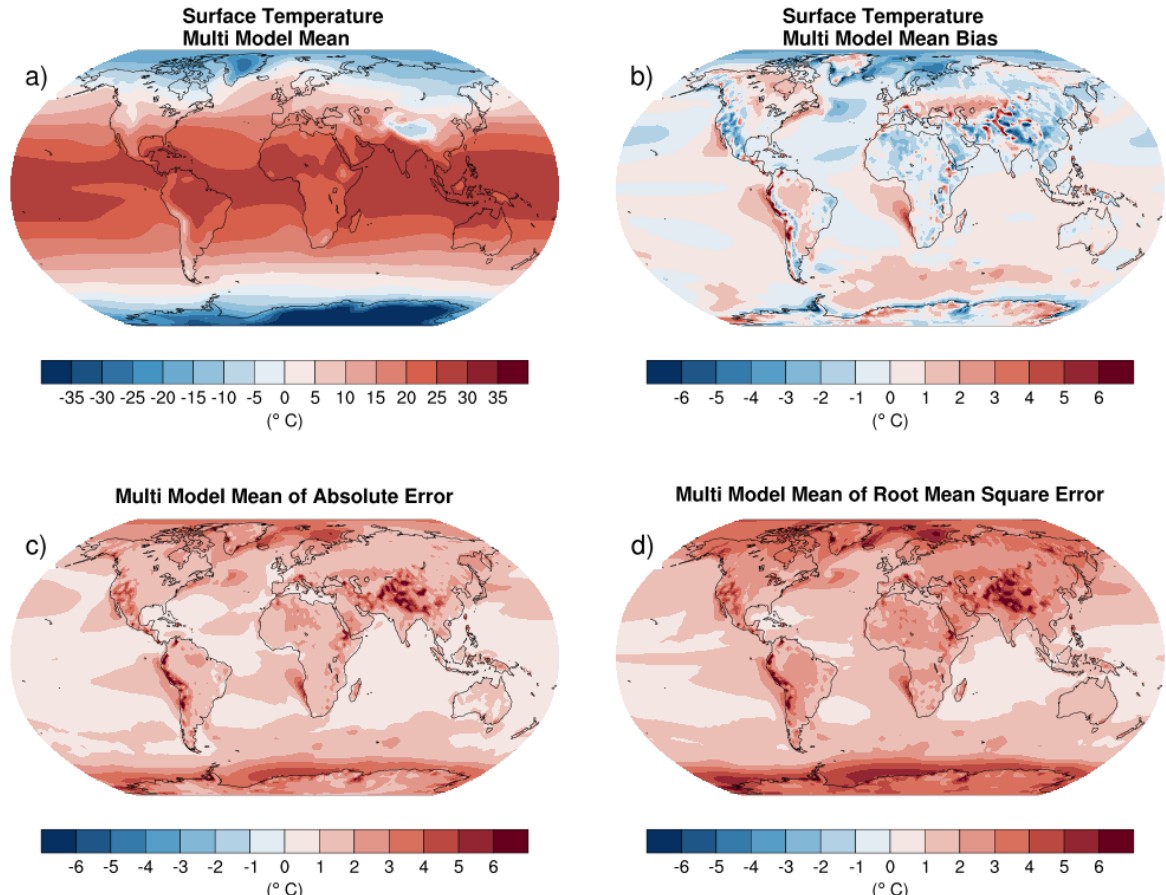

**Figure 5. Annual-mean surface (2 m) air temperature (°C) for the period 1980-2005. (a) Multi-model (ensemble) mean constructed with one realization of all available models used in the CMIP5 historical experiment. (b) Multi-model mean bias as the difference between the CMIP5 multi-model mean and the climatology from ECMWF reanalysis of the global atmosphere and surface conditions (ERA)-Interim (Dee et al., 2011). (c) Mean absolute model error with respect to the climatology from ERA-Interim. (d) Mean root mean square error of the seasonal cycle with respect to the ERA-Interim. Updated from Fig. 9.2 of IPCC WG I AR5 Chapter 9 (Flato et al., 2013) and produced with *recipe_flato13ipcc.yml*, see details in Section 3.2.1.**

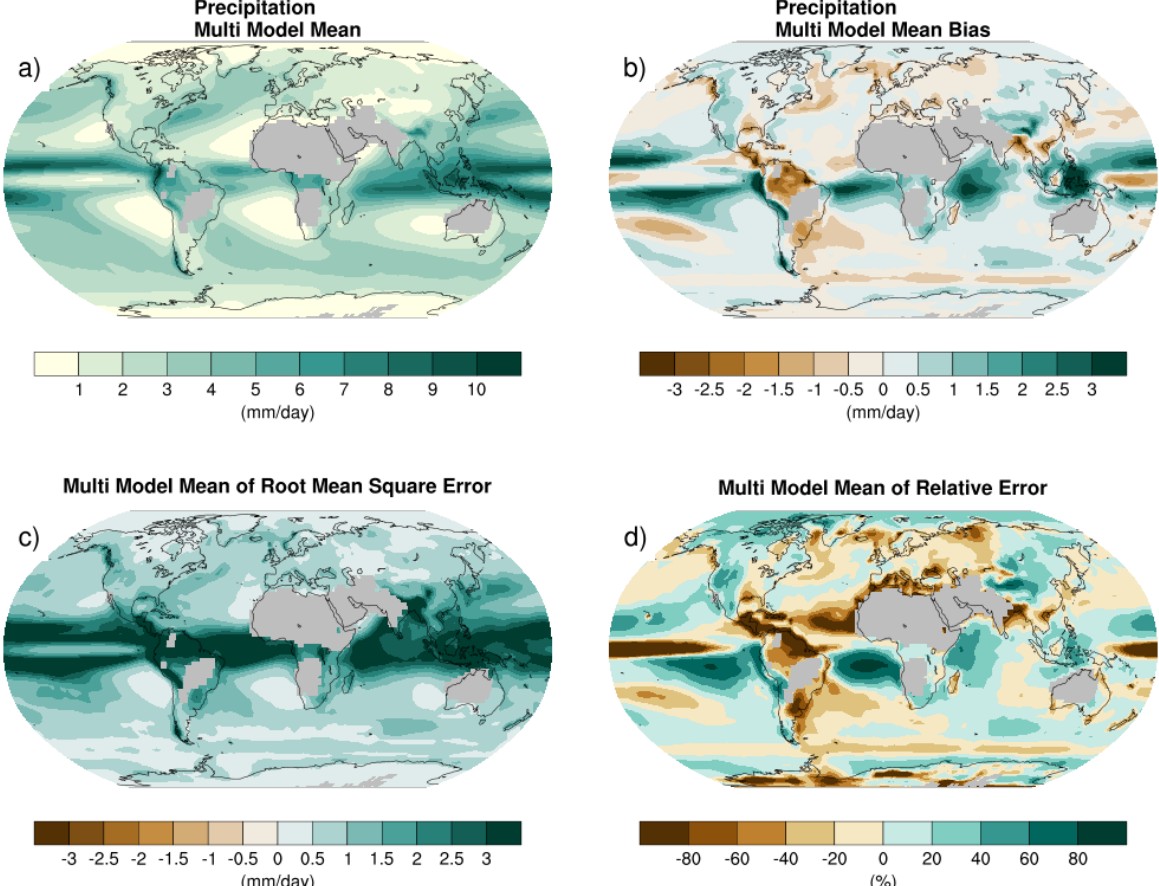

**Figure 6. Annual-mean precipitation rate (mm day$^{-1}$) for the period 1980-2005. (a) Multi-model (ensemble) mean constructed with one realization of all available models used in the CMIP5 historical experiment. (b) Multi-model mean bias as the difference between the CMIP5 multi-model mean and the analyses from the Global Precipitation Climatology Project (Adler et al., 2003). (c) Mean root mean square error of the seasonal cycle with respect toobservations. (d) Mean relative model error with respect to observations. Updated from Fig. 9.4 of IPCC WG I AR5 Chapter 9 (Flato et al., 2013) and produced with *recipe_flato13ipcc.yml*, see details in Section 3.2.1.**


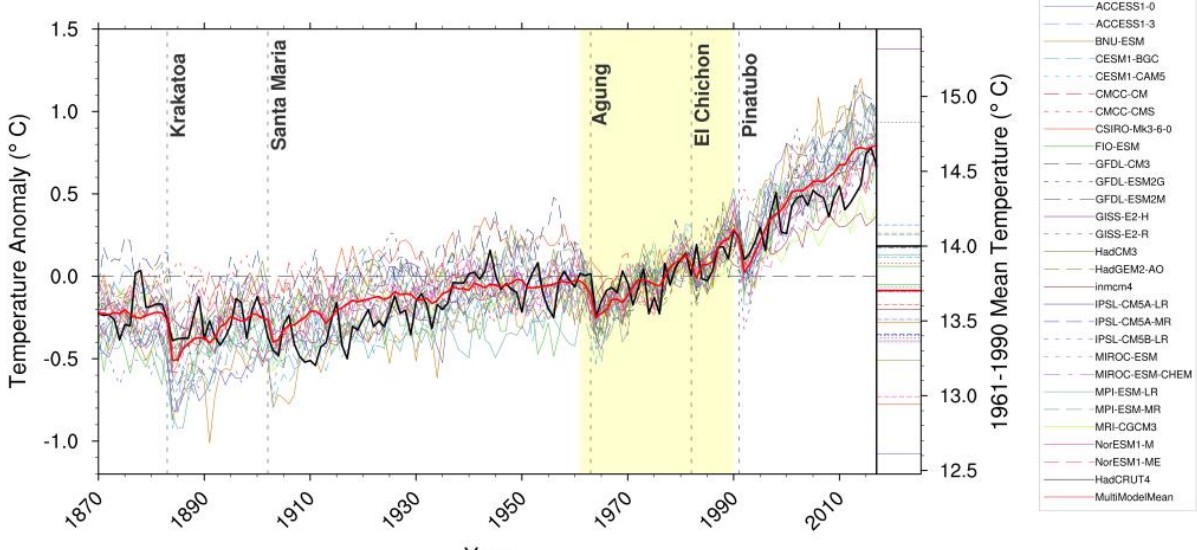


**Figure 7. Anomalies in annual and global mean surface temperature of CMIP5 models and HadCRUT4 observations. Yellow shading indicates the reference period (1961 -1990); vertical dashed grey lines represent times of major volcanic eruptions. The right bar shows the global mean surface temperature of the reference period. CMIP5 model data are subsampled by the HadCRUT4 observational data mask and processed like described in Jones et al. (2013).**
**Jones et al. (2013) All simulations are historical experiments up to and including 2005 and the RCP 4.5 scenario after 2005. Extended from Figure 9.8 of IPCC WG I AR5 Chapter 9 (Flato et al., 2013) and produced with** *recipe_flato13ipcc.yml,* **see details in Section 3.2.1.**

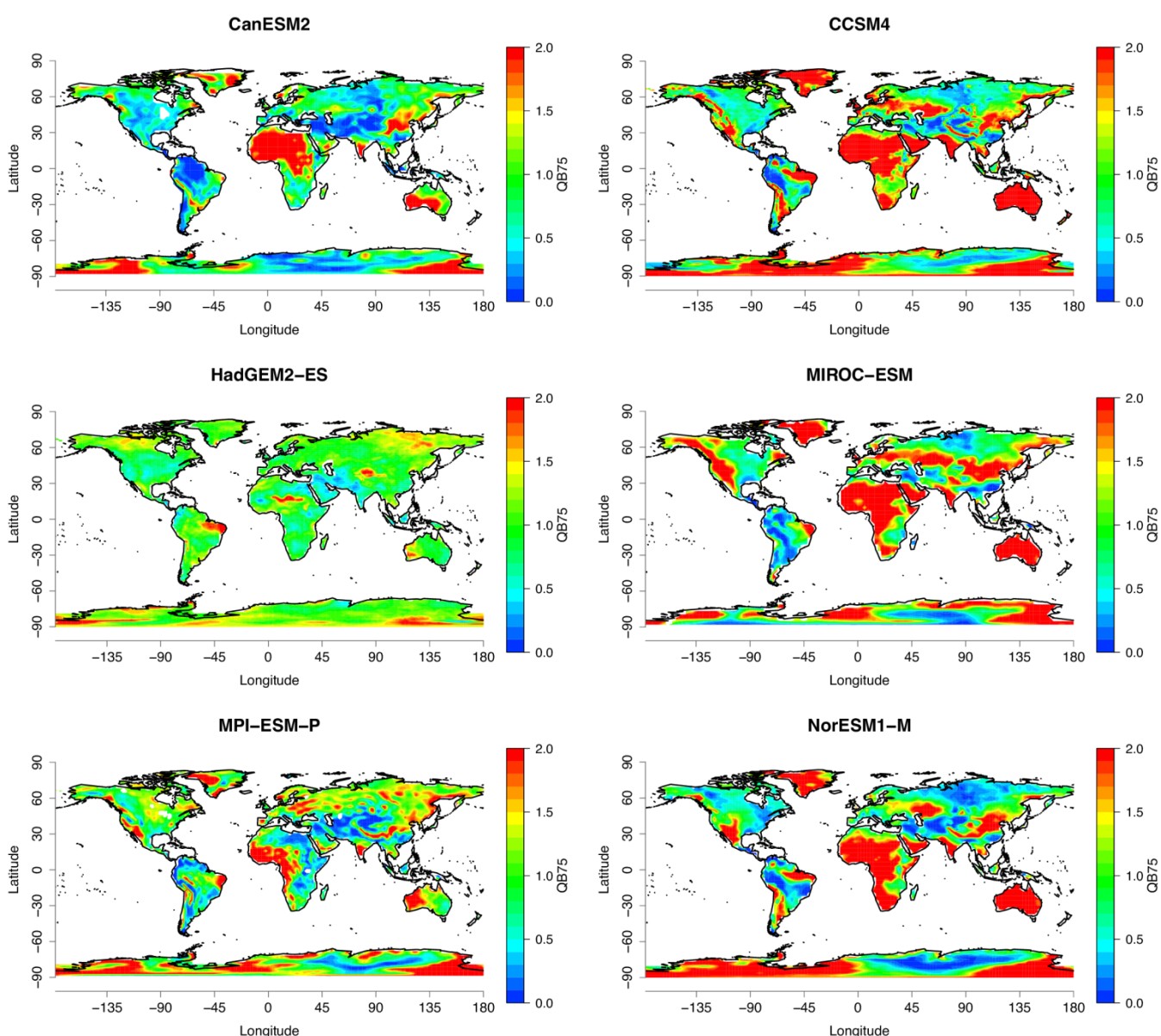

Figure 8. Precipitation quantile bias (75% level, unitless) evaluated for an example subset of CMIP5 models over the period 1979 to 2005 using GPCP-SG v 2.3 gridded precipitation as a reference dataset. Similar to Mehran et al. (2014) and produced with *recipe_quantilebias.yml*. See details in Section 3.2.2.

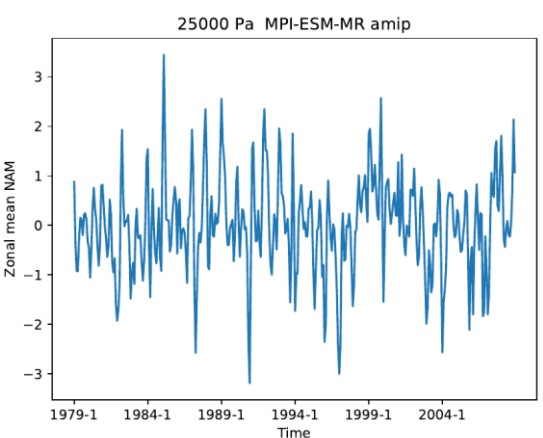
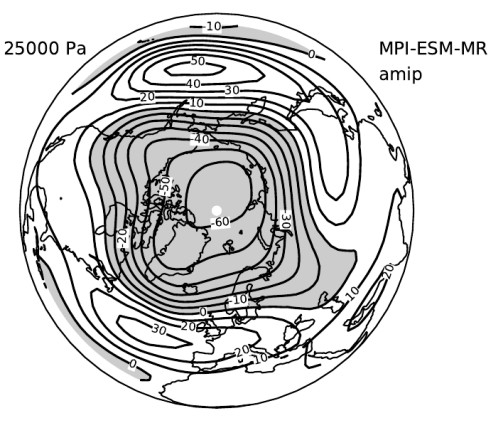

**Figure 9. The standardized zonal mean NAM index (left, unitless) at 250 hPa for the atmosphere-only CMIP5 simulation of the Max Planck Institute for Meteorology (MPI-ESM-MR) model, and the regression map of the monthly geopotential height on this zonal-mean NAM index (right, in meters). Note the variability on different temporal scales of the index, from monthly to decadal. Similar to Figure 2 of Baldwin and Thompson (2009) and produced with *recipe_zmnam.yml*, see details in Section 3.2.3.1.**


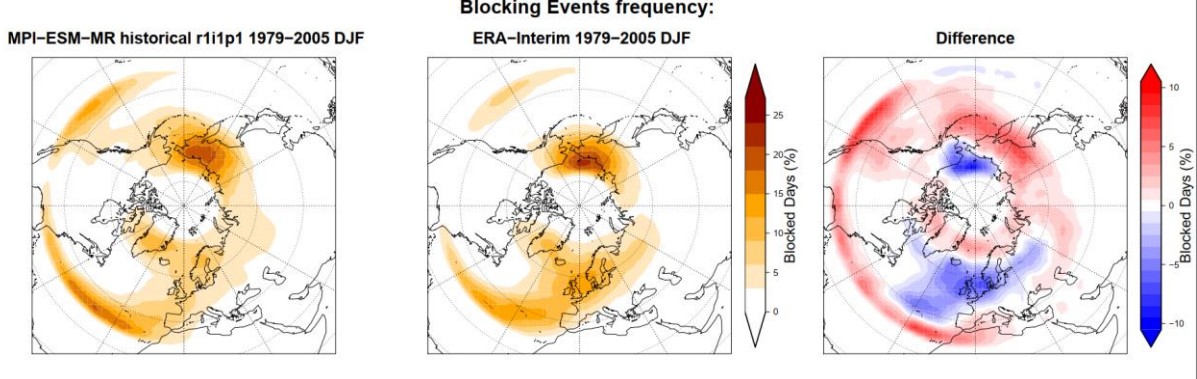

**Figure 10. 2-d Blocking Events frequency (percentage of blocked days) following the Davini et al. (2012) index over the 1979-2005 DJF period for (left) CMIP5 MPI-ESM-MR historical r1i1p1 run (center) ERA-Interim Reanalysis and (right) their differences. Produced with *recipe_miles_block.yml*, see details in Section 3.2.3.2.**


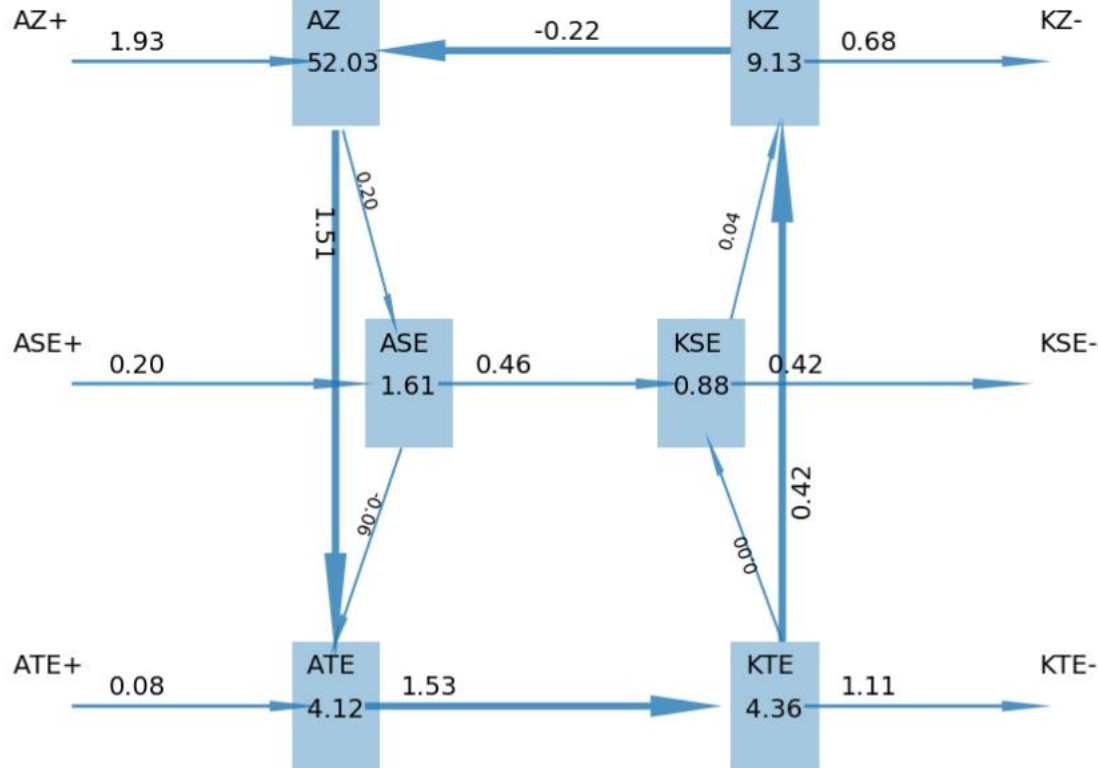

**Figure 11. A Lorenz Energy Cycle flux diagram for one year of a CMIP5 model pre-industrial control run (cfr, Ulbrich and Speth (1991)). "A" stands for available potential energy (APE), "K" for kinetic energy (KE), "Z" for zonal 1115 mean, "S" for stationary eddies, "T" for transient eddies. "+" indicates source of energy, "-" a sink. For the energy reservoirs, the unit of measure is $Jm^{-2}$, for the energy conversion terms, the unit of measure is $Wm^{-2}$. Similar to Figure 5 of Lembo et al. (2019) and produced with *recipe_thermodyn_diagtool.yml*, see details in Section 3.2.4.**


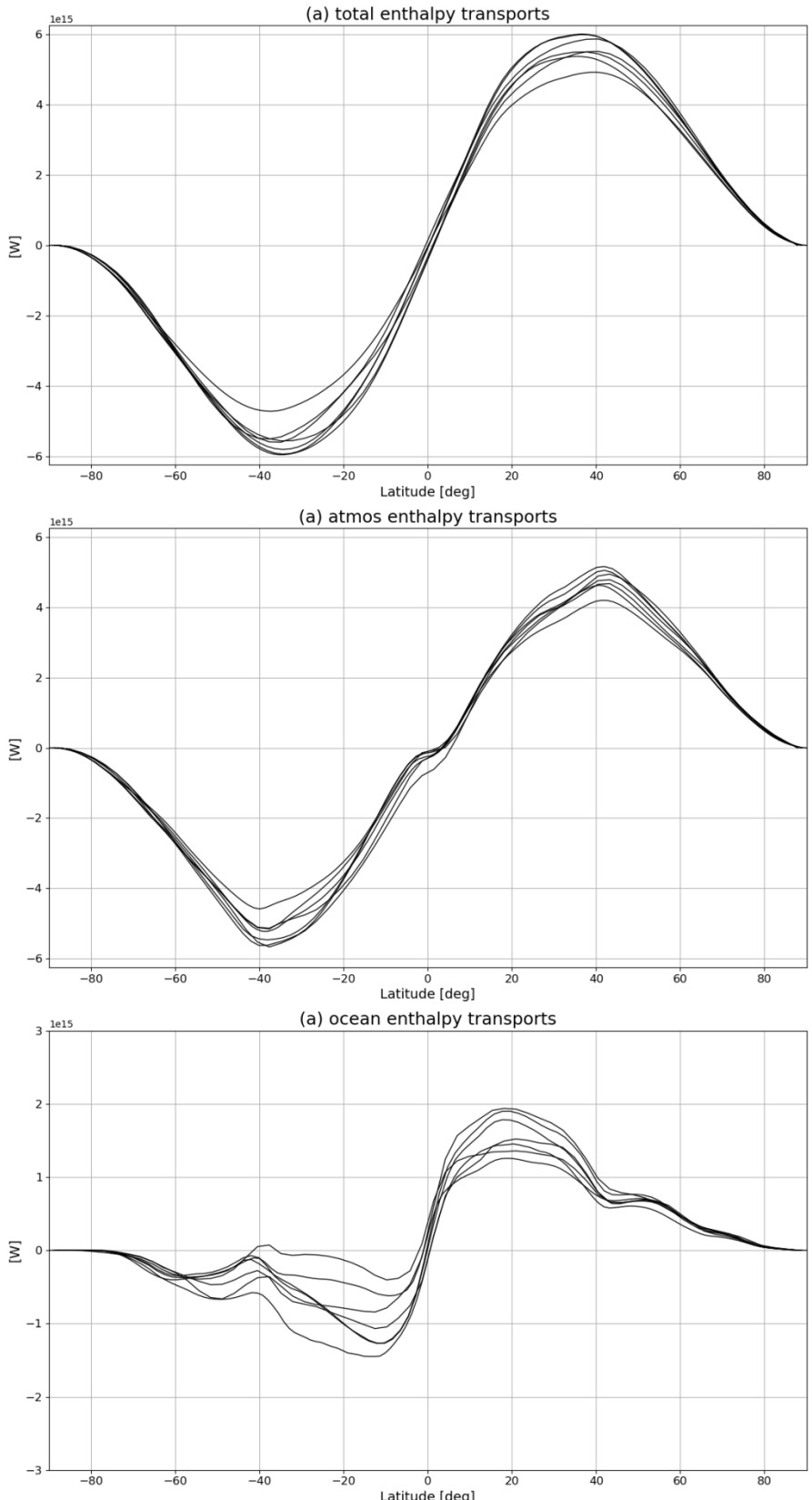


**Figure 12. Annual mean meridional sections of zonal mean meridional total (top), atmospheric (middle), oceanic (bottom) heat transports for 12 CMIP5 models control runs. Transports are implied from meridionally integrating TOA, atmospheric and surface energy budgets (Trenberth et al., 2001), then applying the usual correction accounting for energy imbalances, as in Carissimo et al. (1985). Values are in W. Similar to Figure 8 of Lembo et al. (2019) and**
**produced with *recipe_thermodyn_diagtool.yml*, see details in Section 3.2.4.**

## Niño3.4 SST,TAS,PSL Spatial Composite (DJF$^{+1}$)

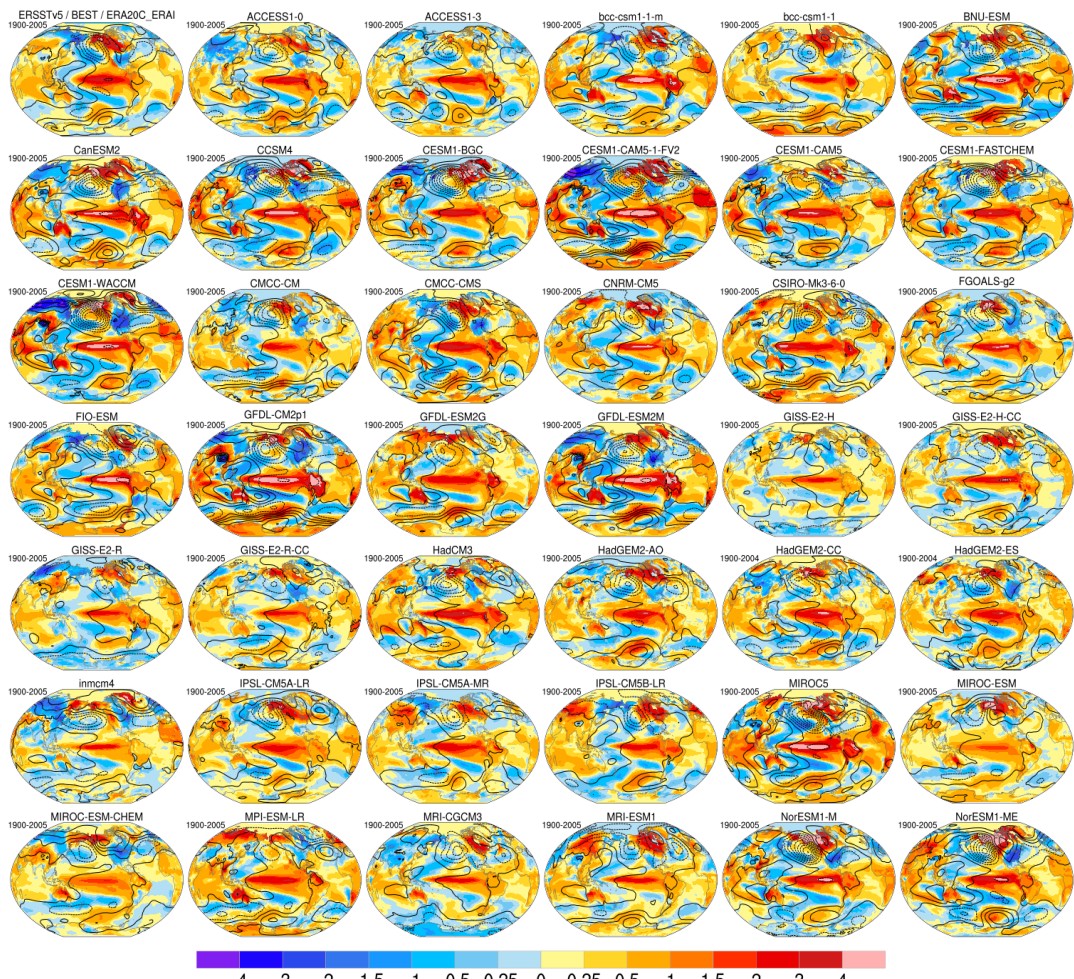

-4  -3  -2  -1.5  -1  -0.5  -0.25  0  0.25  0.5  1  1.5  2  3  4

**Figure 13. Global ENSO teleconnections during the peak phase (December-February) as simulated by 41 CMIP5 models (individual panels labelled by model name) and observations (upper left panel) for the historical period (1900-2005 for models and 1920-2017 for observations). These patterns are based on composite differences between all El Nino events and all La Nina events (using a +/- 1 standard deviation threshold of the Nino3.4 SST Index) occurring in the period of record. Color shading denotes SST and terrestrial TREFHT (◦C), and contours denote SLP (contour interval of 2hPa, with negative values dashed). The period of record is given in the upper left of each panel, and the number of El Nino and La Nina events that contribute to the composites are given in the upper right (for example, "18/14" denotes 18 El Nino events and 14 La Nina events). Observational composites use ERSSTv5 for SST, BEST for TAS and ERA20C updated with ERA-I for PSL. Figure produced with *recipe_CVDP.yml.*, see details in Section 3.2.5.1.**

# AMO (Monthly)

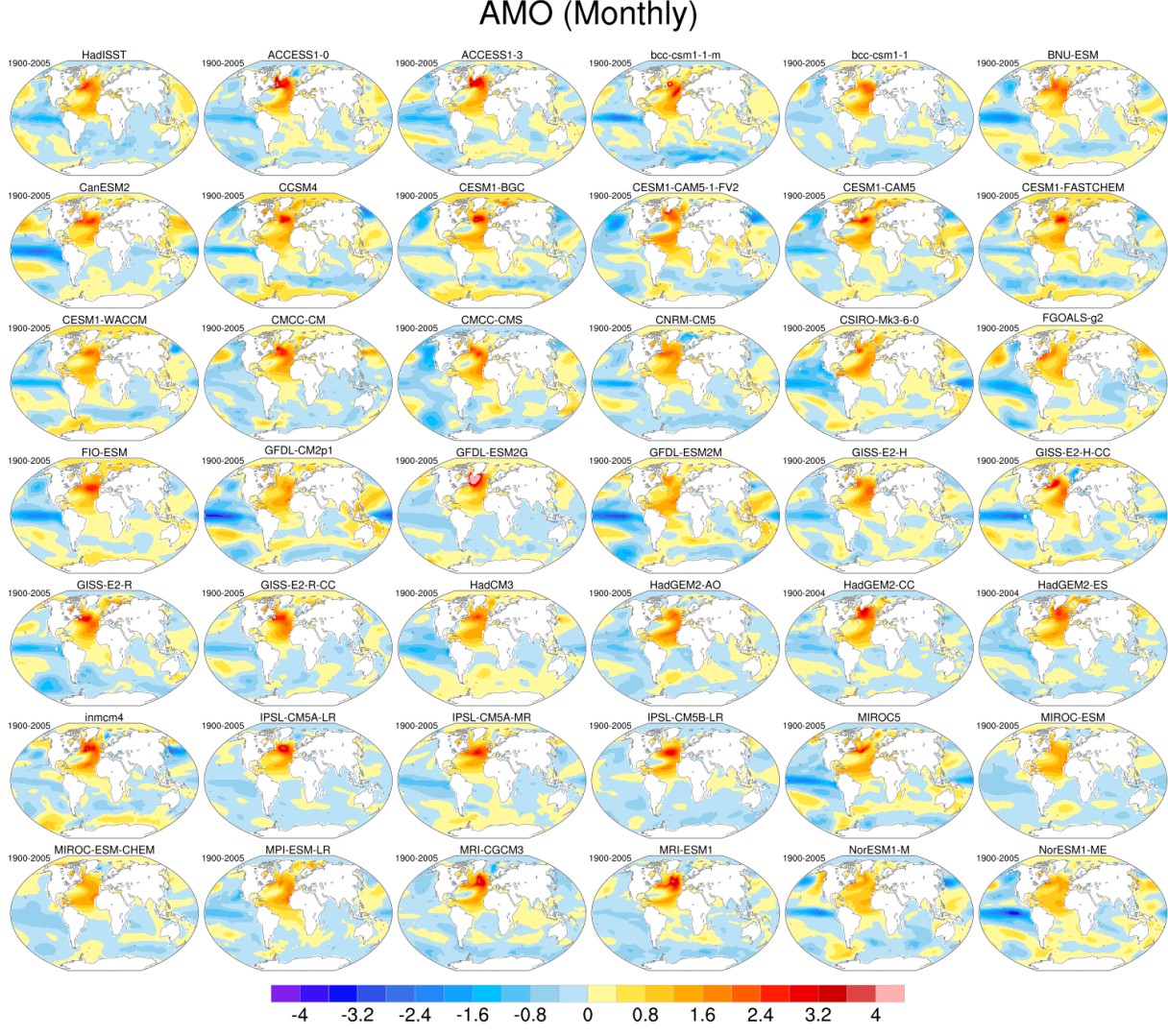

**Figure 14. Representation of the AMO in 41 CMIP5 models (individual panels labelled by model name) and observations (upper left panel) for the historical period (1900-2005 for models and 1920-2017 for observations). These patterns are based regressing monthly SST anomalies (denoted SSTA\*) at each grid box onto the timeseries of the AMO SSTA\* Index (defined as SSTA\* averaged over the North Atlantic 0-60N, 80W-0W), where the asterisk denotes that the global (60N-60S) mean SSTA has been subtracted from SSTA at each grid box following Trenberth and Shea (2006). Figure produced with *recipe_CVDP.yml.*, see details in Section 3.2.5.1.**

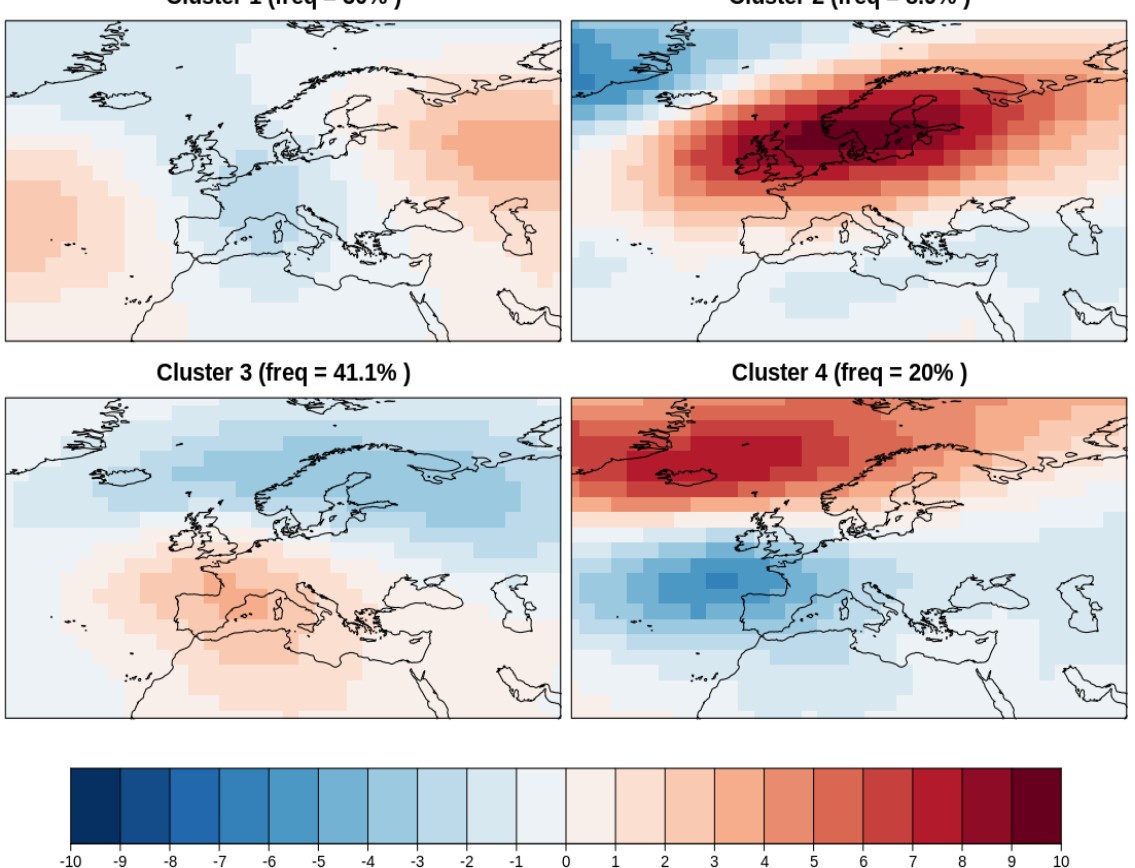

**Figure 15. Four modes of variability for autumn (September-October-November) in the North Atlantic European Sector during the reference period 1971-2000 for the BCC-CSM1-1 historical simulations. The frequency of occurrence of each variability mode is indicated in the title of each map. The four clusters are reminiscent of the Atlantic Ridge, the Scandinavian blocking, the NAO+ and the NAO- pattern, respectively. Result for** *recipe_modes_of_variability.yml***, see details in Section 3.2.5.2.**

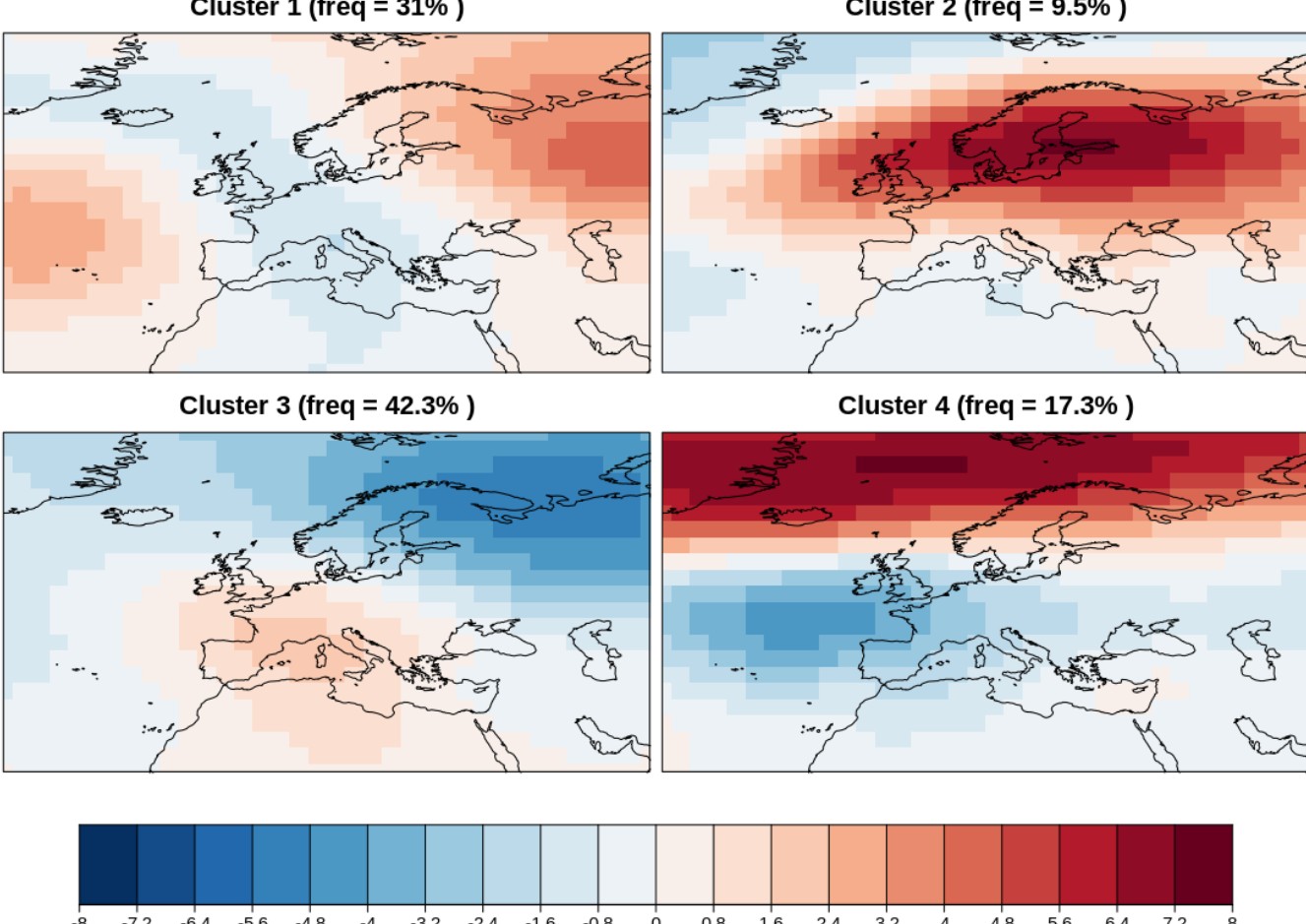

**Figure 16. Four modes of variability for autumn (September-October-November) in the North Atlantic European Sector for the RCP 8.5 scenario using BCC-CSM1-1 future projection during the period 2020-2075. The frequency of occurrence of each variability mode is indicated in the title of each map. The four clusters are reminiscent of the Atlantic Ridge, the Scandinavian blocking, the NAO+ and the NAO- pattern, respectively. Result for** *recipe_modes_of_variability.yml,* **see details in Section 3.2.5.2.**


|       | Obs 1  | Obs 2  | Obs 3  | Obs 4  |
|-------|--------|--------|--------|--------|
| Pre 1 | 107.49 | 349.18 | 304.09 | 375.63 |
| Pre 2 | 280.68 | 149.67 | 405.82 | 449.04 |
| Pre 3 | 303.96 | 497.06 | 112.14 | 505.27 |
| Pre 4 | 415.69 | 529.9  | 491.15 | 122.16 |

**Figure 17. RMSE between the spatial patterns obtained for the future 'Pre' (2020-2075) and the reference 'Obs' (1971-2000) modes of variability from the BCC-CSM1-1 simulations in autumn (September-October-November). Result for *recipe_modes_of_variability.yml* see details in Section 3.2.5.2.**

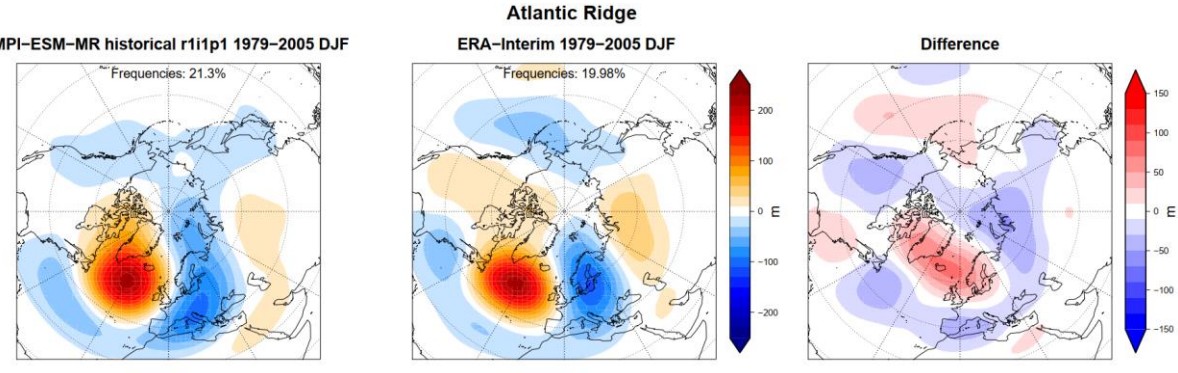

**Figure 18. 500 hPa geopotential height anomalies [m] associated with the Atlantic Ridge weather regime over the 1979-2005 DJF period for (left) CMIP5 MPI-ESM-MR historical r1i1p1 run (center) ERA-Interim reanalysis and (right) their differences. The frequency of occupancy of each regime is reported on the top of each panel. Produced with *recipe_miles_regimes.yml*, see details in Section 3.2.5.2.**

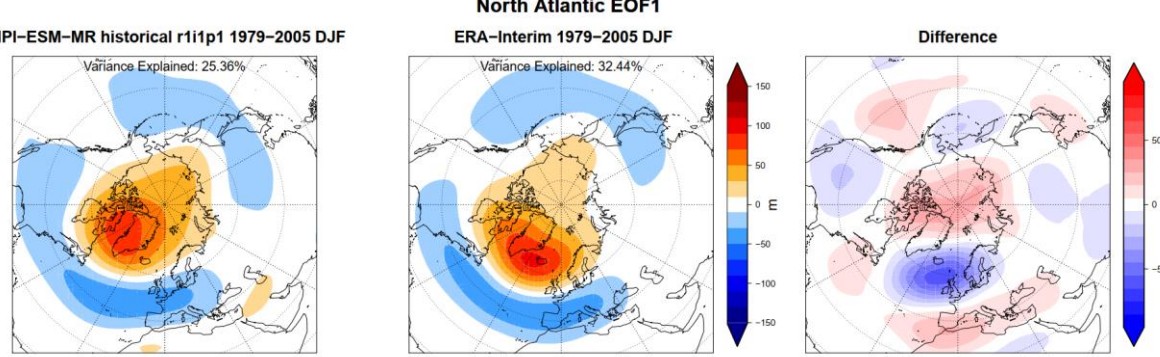


**Figure 19. Linear regression over the 500 hPa geopotential height [m] of the first North Atlantic EOF (i.e. the North Atlantic Oscillation, NAO) over the 1979-2005 DJF period for (left) CMIP5 MPI-ESM-MR historical r1i1p1 run (center) ERA-Interim Reanalysis and (right) their differences. The variance explained is reported on the top of each panel. Produced with *recipe_miles_eof.yml*, see details in Section 3.2.5.3.**


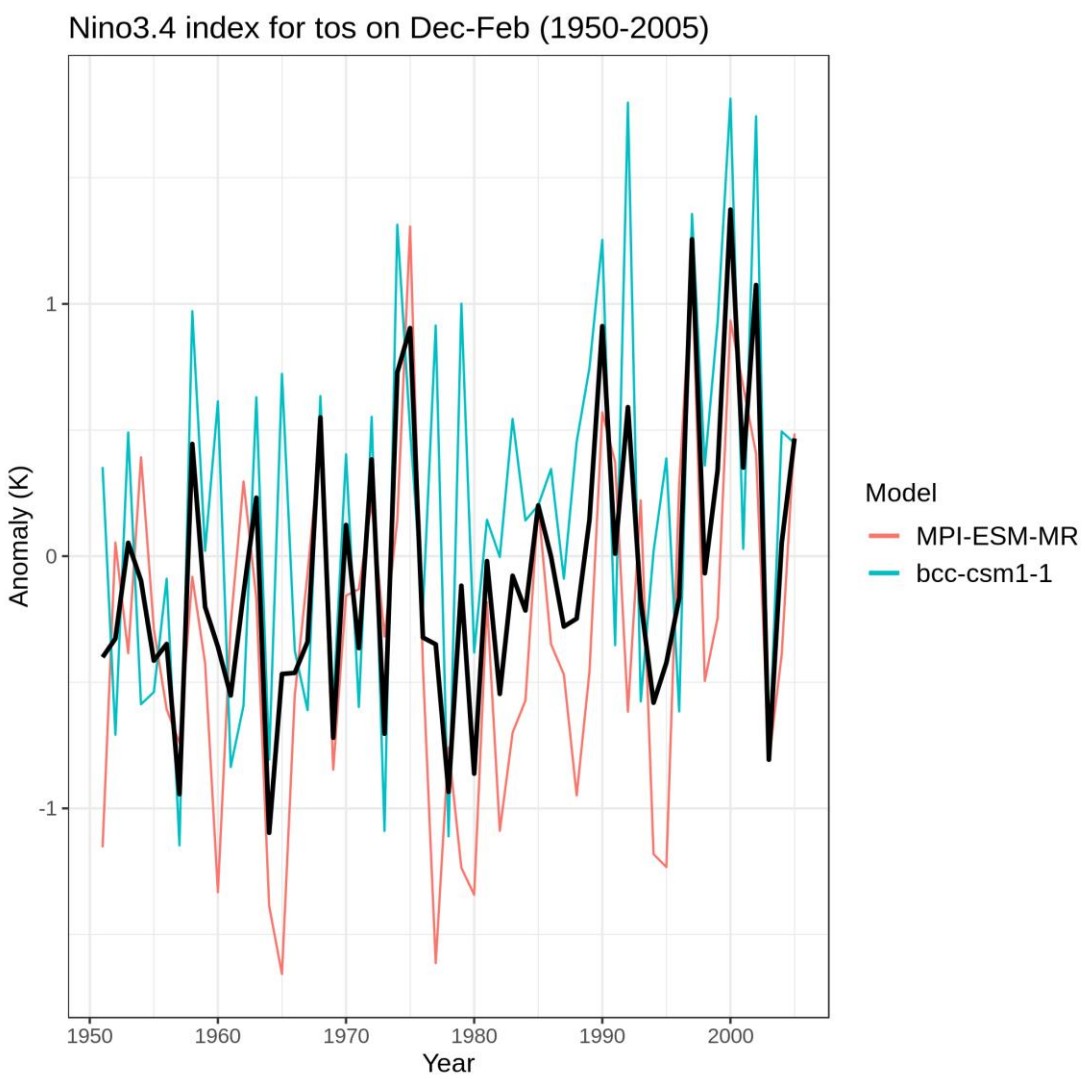

**Figure 20. Time series of the standardized sea surface temperature (tos) area averaged over the Nino 3.4 region during the boreal winter (December-January-February). The time series correspond to the MPI-ESM-MR (red) and BCC-CSM1-1 (blue) models and their mean (black) during the period 1950-2005 for the ensemble r1i1p1 of the historical simulations. Produced with _recipe_combined_indices.yml._, see details in Section 3.2.5.4.**

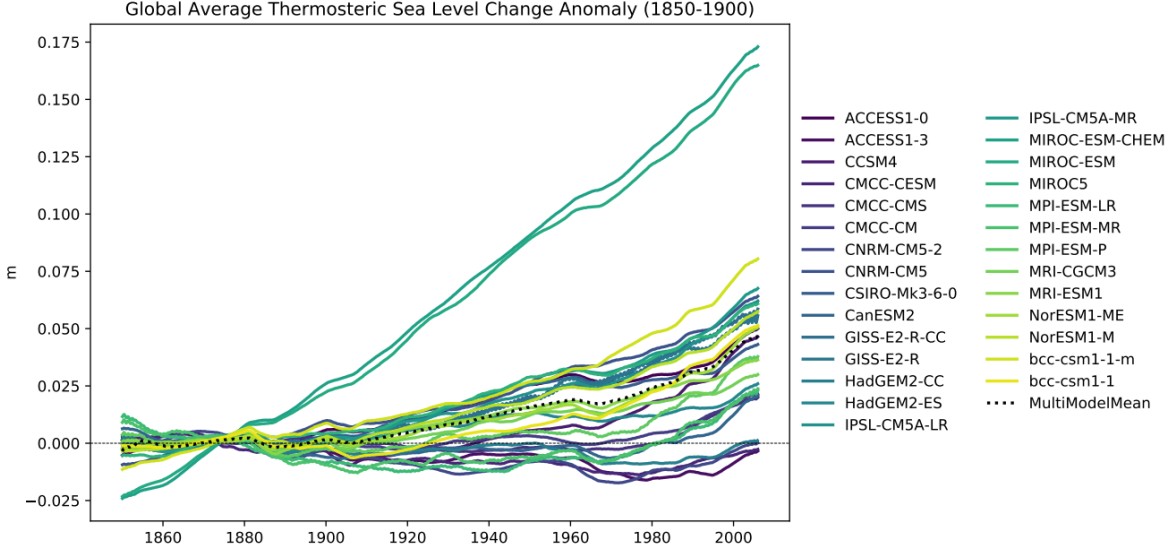

**Figure 21. The volume weighted thermosteric sea level change anomaly in several CMIP5 models, in the historical experiment, in the r1i1p1 ensemble member, with a 6 year moving average smoothing function. The anomaly is calculated against the mean of all years in the historical experiment before 1900. The multi-model mean is shown as a dashed line. Produced with *recipe_ocean_scalar_fields.yml* described in Section 3.3.1.**

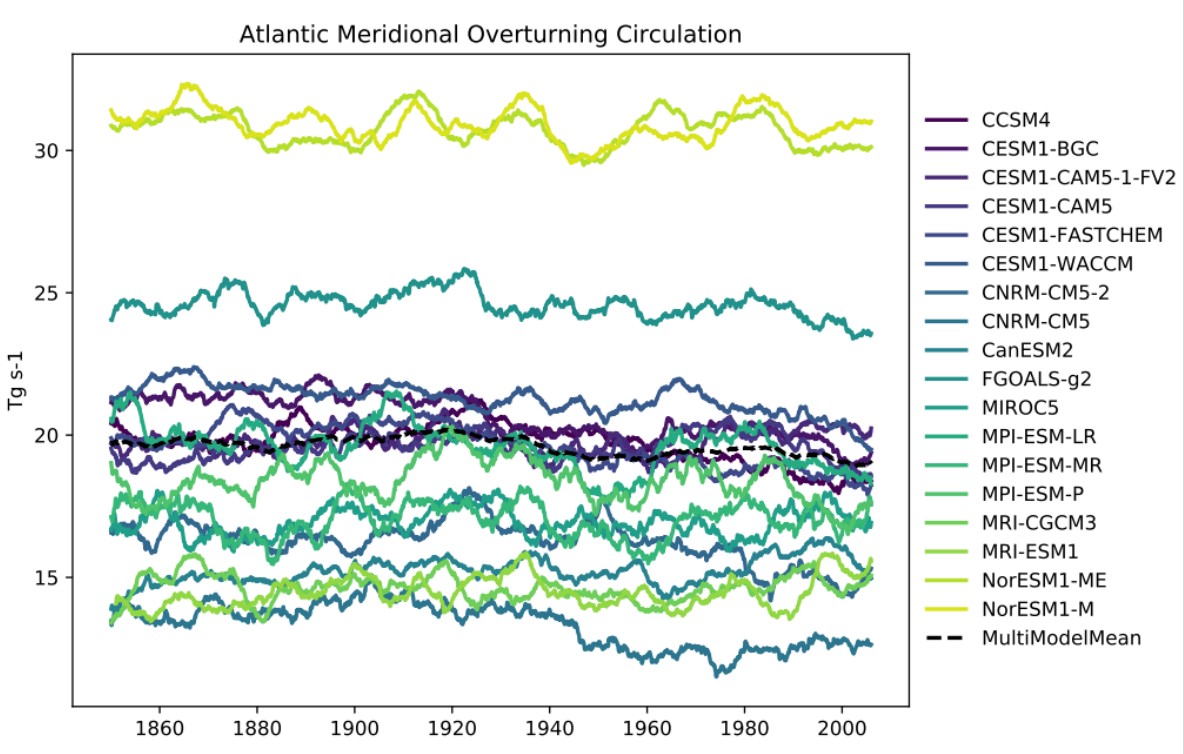

**Figure 22. The Atlantic Meridional Overturn Circulation (AMOC) in several CMIP5 models, in the historical experiment, in the r1i1p1 ensemble member, with a 6 year moving average smoothing function. The multi-model mean is shown as a dashed line. The AMOC indicates the strength of the northbound current and this current transfers heat from tropical water to the North Atlantic. Produced with *recipe_ocean_amoc.yml* described in Section 3.3.1.**


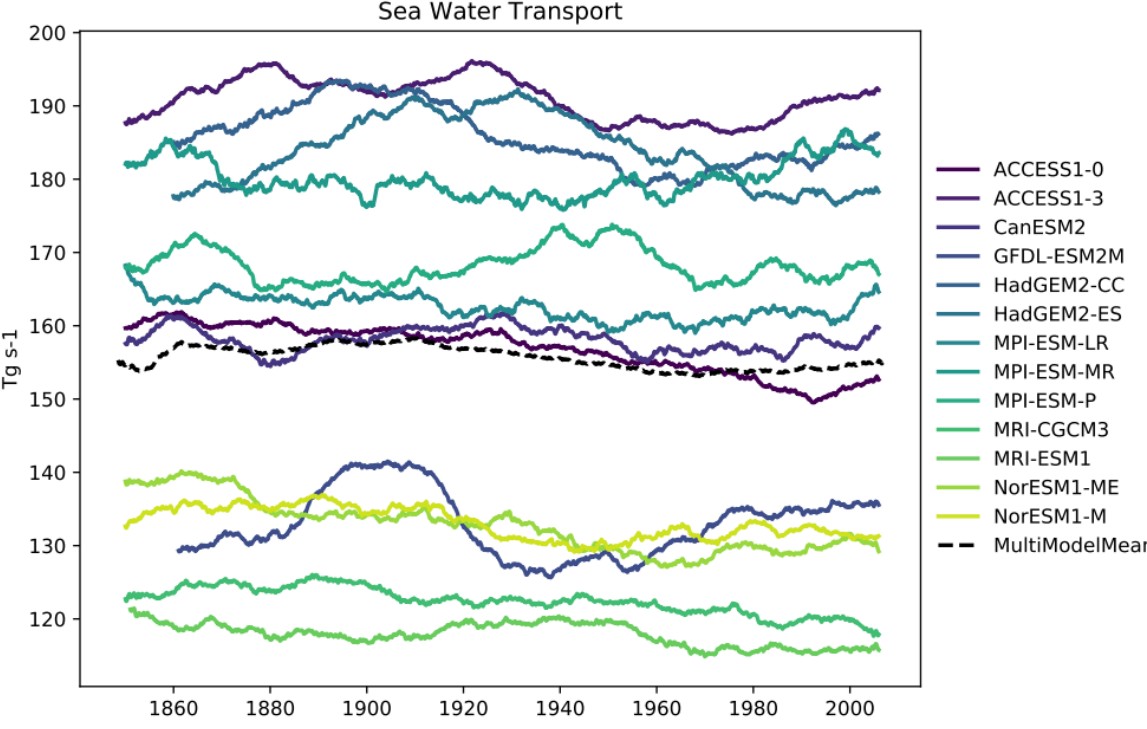


**Figure 23. The Antarctic circumpolar current calculated through Drake Passage for a range of CMIP5 models in the historical experiment in the r1i1p1 ensemble member, with a 6 year moving average smoothing function. The multi-model mean is shown as a dashed line. Produced with *recipe_ocean_amoc.yml* described in Section 3.3.1.**

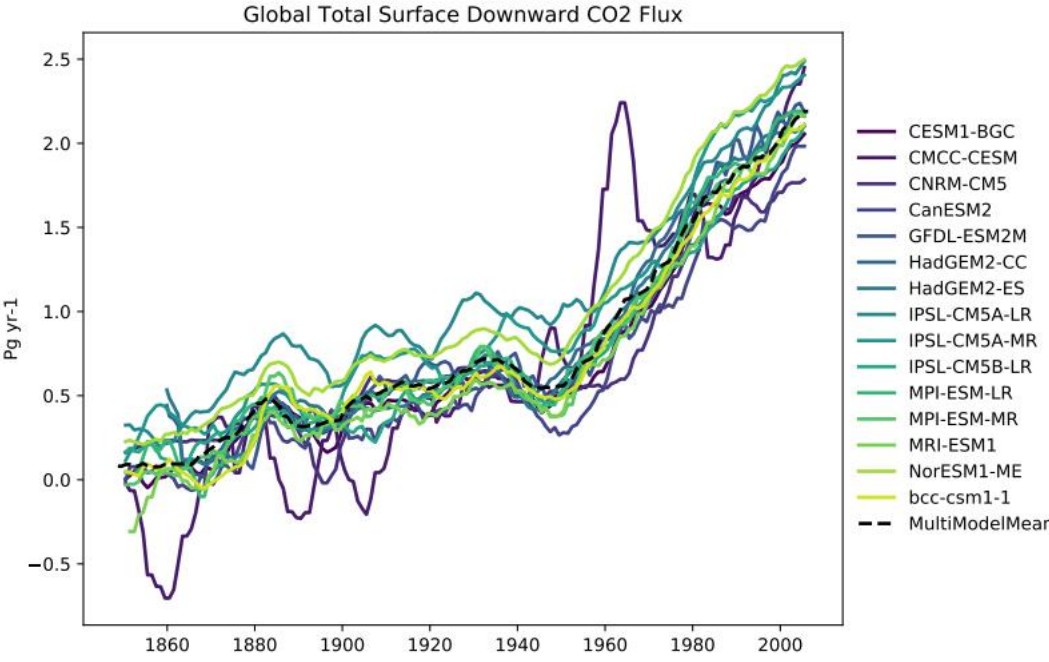


**Figure 24. The global total air to sea flux of CO₂ for a range of CMIP5 models in the historical experiment in the r1i1p1 ensemble member, with a 6 year moving average smoothing function. The multi-model mean is shown as a dashed line. Produced with *recipe_ocean_scalar_fields.yml* described in Section 3.3.1.**

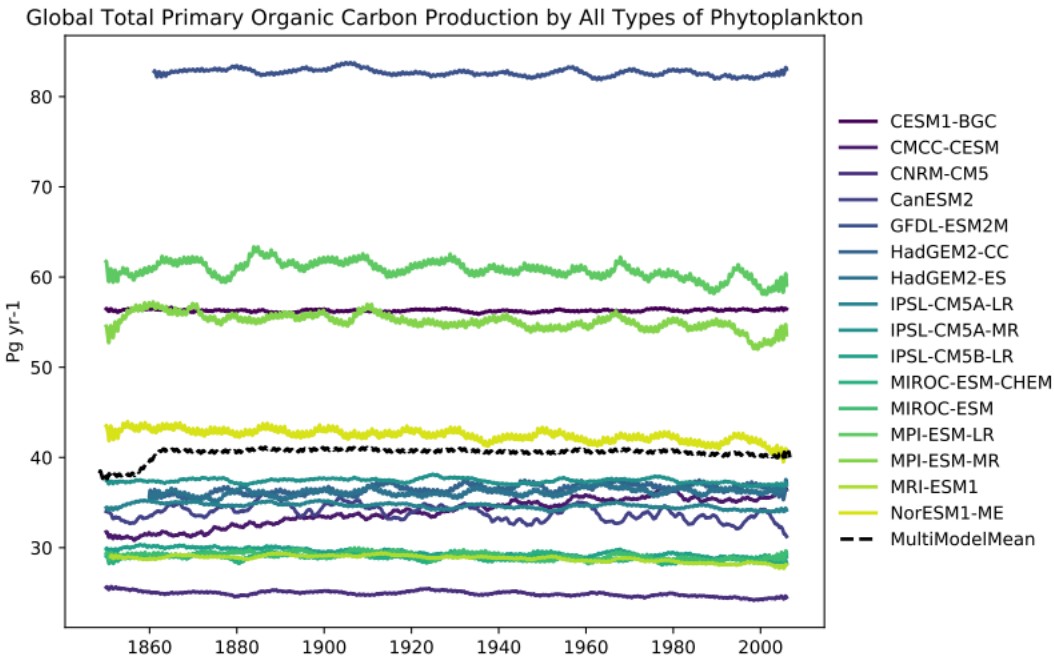


**Figure 25. The global total integrated primary production from phytoplankton for a range of CMIP5 models in the historical experiment in the r1i1p1 ensemble member, with a 6 year moving average smoothing function. The multi-model mean is shown as a dashed line. Produced with *recipe_ocean_scalar_fields.yml* described in Section 3.3.1.**

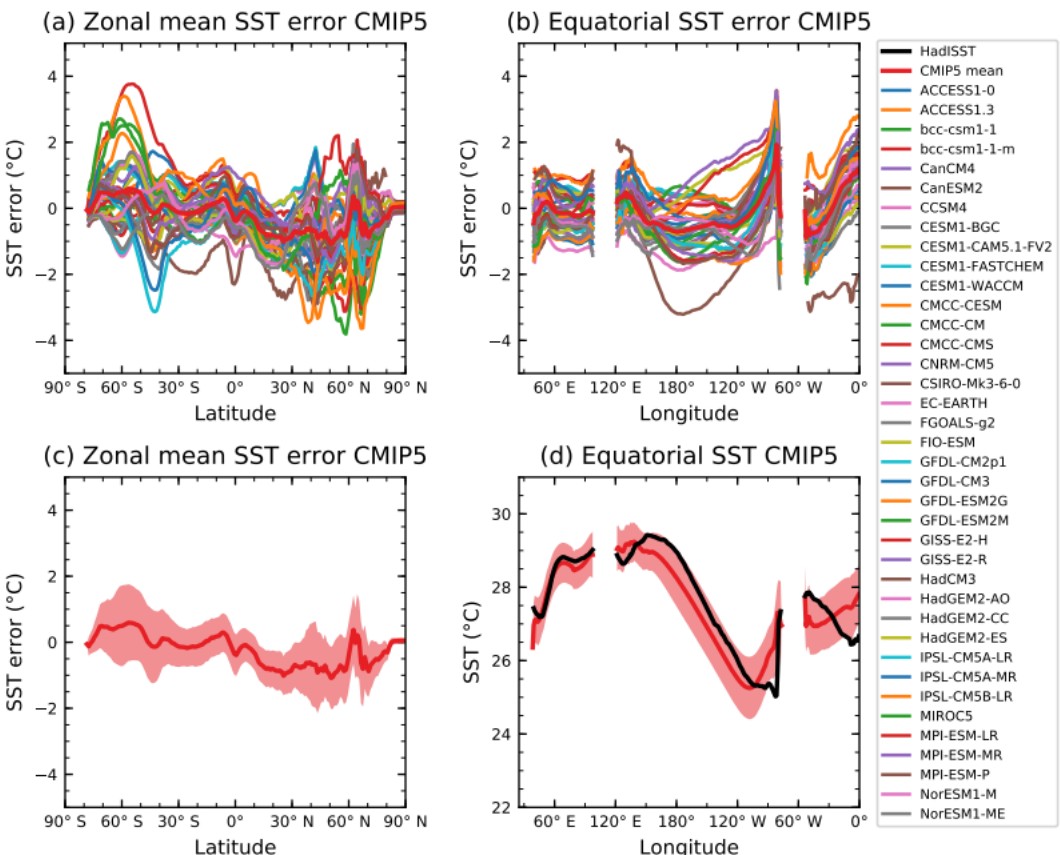


**Figure 26. (a) Zonally averaged sea surface temperature (SST) error in CMIP5 models. (b) Equatorial SST error in CMIP5 models. (c) Zonally averaged multi-model mean SST error for CMIP5 together with inter-model standard deviation (shading). (d) Equatorial multi-model mean SST in CMIP5 together with inter-model standard deviation (shading) and observations (black). Model climatologies are derived from the 1979-1999 mean of the historical simulations. The Hadley Centre Sea Ice and Sea Surface Temperature (HadISST, Rayner et al. (2003)) observational climatology for 1979-1999 is used as a reference for the error calculation (a), (b), and (c); and for observations in (d). Updated from Fig. 9.14 of IPCC WG I AR5 Chapter 9 (Flato et al., 2013) and produced with** *recipe_flato13ipcc.yml,* **see details in 3.3.1.**


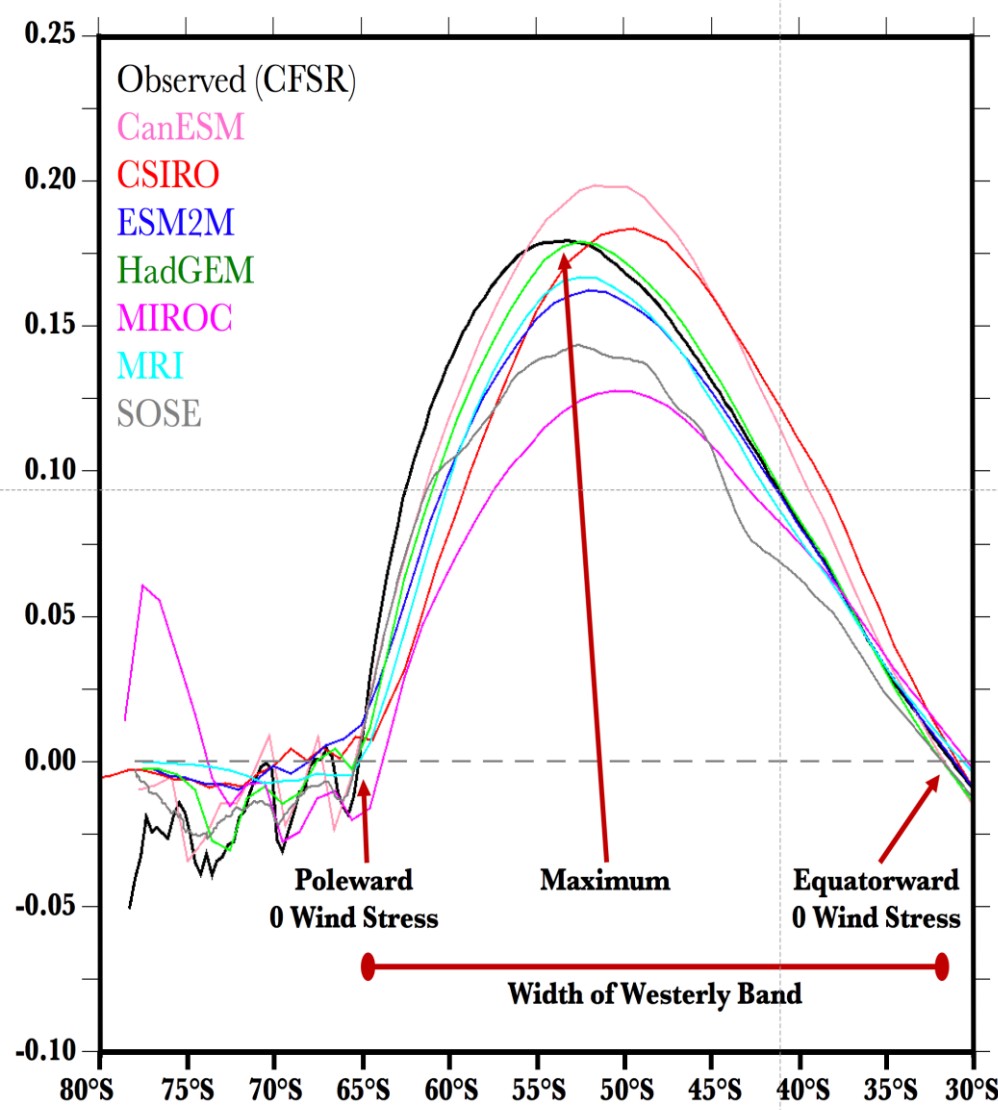

**Figure 27. The zonal and annual means of the zonal wind stress (N/m2) for the reanalysis, six of the CMIP5 simulations and the BSOSE state estimate—note that each of the model simulations (colors) and B-SOSE (gray) have the peak wind stress equatorward of the observations (black). Also shown are the latitudes of the observed "poleward zero wind stress" and the "equatorward zero wind stress" which delineate the "width of the westerly band" that is highly correlated with total heat uptake by the Southern Ocean. Enhanced from figure produced by *recipe_russell18jgr.yml*. see Section 3.3.2. For further discussion of this figure, see the original in Russell et al. (2018).**


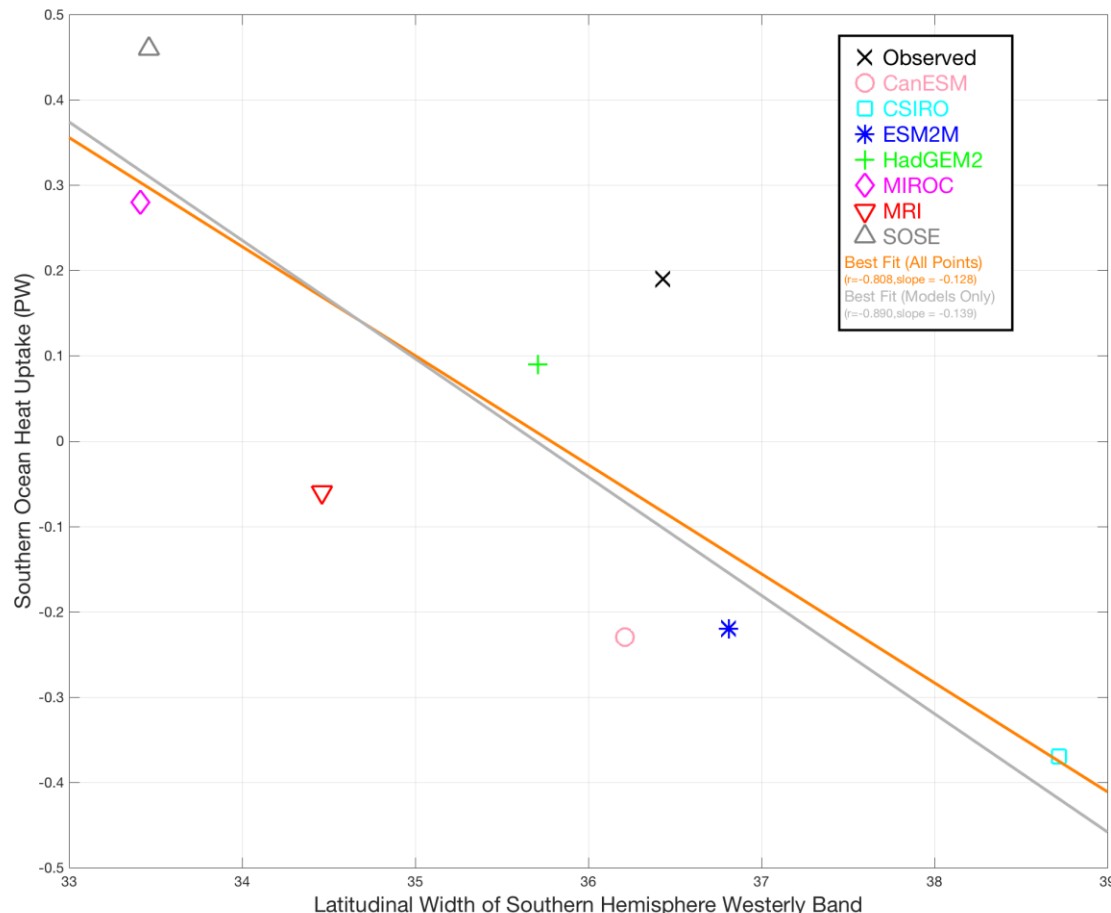

**Figure 28.** Scatter plot of the width of the Southern Hemisphere westerly wind band (in degrees of latitude) against the annual-mean integrated heat uptake south of 30°S (in PW—negative uptake is heat lost from the ocean), along with the ''best fit'' linear relationship for the models and observations shown. Enhanced from figure produced by *recipe_russell18jgr.yml*. see in Section 3.3.2. For further discussion of this figure, see the original in Russell et al. (2018). The calculation of the ''observed'' heat flux into the Southern Ocean is described in the text. The correlation is significant above the 98% level based on a simple t test.

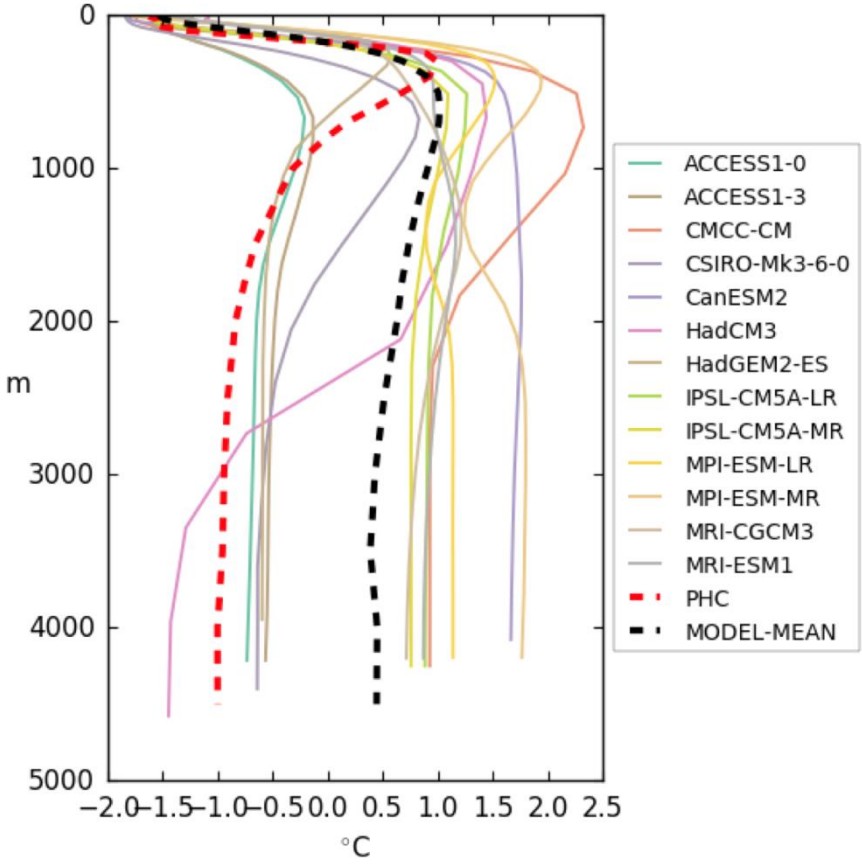

**Figure 29. Mean (1970-2005) vertical potential temperature distribution in the Eurasian basin for CMIP5 coupled ocean models, PHC3 climatology (dotted red line) and multi-model mean (dotted black line). Similar to Figure 7 of Ilıcak et al. (2016) and produced with *recipe_arctic_ocean.yml*, see details in Section 3.3.3.**


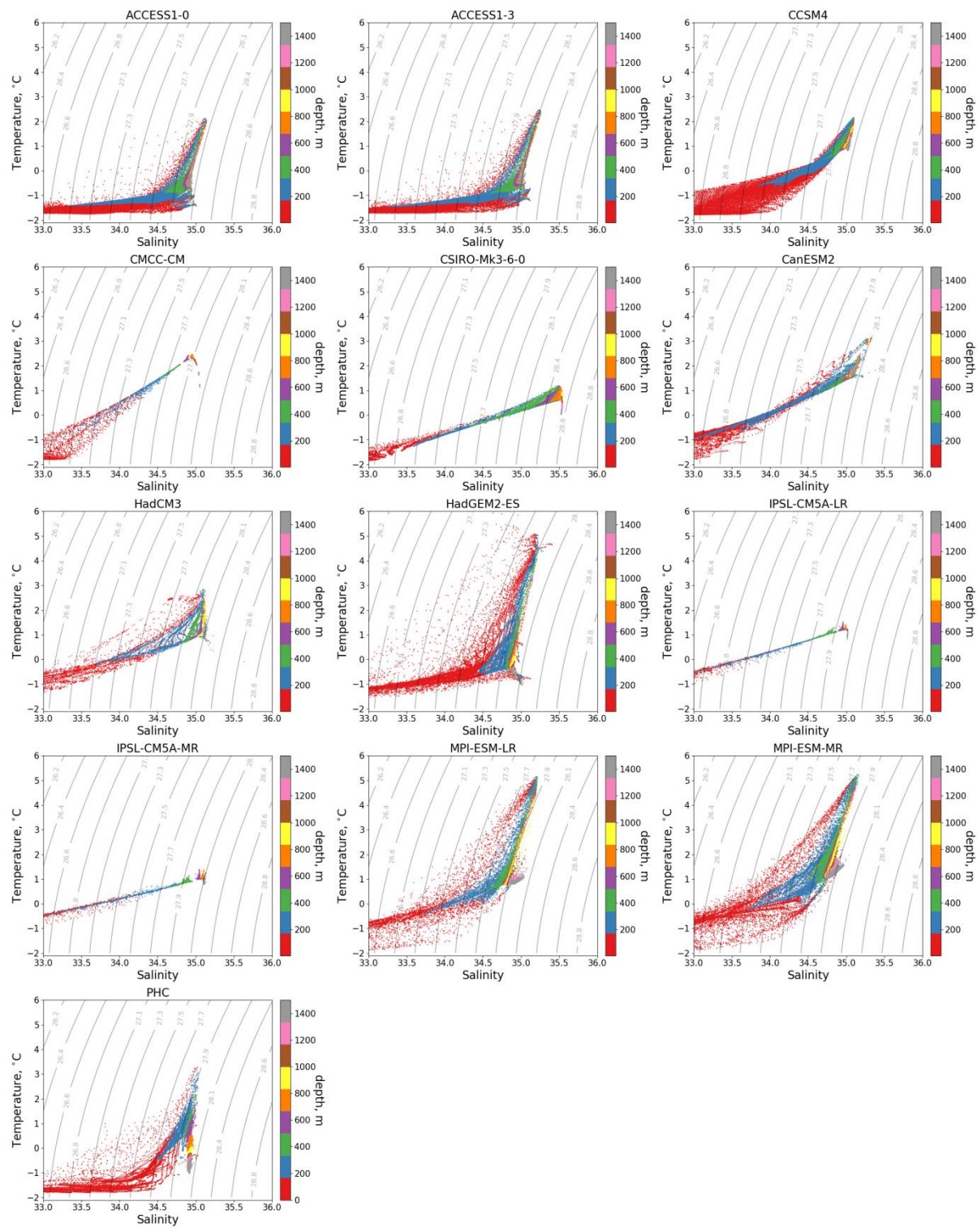

**Figure 30. Mean (1970-2005) T-S diagrams for Eurasian Basin of the Arctic Ocean. PHC3.0 shows climatological values for selected CMIP5 models and PHC3.0 observations. Produced with recipe_arctic_ocean.yml , see details in Section 3.3.3.**

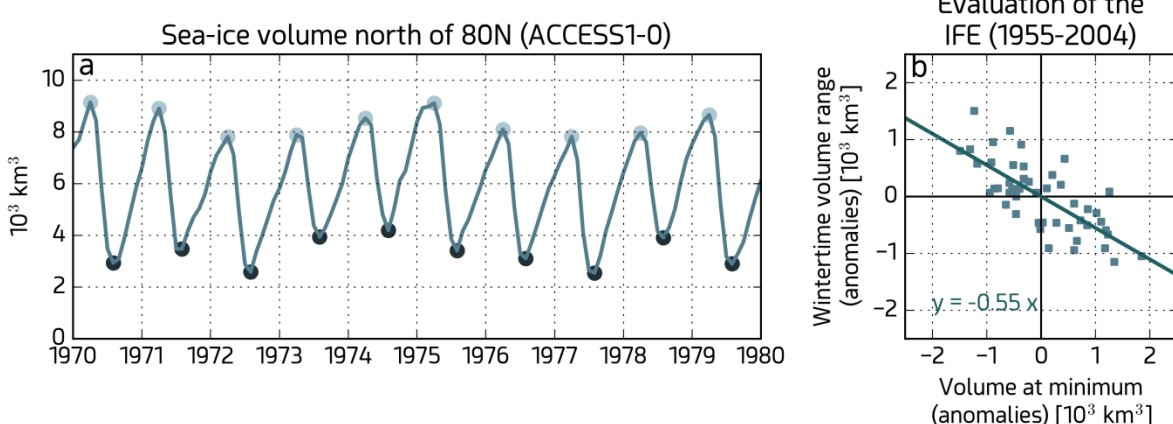

**Figure 31. Quantitative evaluation of the Ice Formation Efficiency (IFE). (a) Example time series (1970-1979) of the monthly mean Arctic sea ice volume north of 80°N of one CMIP5 model (ACCESS1-0), with its annual minimum and maximum values marked with the dark and light dots, respectively. (b) Estimation of the IFE, defined as the regression between anomalies of sea ice volume produced during the growing season (difference between one annual maximum and the preceding minimum) and anomalies of the preceding minimum. A value IFE = -1 means that the late-summer ice volume anomaly is fully recovered during the following winter (strong negative feedback damping all anomalies), while a value IFE = 0 means that the wintertime volume production is essentially decoupled from the late-summer anomalies (inexistent feedback). Similar to Extended Data Figure 7a-b of Massonnet et al. (2018a) and produced with *recipe_seaice_feedback.yml*, see details in Section 3.3.4.**

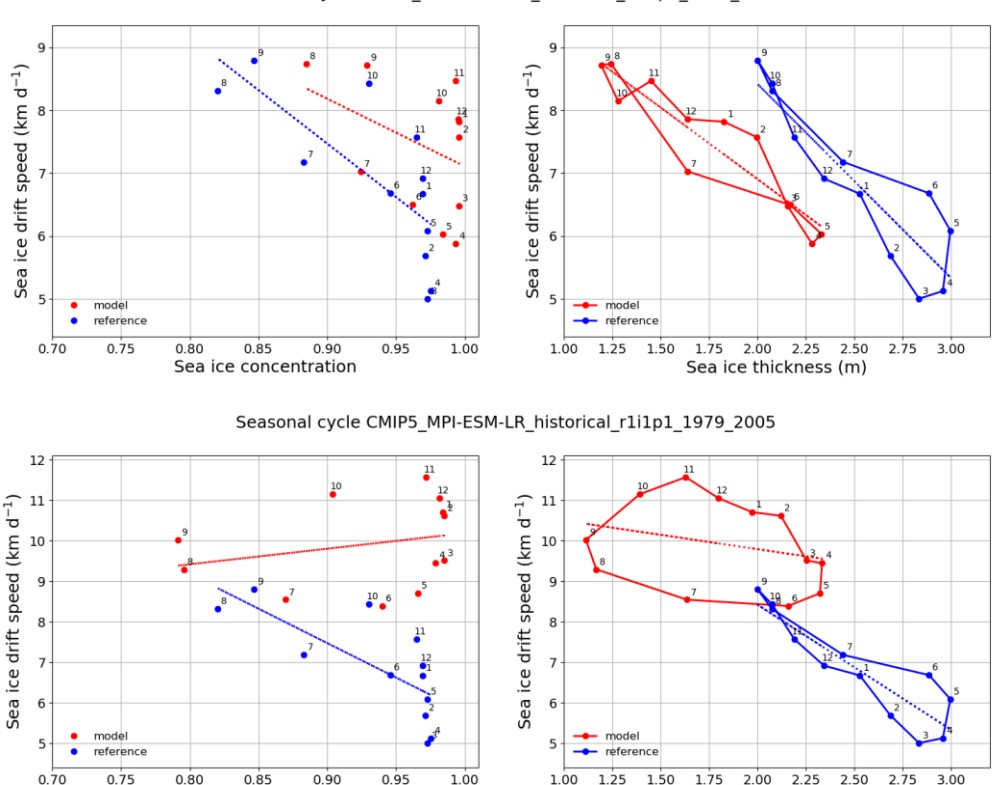

**Figure 32. Scatter plots of modelled (red) and observed (blue) monthly mean sea ice drift speed against sea ice concentration (left panels) and sea ice thickness (right panels) temporally averaged over the period 1979–2005 and spatially averaged over the SCICEX box. Top panels show results from the GDFL-ESM2G model and bottom panels show results from the MPI-ESM-LR model (CMIP5 historical runs). Observations/reanalysis are shown in all panels (IABP for drift speed, OSI-450 for concentration, and PIOMAS for thickness). Numbers denote months. Dotted lines show linear regressions. This figure was produced in a similar way as Figure 4 of Docquier et al. (2017) with *recipe_sea_ice_drift.yml*, see details in Section 3.3.4.**

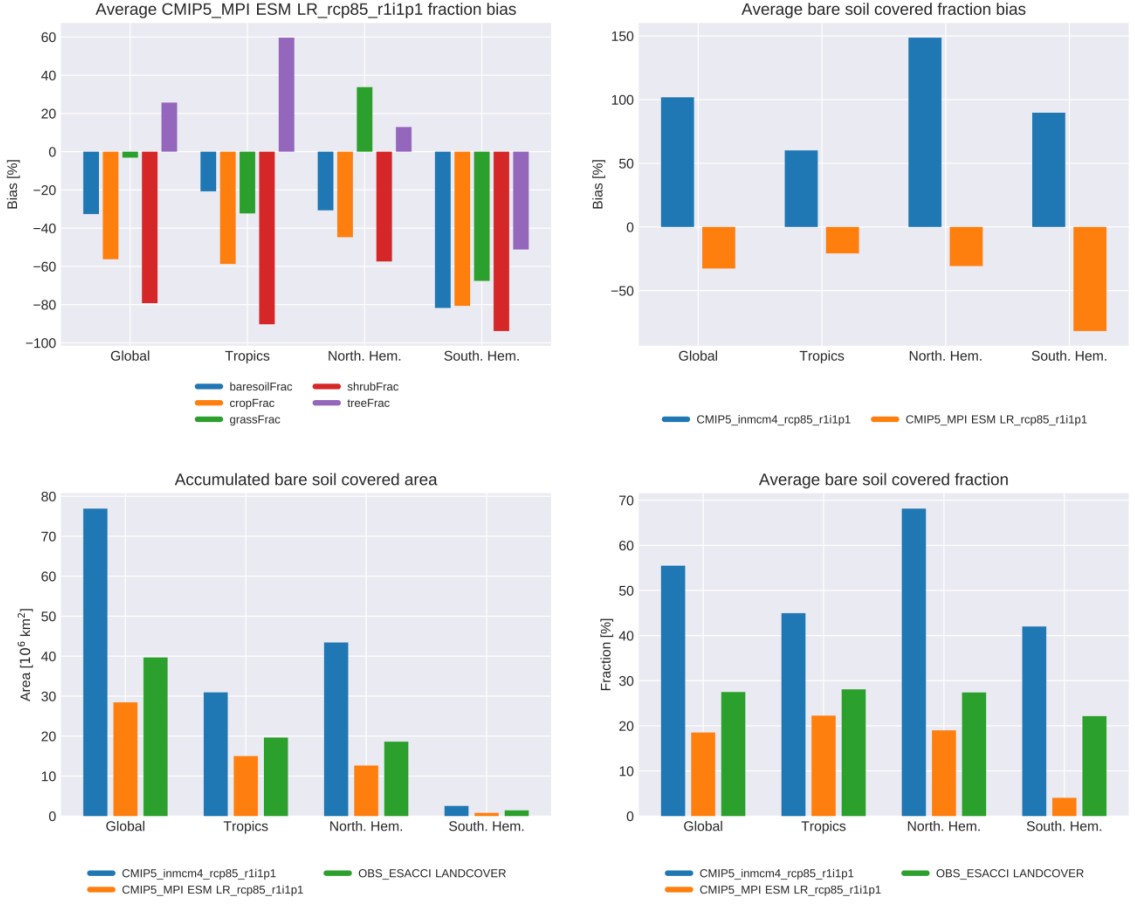

**Figure 33. The panels show plots produced by the metric *recipe_landcover.yml* using model output from historical CMIP5 simulations (period 2008-2012) of the ESMs MPI-ESM and INMCM4 compared to land cover observations provided by ESA CCI for different regions. The upper two panels display the relative bias [%] between the models M and the observation O computed as (M – O) / O x 100. This can be visualized either for one model (i.e. MPI-ESM) and several land cover types (upper, left) or for one land cover type (i.e. bare soil fraction) and all selected models (upper, right). The lower plots display the area [10$^6$ km$^2$] covered by a specific land cover type (i.e. bare soil fraction) for given regions (lower, left) as well as the average cover fractions [%] (lower, right) with respect to the total area of the regions. Thus, the landcover analysis provides a quick overview for major land cover types and the ability of different models to reproduce them. The metric is based on the analysis presented in Lauer et al. (2017) and Georgievski and Hagemann (2018) and discussed in Section 3.4.1.**

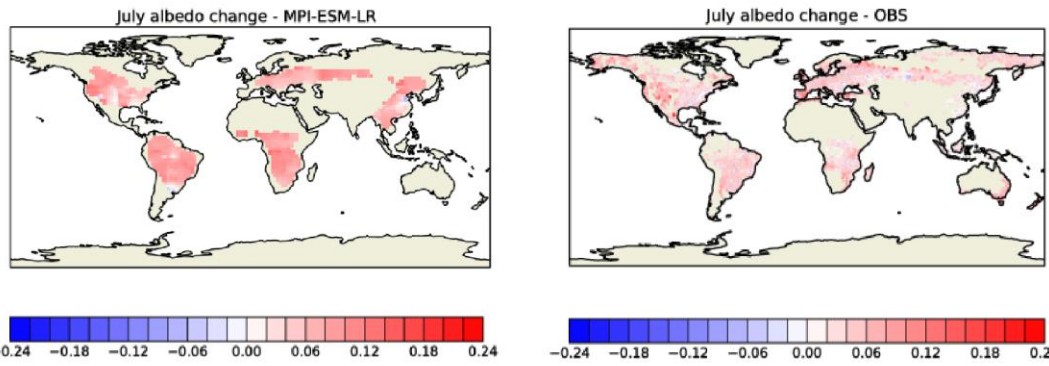

**Figure 34. Potential albedo change due to a transition from landcover type 'tree' to 'crops and grasses' calculated through a multiple linear regression between the present-day land cover fractions (predictors) and albedo (predictands) within a moving window encompassing 5° x 5° grid cells. Results are shown for (left) the MPI-ESM-LR model (2001-2005 July mean) and (right) the observational dataset from Duveiller et al. (2018a) (2008-2012 July mean). Produced with *recipe_landcoveralbedo.yml*, see details in Section 3.4.2 and in Lejeune et al. (2020).**


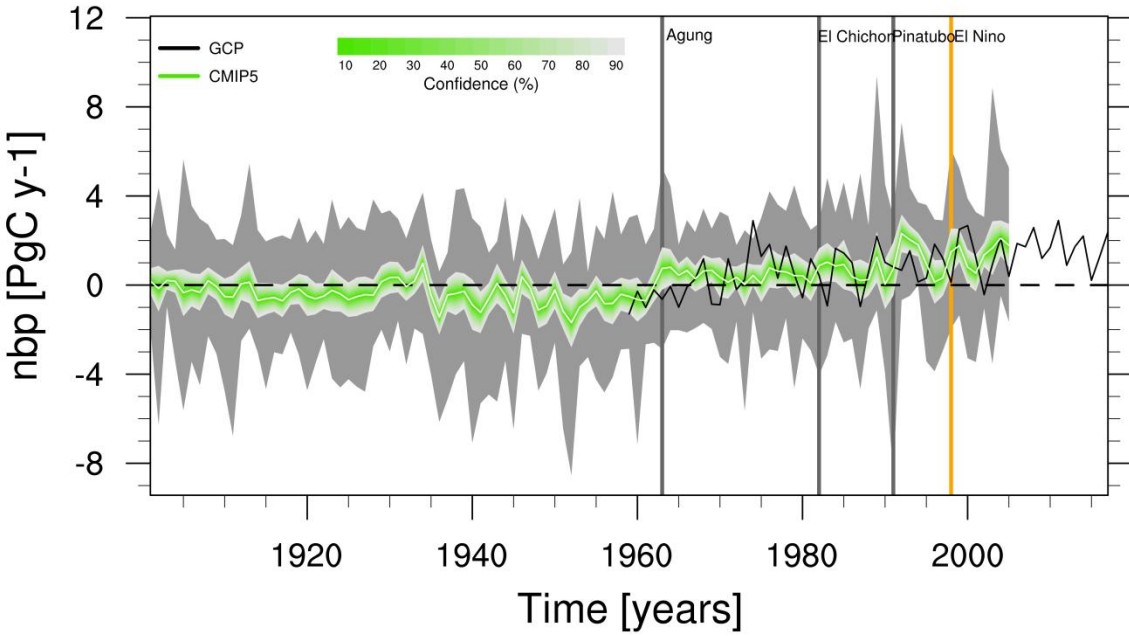

**Figure 35. Timeseries plot of the global land-atmosphere CO₂ flux (*nbp*) for CMIP5 models compared to observational estimates from GCP (Le Quere et al., 2018) (black line). Gray shading represents the range of the CMIP5 models, green shading shows the confidence interval evaluated from the CMIP5 ensemble standard deviation assuming a *t* distribution centered at the multi-model mean (white line). Vertical lines indicate volcanic eruptions (grey) and El Niño events (orange). Similar to Figure 5 of Anav et al. (2013) and produced with *recipe_anav13jclim.yml*, see details in Section 3.5.1.**

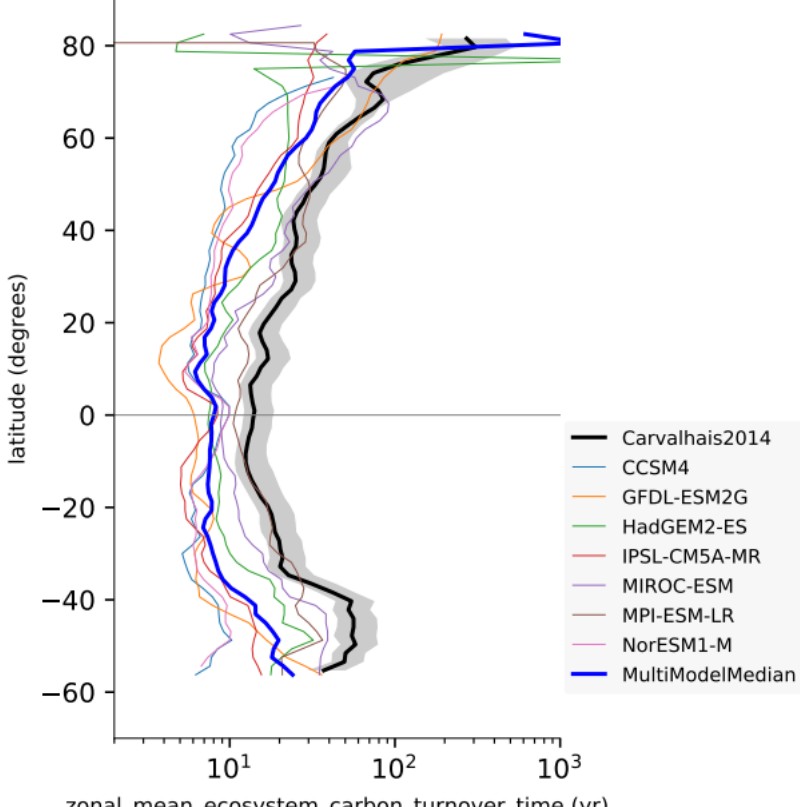

**Figure 36. Zonal distribution of ecosystem turnover time of carbon (in years). The zonal values are calculated as the ratio of total carbon stock and the gross primary productivity per latitude. The individual models are plotted in coloured thin lines, the multimodel ensemble in thick blue line, and the observation-based estimate (Carvalhais et al., 2014) in thick black line with shaded region showing the observational uncertainty. The median of all models is adopted as the multimodel ensemble. Note the logarithmic horizontal axis. Produced with**
***recipe_carvalhais2014nat.yml*, see details in Section 3.5.2.**

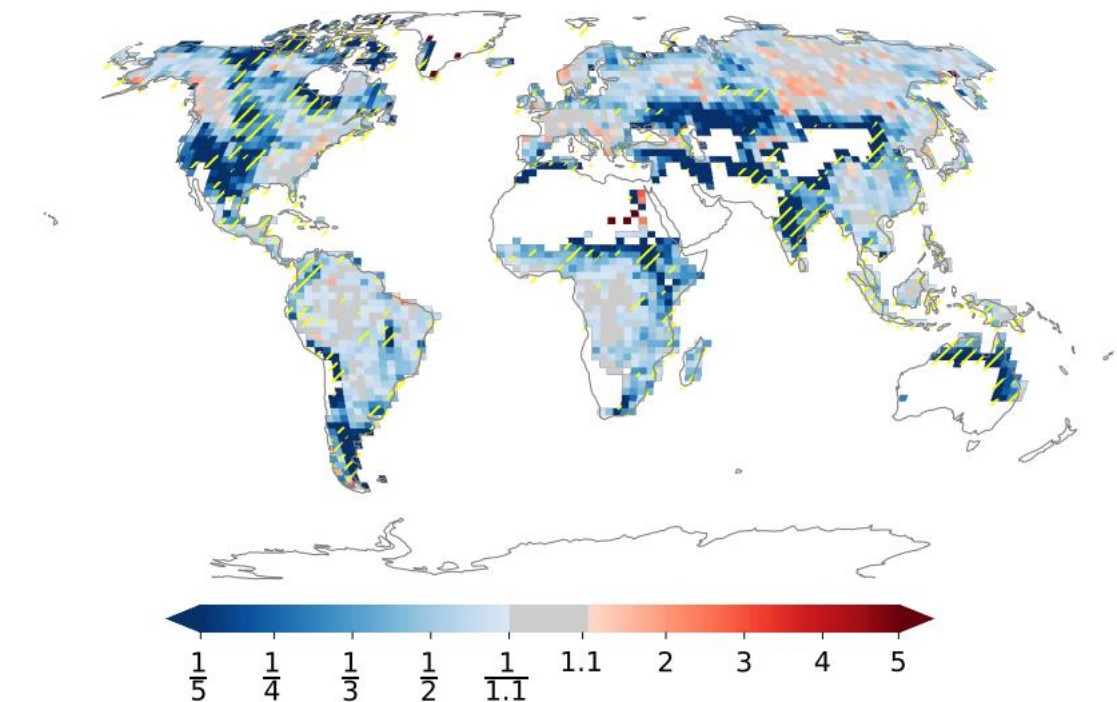

**Figure 37. Global distribution of the biases in the multi-model ensemble ecosystem turnover time of carbon (years) and the multi-model agreement in CMIP5 models. The bias is calculated as the ratio between multi-model ensemble and observation-based estimate (Carvalhais et al., 2014). The stippling indicates the regions where only two or fewer models (out of 10) are within the range of observational uncertainties (5[th] and 95[th] percentiles). Produced with *recipe_carvalhais2014nat.yml*, see details in Section 3.5.2.**


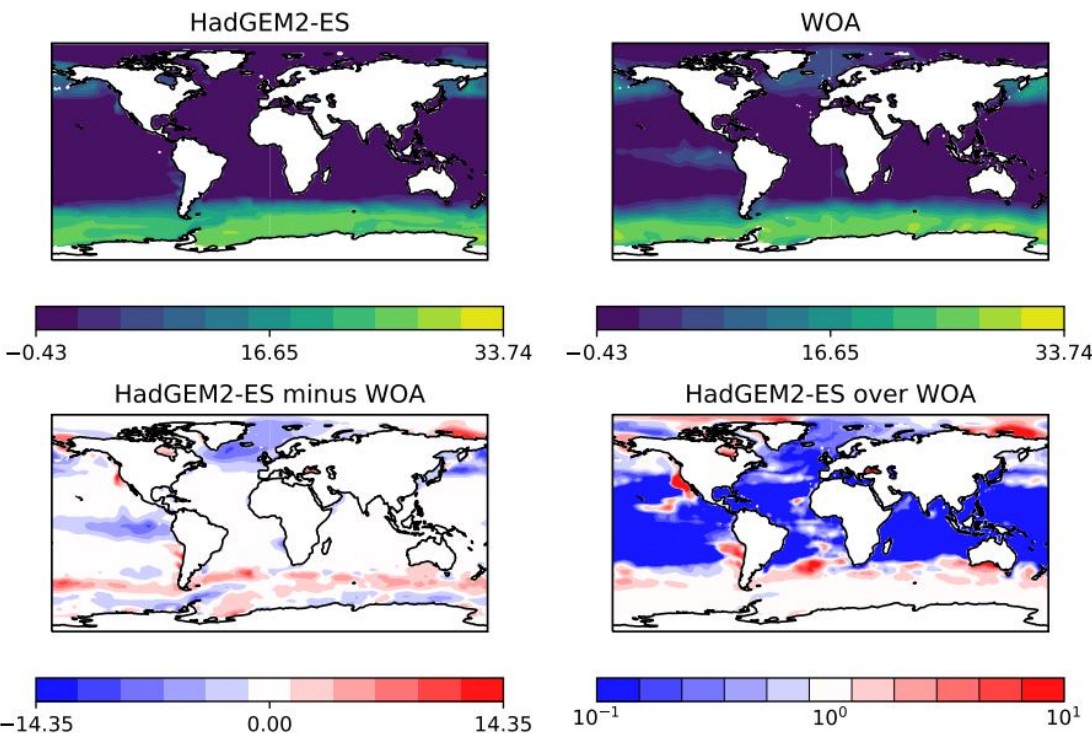

Figure 38. The surface dissolved nitrate concentration in the CMIP5 HadGEM2-ES model compared against the World Ocean Atlas 2013 nitrate. The top two figures show the surface fields, the bottom two show the difference and the quotient between the two datasets. Produced with *recipe_ocean_bgc.yml*, see details in Section 3.5.3.


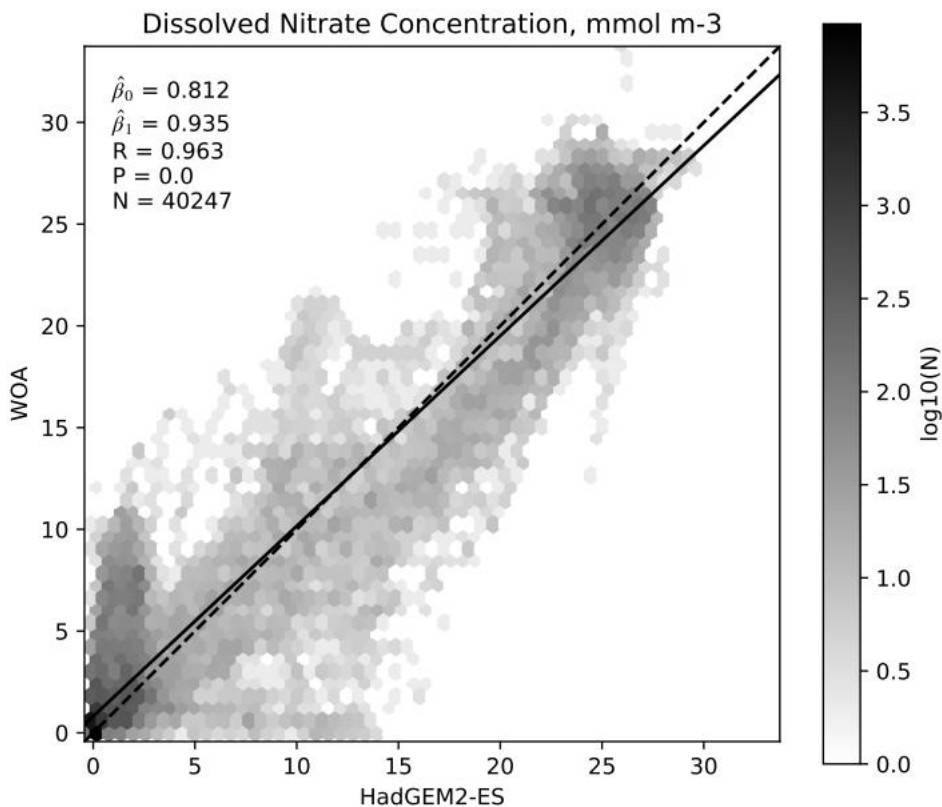

Figure 39. The surface dissolved nitrate concentration in the CMIP5 HadGEM2-ES model [log10(N)] compared against the World Ocean Atlas 2013 nitrate. This figure shows the paired model and observational datasets. A linear regression line of best fit is shown as a black line. A dashed line indicates the 1:1 line. The result of a linear regression are shown in the top left corner of the figure, where $\hat{\beta}_0$ is the intersect, $\beta_1$ is the slope, R is the correlation, P is the P value, and N is the number of data point pairs. Produced with *recipe_ocean_bgc.yml*, see details in Section 3.5.3.



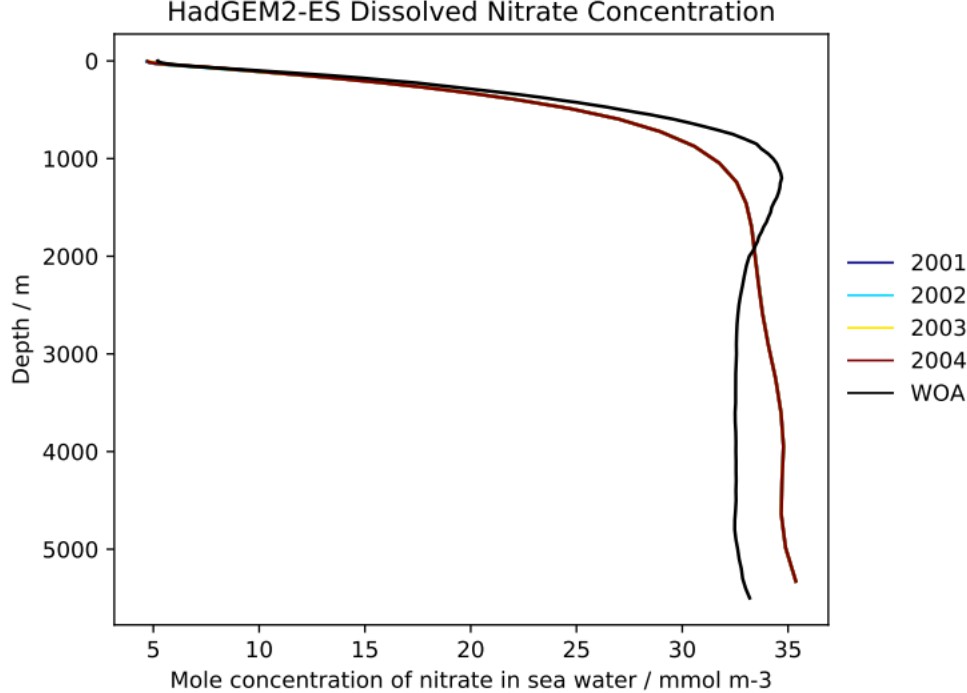

**Figure 40. The global area-weighted average depth profile of the dissolved nitrate concentration in the CMIP5 HadGEM2-ES model and against the World Ocean Atlas 2013. Produced with *recipe_ocean_bgc.yml*, see details in Section 3.5.3.**

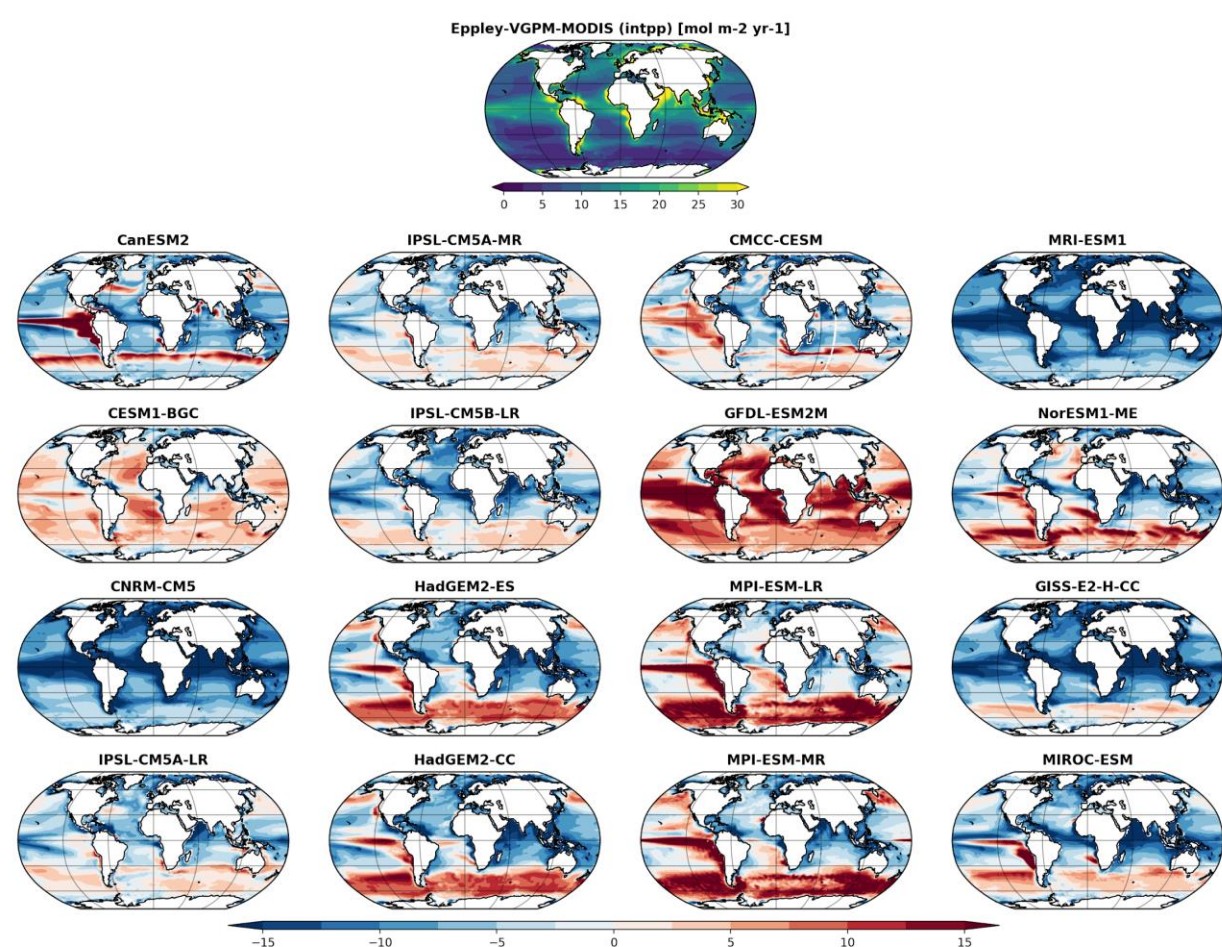

**Figure 41. Global maps of marine Primary Production as carbon (mol m-2 yr-1) estimated from MODIS satellite data using Eppley-VGPM algorithm (Top panel) and differences computed for 16 CMIP5 models data averaged over the period 1995-2004. See Section 3.5.3 for details on *recipe_ocean_bgc.yml*.**


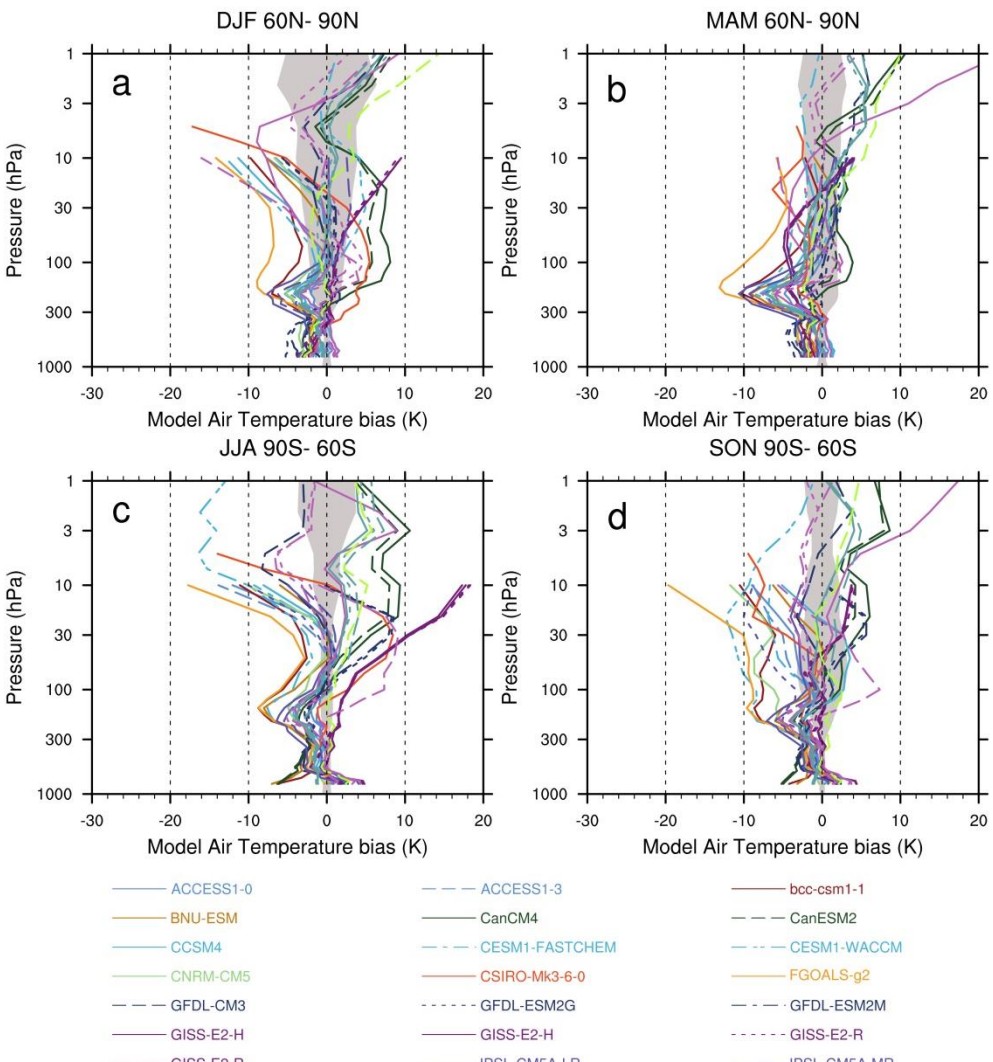

**Figure 42. Climatological mean temperature biases for (top) 60–90N and (bottom) 60–90S for the (left) winter and (right) spring seasons. The climatological means for the CCMs and NCEP data from 1980 to 1999 and for UKMO from 1992 to 2001 are included. Biases are calculated relative to ERA-40 reanalyses. The grey area shows ERA-40 plus and minus 1 standard deviation about the climatological mean. Similar to Figure 1 of Eyring et al. (2006), produced with the *recipe_eyring06jgr.yml*. See details in Section 3.5.4.**


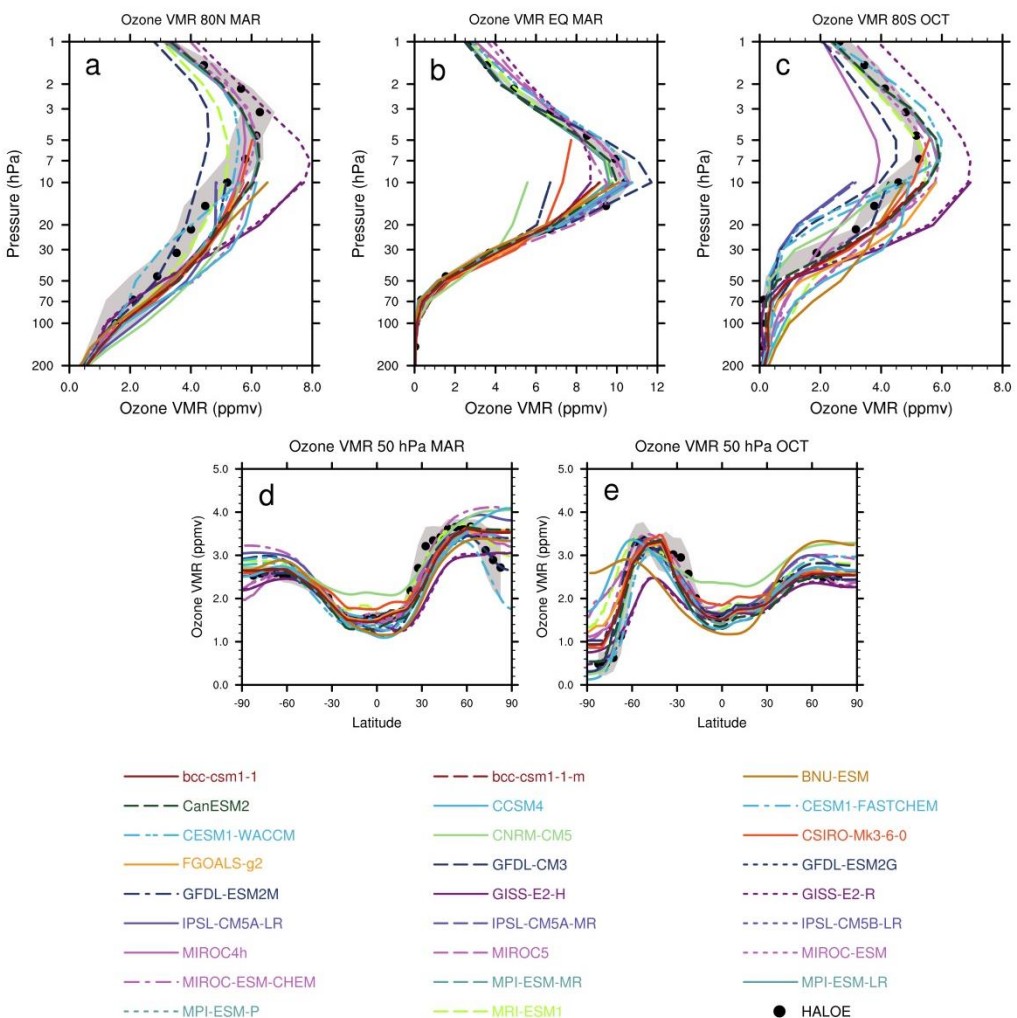


**Figure 43. Climatological zonal mean ozone mixing ratios from the CMIP5 simulations and HALOE in ppmv. Vertical profiles at (a) 80N in March, (b) 0 in March, and (c) 80S in October. Latitudinal profiles at 50 hPa in (d) March and (e) October. The grey area shows HALOE plus and minus 1 standard deviation (s) about the climatological zonal mean. Similar to Figure 5 of Eyring et al. (2006), produced with the *recipe_eyring06jgr.yml*. See**
**details in Section 3.5.4.**

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
