# Peer review of "Earth System Model Evaluation Tool (ESMValTool) v2.0 – extended set of large-scale diagnostics for quasi-operational and comprehensive evaluation of Earth system models in CMIP"

_Geoscientific Model Development, 2019_

## Referee Comment (RC1) · Anonymous Referee #1 · 23 Jan 2020

General Comments

This paper documents the latest version of the ESMValTool v2.0. The paper and the tool have lots of very good points. For example, one can reproduce IPCC Working Group 1 figures fairly easily. As an IPCC author and also a user of the IPCC, this alone makes the tool very valuable to our community. The ability to quickly assess the new CMIP6 models is equally valuable and so on. I strongly support the development of the tool.

[Figure]

That said, I found the paper very hard to read and review. The authors want to discuss each option (what they call recipes) found in the tool. In describing the recipe, it is good to show a figure or 2 to help the reader appreciate what the tool can do. Given the large number of recipes, this makes the paper very long with many figures.

There is a tension between showing the figures as examples and trying to make certain scientific points/conclusion with those figures. I understand this tension well as I have also tried to produce papers like this one. Scientists want to talk about the science, not just use the figure as an example. The problem is the caveats and sometimes the complete justification for the conclusion is left out because the science is not the point. It is an example.

In this paper, there are conclusion sentences in the figure captions and figure caption text in the main body in many, many places. Cleaning these up would greatly help the readability of the paper. As a reader, I strongly dislike conclusions in the figure caption. This is especially true when the conclusion is not restated in the main text. The technique of giving conclusions in the caption is fine in a talk, but not in a paper. It just adds to the length.

Also, many figure captions are missing units and other details needed to document the figure. Sometimes these details are found in the text, but they belong in the caption. Again, this would help the readability of the paper. I highlight some of these in my specific comment section below. As a suggestion to greatly shorten the paper, one could have the nice summary section as section 3 and move all of what is now section 3 to an appendix. I feel this would help most readers and give the authors room to add more overview type of text. If the readers need the details, they can find them in the appendix.

Another general comment is that there needs to be some documentation of the models used in the paper. Shorthand names are given in most cases. Somewhere these names have to be connected to references. Most likely another table is needed.

Finally, it needs to be made clear that most users will not run the tool themselves. The users mainly interact with a browser that displays information which has been all ready computed. This is addressed in section 4 but needs to be made clearer in the abstract, introduction and summary sections. There is a lot of misunderstanding in the community about this tool. Clearing up this aspect will help.

Specific Comments 1. Line 57 – end-to-end provenance – This is jargon. What does this mean? 2. Line 58 – ensure reproducibility – Of what? Need to mention it is reproducibility of the analysis/figures. 3. Line 72 – broad user community - Who do the authors have in mind? Non-specialists would struggle with most of the analysis presented in the paper. 4. Line 77 – 2.1 and 4.7C for a doubling – By when? Is this transient (then a year is needed along with the time averaging period) or equilibrium? 5. Line 105 – to achieve this – To me this sounds like the tool is the solution. It is only part of the solution. Reword. 6. Line 131 – Reference needed for CMOR tables and definitions. I assume the CMIP6 web address is fine. 7. Line 137 – Section 2 needs to mention that the tool helps address a major CMIP focus – to evaluate model performance by comparing models and observations. It should also mention the OBS4MIPS effort here. 8. Line 163-164 – The caption and the text use different words to describe what is shown: error versus deviation. Make them match. The text in the caption (lines 1022-1026) need moved to the text. An important thing to note in the caption is that the colors will change if models are added or removed. (Defining what is meant by "relative"). 9. Lines 170 – 185 – The high correlation for TAS is likely related to mountains and the land-sea mask. Is there anyway to removed these imposed boundary conditions from the analysis? For precipitation, the observations over the ocean are uncertain. Likely more uncertain than the land obs. Is it possible to weight the land more in this analysis? 10. Figure 2, lines 1035-1036 – The sentence that starts "The figure shows both" belongs in the text. 11. Lines 185-201 – What variables were used to make the index? In the caption the phrase that starts "and in this case . . ." (line 1045) belongs in the text. 12. Line 218 – 220 – Figure 4 shows . . . - This is figure caption text. In the figure caption, only the first and last sentences should

remain. The middle sentences belong in the main body text. What are the units? 13. Lines 237 – 243 – Much of this text belongs in the figure caption. The sentence in the caption "Larger biases can be seen" (lines 1062 – 1064) belongs in the text.

\*\*\* Note: I stop commenting on figure caption text in the main body and main body text in the caption below this point. It occurs in most captions. \*\*\*\*

14. Line 253 – peculiar – What is peculiar about these regions? Change to "some". 15. Line 260 – From the caption of figure 8, I have no idea what is plotted in figure 8. What are the units? The main text again describes the figure and belongs in the caption. 16. Line 284, figure 9 – The units for both plots are missing. It is unclear to me what is being plotted. 17. Line 312, figure 10 – Units? The label says %. Percent of what? Occurrence? 18. Line 311 – ZIP – Is this shorthand standard? "Compressed" seems better but less precise. 19. Line 323 – severely impacts – This seems way too strong. It could impact prediction or projections. I have not seen many examples where it does. Change to "could severely impact" or reword. 20. Lines 325 – 365 – I think this discussion could be greatly shortened. Just need to reference Lembo et al. 2019. I do not get much from figure 10. There is a lot of text for little gain. Even the analysis sentence in the figure caption (which belongs in the text) only states a known conclusion. Figure 14 and 15 – what are the units and shadings? 21. Line 385 – Somewhere in this discussion, the point that the observational record in many cases is too short to reliably assess the variability. 22. Line 419 – observed – Change to "reported". 23. Line 421 – unsuitable – This seems too strong. It depends on the questions being asked. 24. Line 410 – Weather regimes – One has to use these with caution. For large climate changes, they may not be useful/reliable. Caveats are needed in this section. 25. Line 436 – Figure 17 – Units? 26. Line 445 – ZIP – See comment #18 above. I have stopped commenting on the use of ZIP. 27. Line 446 – Figure 18 – Units? 28. Line 464 – Figure 19 – Units? 29. Line 492 – strength of northward current – This is incorrect. It is the strength of the overturning circulation – near surface and deep. 30. Line 494 – Figure 22 caption – but it is not clear . . . - This

is a funny statement for this paper. It could be computed using the model spread as an estimate. Also, this phrase belongs in the main body text. 31. Line 561 – Add "of temperature" after "Arctic amplification". 32. Line 570 – Figure 29 caption – A. PHC3 is used in the caption. The label is PHC. Make the labels the same. B. Eurasian Basin – Needs defined in some way. Latitude-longitude? C. Add "in Arctic" after "Eurasian Basin". D. Atlantic water too depth – sentence belongs in text. In the caption, need to say how Atlantic water is defined. 33. Line 577 – Eurasian and Amerasian basins – These basins need defined in some way. 34. Line 589 – 590 – linearly interpolated to climatology levels – Of what? Observations? Need reference. 35. Lines 593 – 597 – Discussion seems to suggest that velocities are interpolated and then transports are computed. If this is the case, this is calculation is wrong. The transport needs to be computed first on the native grid and then interpolated. Doing the velocity first can and will lead to incorrect transports because of issues with the dot product. 36. Lines 600 – 605 – Why use only T to define Atlantic water? It seems like S should be used too. 37. Line 689 – Figure 33 – Relative bias (%) needs better defined. I am not sure what it means. I also do not know what accumulated and averaged bare soil covered area mean. 38. Line 694 – LUC – Defined somewhere? I could not find it. 39. Line 712 – 713 – 5X5 model grid cells – Model grid cells are not 5X5. This is some interpolated grid. 40. Line 728 – Almost only snow-free areas are visible – This makes no sense. 41. Lines 725 – 732 – This result seems suspect to me. It needs checked against models where land cover can change during the integration. Why is the sign of the change the same in both hemispheres given that the data are for July. 42. Line 770 – not well reproduced – This seems too strong given the large observational uncertainty. 43. Line 777 – Figure 36 – The observational uncertainty shading band seems way too narrow. There are many estimates for observed carbon fluxes and they disagree a lot. The internal uncertainty estimates for any given observed data set is typically quite small relative to the disagreement between obs data sets. Therefore the figure and the text are quite misleading. Revise. 44. Line 802 – Figure 39 – Units? 45. Line 807 – but also the spatial distribution . . . - What does this phrase mean? 46. Line 807

– 809 – Figure 40 – Why are there the colored lines on the right side of the figure? The main text refers to a color scale. I do not see a color scale. 47. Line 845 – 858 – Can one access the data plotted in the figures if needed? This would be useful both to IPCC authors and anybody who needs to replot the data. 48. Line 963 - How does the tool handle "bad data"? By bad I mean having the wrong units or the grid is wrong or the data is missing - as examples. How much of the error checking is human and how much is automated? Can other figures be generated if the input data is ok while bad data exists for some other variables? 49. Line 972 – Are the observations available? Saying they are not distributed with the tool does not address this important question.

---

## Referee Comment (RC2) · Anonymous Referee #2 · 24 Jan 2020

General comments:

This manuscript entails a monumental effort in attempting to facilitate the development and evaluation of climate models. Examples for analysis reproducibility, particularly output figures from IPCC chapters is commendable. A pathway to expand this to output figures in the literature is also evident. Model performance metrics, diagnostics for the evaluation of processes in different realms are presented in great detail along with the corresponding recipes. Example figures as a result of integrating community metrics is also shown in the manuscript. The flow and the content could be more con-

[Discussion paper]

sistent so the focus of ESMValTool goals and the impact in doing that is delivered as intended. Some level of brevity, citing references for details, providing more example figures from recipes, pointers to additional recipe documentation – should be made available through an external reference and/or supplementary material. Scalability and interoperability aspects can also be briefly touched upon, providing guidance to the community, making interoperability and practicality– a key to expanding the audience. There is scope for condensing and merging certain sections. Some key points to help improve readability is furnished below in specific comments.

Overall, thank you for the contributions. Please see more comments below.

Specific comments and technical corrections

Page1, Ln 58 Reproducibility - Specifics and explicit wording is required here, as to what aspect is targeted.

Page2, Ln 85

There may be more references that need to be cited while discussing data standardization for CMIP. E.g. https://pcmdi.llnl.gov/mips/cmip5/CMIP5_output_metadata_requirements.pdf ?

Page 3, Ln 92-83: The line about "full rewards of the effort…." should be reworded to provide more of a positive tone to the available observations and model output in standardized format. Expanding what is meant by "full rewards" will be very helpful in this line, rather than the subsequent paragraph.

Page2, Ln 96-97: Please cite or provide links to appropriate references w.r.t data volume estimations for CMIP. Also, what is the database that is being referred to here?

Page 2, Ln 100- I like the addition of "creativity" here.

Page 2, Ln 107- "that provides results…." - Substitute results with something more specific. E.g. analysis products/output?, so it better connects with Ln 108 (This is

realized through..)

Page 3, Ln 115- Does ESMValTool preserve the netCDF metadata (global attributes from input datasets) in output products? How is data provenance established? ( Ln 142 may answer this, please clarify)

Page 3 Ln 118- Consider stating "Figure 1 from their paper, or from Righi et al.. rather than "their Figure 1"

Page 3, Ln 121- The flow from the introduction to companion papers and the present one can be better. Example- Precede the sentence "the use of the tool is demonstrated.." with "In the present paper,.."

Page 3, Ln 124-125, Avoid too many conjunctions (and) here. . "Diagnostics and performance metrics and the variables and observations used".

Page 3, Ln 129: What does "partly also with CMIP3" mean here? Page 3, Ln 130: Is CF-compliance and CMOR-compliance required? Please, also cite CF and CMOR references, expand acronyms. The sentence could be changed to - tool is compatible with any CF and CMOR compliant model output? Please change this as needed so users understand what is ready to be plugged in to ESMValTool, and what requires additional work.

Page 3, Section 2. Ln 131-136. The data descriptions in this section are not satisfactory, especially where the manuscript reads "observation from other sources.." . obs4mip publications should be cited here. It will be nice if the different observation datasets used in recipes can be listed and cited thoroughly. Also, this section could be merged into the final section 6-7 on Code and Data availability.

Page 4, Ln 154- Reproducing IPCC chapter figures is impressive. Are these diagnostics-and-recipes written working directly with the IPCC authors? What is your advice to the IPCC authors to make this effort a success for CMIP6? How are the recipe names constructed- is there a recommended naming convention? How resilient

is ESMValTool to changes like the metadata conventions, DRS, etc from CMIP5 to CMIP6, or say another [inter]national assessment?

Page 4, Ln 160-161: Check and correct line,word spacing.

Page 4, Ln 164: How does one add an alternative observation dataset? One of the companion papers might be addressing this? Page 4, Ln 165: How can additional variables be added? Is it the same as the first version of the tool? Following the citation link here, I still could not get information in two hops. Page 5, Ln 174: Can there be a reference here to the regridding tools used? Why 4x5?

Section 3: Throughout the Overview of recipes, under each sub-section, there can be more consistency. Example: For each recipe, one could ensure these are specified throughout: input (include time-frequency requirement consistently as well), output specifications, source, purpose and significance of the metrics, relevant citations to metrics calculations, summary of the recipe, a sample result. Sticking to this consistently can also condense the text. Suggest just pointing to references like how it was done for CVDP to get more information metrics. [3.3.4] Sea Ice, for instance, can be rewritten to condense text. Are the recipes part of the github repositories? Where can one find them? Though line 145 reads that the intent of the focus of the manuscript is not an assessment of CMIP5 or CMIP6 models, the construction of section 3 is not completely aligned with this. The message needs to be reiterated. If this manuscript is intended to be a documentation paper for the diagnostics and recipes used in ESM-ValTool, the length could be justified to an extent. Otherwise, some sections could be rewritten so focus is retained. Is this manuscript the single source for documentation for all the metrics and recipes? Page 21, Section 4.1. Automatic execution of ESM-ValTool at DKRZ sounds like a nice step to interface with more users. How scalable is this process? Is the idea to expand this to other nodes in ESGF? Is data replication of such huge CMIP6 volumes something that needs to be kept an eye on, leveraging distributed data access protocols or the cloud?

Page 22: Ln 843-844: Section 4 When new plots are created, is there a step that incorporates a basic automated quality assurance conducted? Is there a testing suite for each recipe?

Ln 845: The result browser looks good. Steps to reproducing figures viewed from ESMValTool result browser should be made clearer. This is probably the place where the provenance information captured by ESMValTool will come handy?

How is the performance of running ESMValTool on CMIP data in an automated fashion, and in general from a disconnected sandbox, regardless of ESGF. How is the concept of data versioning incorporated within the automated generation of plots using ESMValTool in ESGF at DKRZ ? When there is bad data retracted on ESGF, and a newer version of data becomes available, what is the current implementation like at the ESMValTool-end or the result-browser to notify its users? If there is no mechanism to notify automatically or not-show-the-corresponding-plots, what is the recommendation to the users? In general, what kind of users does the ESMValTool aim to target?

Ln 857: How does the metadata w.r.t the software version get mapped to the actual source code in GitHub? With data DOIs/data citations widely prevalent for CMIP6, does ESMValTool automatically add data citations to the output figures/files? If not, please provide a pathway to achieve this. Page 24, Ln 943: Please provide an example for "preprocessor settings". Page 24, Ln 948-949. Unable to follow this line "...and tags (i.e. what is reported) ". I think these lines are not adding much value at this point. Page 24, Ln 959. Identifying errors in the simulations early on is a key factor that is penned down as future work here. Even if there are no web-based capabilities, please address if ESMValTool can independenly be installed and run by an individual user at different stages in model running. An idea or vision here to draw more inspiration and motivation for using ESMValTool can be provided.

Ln 955, Again, enhancing quality control is a great use-case, but having ESMValTool run on published data on ESGF does not satisfy this use-case. Stand-alone, this tool

seems to work towards QC. Please clarify.

Section 5. Font size seems to be mixed up in the Summary section - lines till 950 and after 950 are different.

Page 25, Ln 965, Sections 6 and 7 should be condensed into one section. Addressing data citations briefly in Data availability will add more value to the CMIP and ESMVal-Tool efforts.

Comments on Figures:

Adding some of the figures to a supplementary or appendix should be considered. Verify that there is not much redundancy in the text in captions (e.g. Section 3, 5, Figures text). Avoid redundancy where possible.

I find the captions in figures helpful, especially the reference to the corresponding sections. The captions are mostly like IPCC-chapter and documentation paper style. A short caption in bold followed by the description is something that will make the figures stand out.

While specifying OBS in figures, please specify names of OBS in the figures.

Name the variables/fields corresponding to the figures, example Figure 4,5,6 - zonal wind,air temperature,precipitation? respectively.

How are the color palettes picked in general and what flexibility ESMValTool allows w.r.t color palettes?

Expand the acronym QBO in Figure 4, although the section covers it.

Better labeling on the figure itself needed for Fig 15,16 especially.

What is "j" in r1i1j1 in several figure captions - e.g. Fig 22.

In Fig 20, use r1i1p1 to be consistent, not r1p1i1.

Is Figure 26 a reproduction of Fig 9.14 from AR5, Chapter 9. (Including chapter helped

me since there is some ambiguity looking up for Fig 9.14 from AR5).

In Figure 30, "typo" - "ether" vs either;

In Figure 34- typo: predictand, not predictant.

Please use long names on the figure themselves, not the short CMOR names (example Figure 35, Y axis); Units missing in some of the figures, e.g Figure 39.

In the summary section– Given the challenges of CMIP6 (and beyond) and the scientists all over the globe working on multiple research areas, this manuscript should include something along the lines of the role and future of ESMValTool in the community as a whole and how it can be interoperable with overlapping efforts. The ability to cross-function using tools like ESMValTool and making them more inter-operable is a key challenge. The experience from developing ESMValTool in the form of these manuscripts is helpful to the community, and it can also be helpful for the expansion of metrics-and-recipes used in ESMValTool.

---

## Short Comment (SC1) · 28 Jan 2020

The text mentions that Figures 42 and 43 consist of an adaptation of diagnostics included in the CCMValDiag tool. It is very valuable to have recipe_eyring06jgr.yml in v2.0 of ESMValTool (it is not yet the case as of January 2020), as well as a recipe with all diagnostics of Eyring et al, JGR 2013, "Long-term ozone changes and associated climate impacts in CMIP5 simulations" that includes interesting diagnostics with regards to ozone evaluation.

[Figure]

Second, as of January 2020, it is not easy at all to install ESMValTool v2.0, unlike v1.0. For users without conda, that is only recommended in the install web page, the installation with the "pip install ." command, as written in the install web page does not work. Documentation should be amended, in particular for the use of more "dummy" users, if ESMValTool v2.0 is to be used by a wide community. And I think it deserves it.

---

## Author Comment (AC1) · 19 Mar 2020

**Reply to Anonymous Referee #1**

We thank the reviewer for the very detailed and helpful comments. We have now revised our manuscript in light of these and the other review comments we have received. A pointwise reply is given below.

**General comments:**

[Figure]

This paper documents the latest version of the ESMValTool v2.0. The paper and the tool have lots of very good points. For example, one can reproduce IPCC Working Group 1 figures fairly easily. As an IPCC author and also a user of the IPCC, this alone makes the tool very valuable to our community. The ability to quickly assess the new CMIP6 models is equally valuable and so on. I strongly support the development of the tool.

That said, I found the paper very hard to read and review. The authors want to discuss each option (what they call recipes) found in the tool. In describing the recipe, it is good to show a figure or 2 to help the reader appreciate what the tool can do. Given the large number of recipes, this makes the paper very long with many figures. There is a tension between showing the figures as examples and trying to make certain scientific points/conclusion with those figures. I understand this tension well as I have also tried to produce papers like this one. Scientists want to talk about the science, not just use the figure as an example. The problem is the caveats and sometimes the complete justification for the conclusion is left out because the science is not the point. It is an example.

In this paper, there are conclusion sentences in the figure captions and figure caption text in the main body in many, many places. Cleaning these up would greatly help the readability of the paper. As a reader, I strongly dislike conclusions in the figure caption. This is especially true when the conclusion is not restated in the main text. The technique of giving conclusions in the caption is fine in a talk, but not in a paper. It just adds to the length.

Also, many figure captions are missing units and other details needed to document the figure. Sometimes these details are found in the text, but they belong

**in the caption. Again, this would help the readability of the paper. I highlight some of these in my specific comment section below. As a suggestion to greatly shorten the paper, one could have the nice summary section as section 3 and move all of what is now section 3 to an appendix. I feel this would help most readers and give the authors room to add more overview type of text. If the readers need the details, they can find them in the appendix.**

**Another general comment is that there needs to be some documentation of the models used in the paper. Shorthand names are given in most cases. Somewhere these names have to be connected to references. Most likely another table is needed.**

**Finally, it needs to be made clear that most users will not run the tool themselves. The users mainly interact with a browser that displays information which has been all ready computed. This is addressed in section 4 but needs to be made clearer in the abstract, introduction and summary sections. There is a lot of misunderstanding in the community about this tool. Clearing up this aspect will help.**

Thanks for valuing the development of the tool and for your detailed comments! We have followed your suggestion to move the conclusions from the figure captions to the text. We have also revisited all figure captions to include units and other details as requested in the specific comments below. We have however not moved Section 3 to an Appendix, as the structure of the paper is clear and readers who are not interested in the scientific motivation (first paragraph in each subsection) or in a particular recipe can skip this. For the majority of the readers we expect, however, that this is important information. Also from previous discussions with the community we feel it is important to scientifically justify why a certain diagnostic or recipe is included which is why we

prefer keeping the description of diagnostics and the structure of Section 3 as is. The goal of the paper is to document all large-scale diagnostics and recipes that are newly included from v1 to v2 for the community in a peer-reviewed paper. This will allow those interested in a particular diagnostic or recipe to find some more background in this paper. To consider the comment on length, we have therefore not removed individual sections and have not created an Appendix, but we have streamlined the text further, following the specific comments below and by the other reviewers. Following the reviewer's suggestion, we have included another table on the CMIP5 models used in the paper and included the references to the model papers. On the last point, there are different types of users, those who only look at the results provided through the web browser, and those who use the tool for their scientific work. We have made this clearer throughout the text as suggested.

**Specific comments:**

**1. Line 57 – end-to-end provenance – This is jargon. What does this mean?**
End-to-end provenance is the history of an item of data from its creation to its present state. It includes details about the steps that were executed in order to produce the data in its current form. For the ESMValTool this means that it is kept track of which input data have been used and which processing steps have been applied to produce a given product, i.e. typically a netCDF file or a plot. This is explained in more detail in the companion paper Righi et al. (2020), which we now explicitly refer to in the text in the revised paper.

**2. Line 58 – ensure reproducibility – Of what? Need to mention it is reproducibility of the analysis/figures.**
Extended as suggested.

**3.   Line 72 – broad user community - Who do the authors have in mind? Non-specialists would struggle with most of the analysis presented in the paper.**
User community further clarified.

**4.   Line 77 – 2.1 and 4.7C for a doubling – By when?   Is this transient (then a year is needed along with the time averaging period) or equilibrium?**
Clarified.

**5.   Line 105 – to achieve this – To me this sounds like the tool is the solution. It is only part of the solution. Reword.**
Reworded.

**6.   Line 131 – Reference needed for CMOR tables and definitions.   I assume the CMIP6 web address is fine.**
Reference added.

**7.   Line 137 – Section 2 needs to mention that the tool helps address a major CMIP focus – to evaluate model performance by comparing models and observations. It should also mention the OBS4MIPS effort here.**
Good point and added.

**8.   Line 163-164 – The caption and the text use different words to describe what is shown: error versus deviation. Make them match. The text in the caption (lines 1022-1026) need moved to the text.   An important thing to note in the caption is that the colors will change if models are added or removed. (Defining what is meant by "relative").**
Changed as suggested. As said above, the text in the caption has been moved to the text. The term "relative" has been further described in the caption.

**9. Lines 170 – 185 – The high correlation for TAS is likely related to mountains and the land-sea mask. Is there anyway to removed these imposed boundary conditions from the analysis? For precipitation, the observations over the ocean are uncertain. Likely more uncertain than the land obs. Is it possible to weight the land more in this analysis?**

This diagnostic resembles the analysis of Gleckler et al. (2008) as also used in IPCC AR5. In their analysis, no weighting of specific regions or geographic features has been applied. Theoretically, this could be done as an extension of the portrait diagram of Gleckler et al. (2008). Since this diagram is rather intended as a summary providing an overview of the models' performance as a starting point for more detailed analyses, such weighting is (in our opinion) better suited for additional (other) diagnostics.

**10. Figure 2, lines 1035-1036 – The sentence that starts "The figure shows both" belongs in the text.**

Sentence moved as suggested.

**11. Lines 185-201 – What variables were used to make the index? In the caption the phrase that starts "and in this case . . ." (line 1045) belongs in the text.**

Variables clarified and sentence moved as suggested.

**12. Line 218 – 220 – Figure 4 shows . . . - This is figure caption text. In the figure caption, only the first and last sentences should remain. The middle sentences belong in the main body text. What are the units?**

Changed as suggested.

**13. Lines 237 – 243 – Much of this text belongs in the figure caption. The sentence in the caption "Larger biases can be seen" (lines 1062 – 1064) belongs**

**in the text. *** Note: I stop commenting on figure caption text in the main body and main body text in the caption below this point. It occurs in most captions. ******

Changed as suggested.

**14. Line 253 – peculiar – What is peculiar about these regions? Change to "some".**
Changed as suggested.

**15. Line 260 – From the caption of figure 8, I have no idea what is plotted in figure 8. What are the units? The main text again describes the figure and belongs in the caption.**
The text provides a definition and a reference for the quantile bias. The quantile bias, being a ratio, is unitless, we added this in the figure caption. We also changed the figure caption to clarify that the discussion in the caption is an example of how such a figure could be used.

**16. Line 284, figure 9 – The units for both plots are missing. It is unclear to me what is being plotted.**
Units clarified and caption improved.

**17. Line 312, figure 10 – Units? The label says percent. Percent of what? Occurrence?**
Blocking events frequency measures the percentage of blocked days in the 1979-2008 period (considering only the winter period DJF). We modified the caption of fig. 10 to specify this.

**18. Line 311 – ZIP – Is this shorthand standard? "Compressed" seems better but less precise.**

Indeed *zip* is standard compression format. We added "compressed" in the text.

**19. Line 323 – severely impacts – This seems way too strong. It could impact prediction or projections. I have not seen many examples where it does. Change to "could severely impact" or reword.**
Changed as suggested.

**20. Lines 325 – 365 – I think this discussion could be greatly shortened. Just need to reference Lembo et al. 2019. I do not get much from figure 10. There is a lot of text for little gain. Even the analysis sentence in the figure caption (which belongs in the text) only states a known conclusion. Figure 14 and 15 – what are the units and shadings?**
Discussion shortened as suggested and units clarified.

**21. Line 385 – Somewhere in this discussion, the point that the observational record in many cases is too short to reliably assess the variability.**
Added.

**22. Line 419 – observed – Change to "reported".**
Changed as suggested.

**23. Line 421 – unsuitable – This seems too strong. It depends on the questions being asked.**
Agreed, we changed the text to say "Forecast systems …. may have difficulties in reproducing climate variability and its long-term changes….".

**24. Line 410 – Weather regimes – One has to use these with caution. For large climate changes, they may not be useful/reliable. Caveats are needed in this section.**

Yes, indeed the tool should be used with caution when applied (for example) to future scenario simulation where large climate changes are implied. We added a sentence to point this out and to explain how one could proceed in this case.

**25. Line 436 – Figure 17 – Units?**
Units clarified.

**26. Line 445 – ZIP – See comment no 18 above. I have stopped commenting on the use of ZIP.**
Changed, see also response above.

**27. Line 446 – Figure 18 – Units?**
The units are already specified close to the colorbar, it is meters [m] (written rotated along the colorbar).

**28. Line 464 – Figure 19 – Units?**
Units clarified.

**29. Line 492 – strength of northward current – This is incorrect. It is the strength of the overturning circulation – near surface and deep.**
Changed as suggested.

**30. Line 494 – Figure 22 caption – but it is not clear . . . - This is a funny statement for this paper. It could be computed using the model spread as an estimate. Also, this phrase belongs in the main body text.**
Moved to the main body of the text and sentence rephrased as suggested.

**31. Line 561 – Add "of temperature" after "Arctic amplification".**
Changed to "... processes are Arctic atmospheric temperature warming amplification

...".

**32. Line 570 – Figure 29 caption – A. PHC3 is used in the caption. The label is PHC. Make the labels the same. B. Eurasian Basin – Needs defined in some way. Latitude-longitude? C. Add "in Arctic" after "Eurasian Basin". D. Atlantic water too depth – sentence belongs in text. In the caption, need to say how Atlantic water is defined.**

A. Changed as suggested.

B., C. Changed to "Eurasian Basin of the Arctic Ocean (as defined in Holloway et al., 2007)"

D. We have removed the sentence from the figure captions, following the reviewer's general suggestion above.

**33. Line 577 – Eurasian and Amerasian basins – These basins need defined in some way.**

Added "(as defined in Holloway et al., 2007)".

**34. Line 589 – 590 – linearly interpolated to climatology levels – Of what? Observations? Need reference.**

Clarified and reference added.

**35. Lines 593 – 597 – Discussion seems to suggest that velocities are interpolated and then transports are computed. If this is the case, this is calculation is wrong. The transport needs to be computed first on the native grid and then interpolated. Doing the velocity first can and will lead to incorrect transports because of issues with the dot product.**

We do not compute transports, only transects of scalar properties. Although it is clear that the confusion comes from the mention of the exchange between basins. We removed this part of the sentence and add explicitly that only temperature and salinity

are considered. The new sentence reads as follows: "For each point, a vertical profile of temperature or salinity on the original model levels is interpolated."

**36. Lines 600 – 605 – Why use only T to define Atlantic water? It seems like S should be used too.**
Usually, in the literature salinity is not used when Atlantic Water of the Arctic Ocean is defined.

**37. Line 689 – Figure 33 – Relative bias (percent) needs better defined. I am not sure what it means. I also do not know what accumulated and averaged bare soil covered area mean.**
The relative bias is computed as the difference between simulated and observed land cover area divided by the observed area. It is further converted into percentage (multiplying by 100). The accumulated land cover area is the sum of the surface area covered with a specific land cover type (here bare soil) in a given hemisphere or region, while the average covered fractions gives the ratio of the accumulated area to the total area of the hemisphere or region and, again, is converted to percentage.

**38. Line 694 – LUC – Defined somewhere? I could not find it.**
This was a typo, thanks for spotting. Corrected to LCC.

**39. Line 712 – 713 – 5X5 model grid cells – Model grid cells are not 5X5. This is some interpolated grid.**
The reviewer is right that the provided figure had been produced with an algorithm applied on an interpolated grid. However, the diagnostic also provides the option to run the algorithm on the native grid. For the new version of the manuscript, we now provide the figure obtained by running the algorithm on the native grid of the MPI-ESM-LR model. This figure version exhibits very small differences compared to the previous one.

[Figure]

**40. Line 728 – Almost only snow-free areas are visible – This makes no sense.**

We now provide more background to support this outcome: during the month of July, there is a very limited number of grid cells where the snow area fraction exceeds 0.9 (the criterion for them to be considered snow-covered). Moreover, in order to ensure that the reconstructed signal is of good quality, some more criteria need to be fulfilled for a grid cell to be included in the regression algorithm: at least 15 grid cells – either snow-free or snow-covered – need to be present in a moving window, and the sum of the considered land cover types has to equal at least 90% of the grid cell. This explains why no values are available for snow-covered areas in July for the MPI-ESM-LR model. A full description of the methodology has been included in the following paper submitted to Earth System Dynamics: Lejeune et al., Biases in the albedo sensitivity to deforestation in CMIP5 models and their impacts on the associated historical Radiative Forcing. This recently submitted paper is now referred to in the new version of the manuscript. Consistently with this new background information, we have revised the formulation for this part of the manuscript: "It can calculate albedo estimates for each of these two cases and each of the three land cover types, given that some criteria are fulfilled: the regressions are only conducted in the big boxes with a minimum number of 15 grid cells (either snow-free or snow-covered), taking only into account the grid cells where the sum of the area fractions occupied by the three considered land cover types exceeds 90%. The algorithm eventually plots global maps of the albedo changes associated with..."

**41. Lines 725 – 732 – This result seems suspect to me. It needs checked against models where land cover can change during the integration. Why is the sign of the change the same in both hemispheres given that the data are for July.**

Our algorithm provides the potential albedo change associated to a transition between

two land cover types. In the case of Fig. 34, it is the change associated to a transition from trees to crops/grasses, which is positive in all seasons in most areas. This is in line with satellite-derived estimates of this albedo change for July, as shown by the right-hand side of Fig. 34. To make this clearer, we added the word "potential" at the beginning of the legend of Fig. 34.

**42. Line 770 – not well reproduced – This seems too strong given the large observational uncertainty.**
The text has been revised to clarify the statement: "This emergent ecosystem property, calculated, for example, as a ratio of long-term average total carbon stock to gross primary productivity, has been extensively used to evaluate ESM simulations (Todd-Brown et al., 2013, Carvalhais et al., 2014; Koven et al., 2015, Koven et al., 2017). Despite the large range of observational uncertainties and sources, ESM simulations consistently exhibit a robust correlation with the observation ensembles, but with a substantial underestimation bias."

**43. Line 777 – Figure 36 – The observational uncertainty shading band seems way too narrow. There are many estimates for observed carbon fluxes and they disagree a lot. The internal uncertainty estimates for any given observed data set is typically quite small relative to the disagreement between obs data sets. Therefore the figure and the text are quite misleading. Revise.**
The figure and the observational data used for the figure have been updated so that it is using exactly the same as the original study of Carvalhais et al., 2014. The reviewer is correct that the observational uncertainty in the submitted version of the manuscript was way too narrow. We have updated the figure. In the updated figure as well, there is a clear difference between the models and observation, and most models are outside the observation range. We have revised the text for possible uncertainties with observation-based estimates. The text has been changed to: "Most CMIP5 models (and multi-model ensemble) have a much shorter turnover time than the observationbased estimate across the whole latitudinal range. Even though different estimates of observation-based carbon fluxes and stocks can vary significantly, a recent study (Fan et al., 2020; Figure 5a) shows that the zonal distributions of observation-based estimates of turnover time is robust against the differences in observations."

**44. Line 802 – Figure 39 – Units?**
The figure has been updated. Note that this figure now uses a linear scale on the x and y axis.

**45. Line 807 – but also the spatial distribution . . . - What does this phrase mean?**
The spatial distribution is the distribution of the values of data in space (a map). To clarify, the text has been changed to "When viewed together, Figures 38 and 39 show the biases between the model and the observations in the surface layer relative to each other, both in terms of their spatially-independent distribution in fig. 38 and their spatially-dependent distribution in figure 39."

**46. Line 807 – 809 – Figure 40 – Why are there the colored lines on the right side of the figure? The main text refers to a color scale. I do not see a color scale.**
For clarity, this has been changed to: "Figure 40 shows the global average depth profile of the dissolved nitrate concentration in the CMIP5 HadGEM2-ES model and against the World Ocean Atlas dataset. The colour scale indicates the annual average, although in this specific case there is little observed inter-annual variability so the annual averages are closely overlaid. Nevertheless, this class of figure can be useful to evaluate biases between model and observations over the entire depth profile of the ocean and can also be used to identify long term changes in the vertical structure of the ocean models."

**47. Line 845 – 858 – Can one access the data plotted in the figures if needed? This would be useful both to IPCC authors and anybody who needs to replot the data.**

Yes, this is possible, as every diagnostic in the ESMValTool generates one (or more) netCDF file(s) containing the data resulting from the analysis and used for plotting.

**48. Line 963 - How does the tool handle "bad data"? By bad I mean having the wrong units or the grid is wrong or the data is missing - as examples. How much of the error checking is human and how much is automated? Can other figures be generated if the input data is ok while bad data exists for some other variables?**

Checking for errors in the input data is mostly automated in the CMOR checking module of the ESMValCore preprocessor (see Righi et al., 2020 for details). This module checks for CMOR compliance of the input data and helps to identify common errors such as inconsistent units, wrong coordinates, bad missing values, etc. Errors in the actual data are however hard to detect in an automated way.

**49. Line 972 – Are the observations available? Saying they are not distributed with the tool does not address this important question.**

Due to license issues, redistribution of observational data is not allowed in most cases, but the tool provides CMORizing scripts for each observational dataset used in the recipes that is not an obs4MIPs dataset. These scripts include detailed information on how to download and process the data for usage with the tool (see Righi et al. (2020) for details).

**References**

Righi, M., Andela, B., Eyring, V., Lauer, A., Predoi, V., Schlund, M., Vegas-Regidor, J., Bock, L., Brötz, B., de Mora, L., Diblen, F., Dreyer, L., Drost, N., Earnshaw, P., Hassler,

B., Koldunov, N., Little, B., Loosveldt Tomas, S., and Zimmermann, K.: Earth System Model Evaluation Tool (ESMValTool) v2.0 – technical overview, Geosci. Model Dev., 13, 1179–1199, https://doi.org/10.5194/gmd-13-1179-2020, 2020.

---

## Author Comment (AC2) · 19 Mar 2020

**Reply to Anonymous Referee #2**

We thank the reviewer for the very detailed and helpful comments. We have now revised our manuscript in light of these and the other review comments we have received. A pointwise reply is given below.

**General comments:**

**This manuscript entails a monumental effort in attempting to facilitate the development and evaluation of climate models. Examples for analysis reproducibility, particularly output figures from IPCC chapters is commendable. A pathway to expand this to output figures in the literature is also evident. Model performance metrics, diagnostics for the evaluation of processes in different realms are presented in great detail along with the corresponding recipes. Example figures as a result of integrating community metrics is also shown in the manuscript. The flow and the content could be more consistent so the focus of ESMValTool goals and the impact in doing that is delivered as intended. Some level of brevity, citing references for details, providing more example figures from recipes, pointers to additional recipe documentation – should be made available through an external reference and/or supplementary material. Scalability and interoperability aspects can also be briefly touched upon, providing guidance to the community, making interoperability and practicality– a key to expanding the audience. There is scope for condensing and merging certain sections. Some key points to help improve readability is furnished below in specific comments. Overall, thank you for the contributions. Please see more comments below.**

Thanks for your suggestions. Please see our responses to the specific comments below how we have addressed them.

**Specific comments and technical corrections:**

**Page1, Ln 58 Reproducibility - Specifics and explicit wording is required here, as to what aspect is targeted.**

We have rephrased accordingly.

**Page2, Ln 85 There may be more references that need to be cited while discussing data standardization for CMIP.**
**E.g. https://pcmdi.llnl.gov/mips/cmip5/CMIP5_output_metadata_requirements.pdf?**
More references added.

**Page 3, Ln 92-83: The line about "full rewards of the effort. . .." should be reworded to provide more of a positive tone to the available observations and model output in standardized format. Expanding what is meant by "full rewards" will be very helpful in this line, rather than the subsequent paragraph.**
The paragraph has been reworded.

**Page2, Ln 96-97: Please cite or provide links to appropriate references w.r.t data volume estimations for CMIP. Also, what is the database that is being referred to here?**
References added and the database clarified.

**Page 2, Ln 100- I like the addition of "creativity" here.**
Thanks.

**Page 2, Ln 107- "that provides results. . .." - Substitute results with something more specific. E.g. analysis products/output?, so it better connects with Ln 108 (This is realized through..)**
Changed as suggested.

**Page 3, Ln 115- Does ESMValTool preserve the netCDF metadata (global attributes from input datasets) in output products? How is data provenance established? ( Ln 142 may answer this, please clarify)**

The ESMValTool does preserve the netCDF metadata including the global attributes. These metadata are also written to the products (netCDF and plots) using W3C-PROV (Python package prov v1.5.3). Details can be found in the technical ESMValTool description paper Righi et al. (2020 that we now explicitly refer to here).

**Page 3 Ln 118- Consider stating "Figure 1 from their paper, or from Righi et al.. rather than "their Figure 1"**
Changed as suggested.

**Page 3, Ln 121- The flow from the introduction to companion papers and the present one can be better. Example- Precede the sentence "the use of the tool is demonstrated.." with "In the present paper,.."**
Changed as suggested.

**Page 3, Ln 124-125, Avoid too many conjunctions (and) here. . "Diagnostics and performance metrics and the variables and observations used".**
Sentence rewritten for clarity.

**Page 3, Ln 129: What does "partly also with CMIP3" mean here?**
Partly refers to the fact that not all diagnostics can be run with CMIP3 data because for some diagnostics not all required variables are included in the CMIP3 data request.

**Page 3, Ln 130: Is CF-compliance and CMOR-compliance required? Please, also cite CF and CMOR references, expand acronyms. The sentence could be changed to - tool is compatible with any CF and CMOR compliant model output? Please change this as needed so users understand what is ready to be plugged in to ESMValTool, and what requires additional work.**
The ESMValTool requires that input data are following the CMOR standard. CMOR is based on the CF conventions but defines some additional metadata on top of it.

For details, see Righi et al. (2020). CF and CMOR references are now added in the revised manuscript and abbreviations are defined.

**Page 3, Section 2. Ln 131-136. The data descriptions in this section are not satisfactory, especially where the manuscript reads "observation from other sources.." . obs4mip publications should be cited here. It will be nice if ty the different observation datasets used in recipes can be listed and cited thoroughly. Also, this section could be merged into the final section 6-7 on Code and Data availability.**
We now refer to Table 3 in Righi et al. (2020), which contains a list of observational data from external sources', i.e. observations that are not from obs4MIPs.

**Page 4, Ln 154-**

- **Reproducing IPCC chapter figures is impressive. Are these diagnostics-and-recipes written working directly with the IPCC authors? What is your advice to the IPCC authors to make this effort a success for CMIP6?**
  Thanks! For the IPCC AR5 Chapter 9 diagnostics and recipes, they were written by the ESMValTool development team after the publication of the AR5. For AR6 Working Group I, several chapters are using the ESMValTool to produce their figures. The diagnostics and recipes are then written by the chapter scientists or the lead or contributing authors of the chapter, with support from the ESMValTool core development team. Our advice to the authors of the IPCC AR6 is to write a recipe for each chapter, so that figures can be reproduced any time. This would be a huge present to those involved in a possible AR7. It would enable a direct and prompt comparison to new model generations.

- **How are the recipe names constructed- is there a recommended naming convention?**
  For recipes reproducing the analysis of a refereed publication, we use [firstauthor] [year] [journal-abbreviation], for example "recipe_lauer13jclim.yml". The same applies to IPCC chapters, e.g. "recipe_flato13ipcc.yml". For other recipes there is no strict rule, but the authors are advised to give appropriate names reflecting the recipe content, e.g. "recipe_heatwaves_coldwaves.yml"

- **How resilient is ESMValTool to changes like the metadata conventions, DRS, etc from CMIP5 to CMIP6, or say another [inter]national assessment?**
  The ESMValTool has been developed in a flexible way allowing defining the DRS structure via a configuration file. Metadata conventions are imported from the obs4mips/CMIP3/CMIP5/CMIP6 tables and can be easily extended with new tables.

**Page 4, Ln 160-161: Check and correct line,word spacing.**
Corrected.

**Page 4, Ln 164: How does one add an alternative observation dataset? One of the companion papers might be addressing this? Page 4, Ln 165: How can additional variables be added? Is it the same as the first version of the tool? Following the citation link here, I still could not get information in two hops.**
Alternative observational datasets are specified in the "recipe" (if supported by the diagnostic). Additional variables can be added by custom CMOR tables similarly to ESMValTool v1.0. In case of derived variables, Python scripts have to be provided to do the actual calculations. This has changed compared to version 1.0 in which these variable derivation scripts were written in NCL. We refer to the extensive ESMValTool documentation for more details: https://esmvaltool.readthedocs.io/en/latest.

**Page 5, Ln 174: Can there be a reference here to the regridding tools used? Why 4x5?**
Regridding is done by the Python Iris package, which offers different regridding

schemes. We have added a reference to the Iris user's guide at (https://scitools.org.uk/iris/docs/latest/index.html#). The grid resolution of 4x5 degrees has been chosen to be as consistent as possible with the equivalent diagnostic used in IPCC AR5.

**Section 3:**

- **Throughout the Overview of recipes, under each sub-section, there can be more consistency. Example: For each recipe, one could ensure these are specified throughout: input (include time-frequency requirement consistently as well), output specifications, source, purpose and significance of the metrics, relevant citations to metrics calculations, summary of the recipe, a sample result. Sticking to this consistently can also condense the text.**
  We have already defined a common structure how to discuss each of the recipes in each of the subsections. This is already described in the manuscript: "In each subsection, we first scientifically motivate the inclusion of the recipe by reviewing the main systematic biases in current ESMs and their importance and implications. We then give an overview of the recipes that can be used to evaluate such biases along with the diagnostics and performance metrics included, and the required variables and corresponding observations that are used in ESMValTool v2.0. For each recipe, we provide 1-2 example figures that are applied to either all or a subset of the CMIP5 models."

- **Suggest just pointing to references like how it was done for CVDP to get more information metrics.**
  We already include many references in the submitted version of the manuscript. For CVDP this is more straight forward than for other recipes, as an externally developed tool exists.

- **[3.3.4] Sea Ice, for instance, can be rewritten to condense text.**
  Changed as suggested and sea ice section shortened.

- **Are the recipes part of the github repositories? Where can one find them?**
  All recipes are included in the ESMValTool repository on GitHub. The directory structure of the ESMValTool is outlined in the technical description paper by Righi et al. (2020). All recipes can be found in the directory *https://github.com/ESMValGroup/ESMValTool/tree/master/esmvaltool/recipes.*

- **Though line 145 reads that the intent of the focus of the manuscript is not an assessment of CMIP5 or CMIP6 models, the construction of section 3 is not completely aligned with this. The message needs to be reiterated. If this manuscript is intended to be a documentation paper for the diagnostics and recipes used in ESMValTool, the length could be justified to an extent. Otherwise, some sections could be rewritten so focus is retained.**
  The goal of the manuscript is indeed to document the diagnostics and recipes available in the ESMValTool. Assessing the CMIP models is indeed not the scope of the paper, nevertheless Section 3 presents a few examples to show how the ESMValTool output could support the scientific interpretation. We would like to stick to this structure as it turned out to be useful for users and developers, but we have followed the reviewer 1's comment to move the sentence on the results from the caption to the text which hopefully also addresses this point. We have also further shortened the paper by following specific comments from both reviewers.

- **Is this manuscript the single source for documentation for all the metrics and recipes?**
  No, additional documentation will be provided in the companion papers Lauer et al. (description of diagnostics for emergent constraints and future projections from Earth system models in CMIP) and Weigel et al. (description of diagnostics for extreme events, regional model and impact evaluation and analysis of Earth

system models in CMIP). The manuscript refers to these additional papers.

- **Page 21, Section 4.1. Automatic execution of ESMValTool at DKRZ sounds like a nice step to interface with more users. How scalable is this process? Is the idea to expand this to other nodes in ESGF? Is data replication of such huge CMIP6 volumes something that needs to be kept an eye on, leveraging distributed data access protocols or the cloud?**
  The automatic execution of the ESMValTool has been tested only at DKRZ so far, but thanks to the tool's flexibility it can be easily ported to other ESGF nodes. Data replication is certainly an issue, as discussed in detail in Eyring et al. (ESSD, 2016).

**Page 22: Ln 843-844: Section 4**

- **When new plots are created, is there a step that incorporates a basic automated quality assurance conducted?**
  The ESMValTool includes checks for data availability and CMOR compliance of all input datasets which provides some first basic quality control. However, at the moment the output of diagnostics (such as plots) has to be checked manually by a scientist to identify anything beyond missing or badly formatted data.

- **Is there a testing suite for each recipe?**
  We have chosen not to ask scientists to implement unit tests for their recipes, because this would make the threshold to contribute diagnostics to ESMValTool too high for many scientists. ESMValCore, the part of the tool that runs the diagnostic scripts and does data quality checks, pre-processing, and records provenance, is rigorously tested with unit tests, to ensure that the tool can be used reliably. In addition to that, we are working on setting up automated tests that regularly run all recipes on a server at DKRZ, both during the development of new recipes

as well as after incorporation into ESMValTool, to ensure that recipes are working and produce consistent output. At the moment this is still a manual process, but we hope to make progress on this in the next few months, as the hardware becomes available.

- **Ln 845: The result browser looks good. Steps to reproducing figures viewed from ESMValTool result browser should be made clearer. This is probably the place where the provenance information captured by ESMVal-Tool will come handy?**
  Yes, this is the typical target application of the provenance system, as detailed in the technical overview paper (Righi et al., 2020). The steps to produce the figures in the ESMValTool result browser are now further detailed in the revised manuscript.

- **How is the performance of running ESMValTool on CMIP data in an automated fashion, and in general from a disconnected sandbox, regardless of ESGF.**
  The ESMValTool performance has little to do with the automatization. Again we refer to the technical overview for more details on the tool's performance.

- **How is the concept of data versioning incorporated within the automated generation of plots using ESMValTool in ESGF at DKRZ ? When there is bad data retracted on ESGF, and a newer version of data becomes available, what is the current implementation like at the ESMValTool-end or the result-browser to notify its users? If there is no mechanism to notify automatically or not-show-the-corresponding-plots, what is the recommendation to the users? In general, what kind of users does the ESMValTool aim to target?**
  The figures on the result browser can be sorted by recipe used to produce this figure. By clicking on the figure, also all input datasets can be listed. This

information should be enough as a starting point for reproducing a given figure for anyone familiar with running the ESMValTool. If specific dataset versions are crucial, the whole metadata provided with the plots have to be read and taken into account. By default, the ESMValTool always uses the latest version of a dataset available. The ESMValTool and the result-browser are not capable of notifying users of any changes. Recommendation to any user is therefore to check the result-browser frequently and/or get into contact with the ESMValTool team as stated on the webpage.

**Ln 857: How does the metadata w.r.t the software version get mapped to the actual source code in GitHub? With data DOIs/data citations widely prevalent for CMIP6, does ESMValTool automatically add data citations to the output figures/files? If not, please provide a pathway to achieve this. Page 24, Ln 943: Please provide an example for "preprocessor settings". Page 24, Ln 948-949. Unable to follow this line "...and tags (i.e. what is reported) ". I think these lines are not adding much value at this point.**

The exact version number of the tool used to produce a plot and written to the metadata corresponds to the release tag on GitHub, e.g. "2.0.0b6". Data citations are only added for observational data, while for model data all metadata are preserved. This typically includes the "tracking id" that can be used to exactly identify the dataset. The preprocessor settings are discussed in detail in the technical overview paper by Righi et al. (2020). We therefore do not see any use of repeating these here.

**Page 24, Ln 959. Identifying errors in the simulations early on is a key factor that is penned down as future work here. Even if there are no web-based capabilities, please address if ESMValTool can independently be installed and run by an individual user at different stages in model running. An idea or vision here to draw more inspiration and motivation for using ESMValTool can be**

**provided.**
The ESMValTool can be installed within the user space, so each user can install and develop the tool independently.

**Ln 955, Again, enhancing quality control is a great use-case, but having ESMValTool run on published data on ESGF does not satisfy this use-case. Stand-alone, this tool seems to work towards QC. Please clarify.**
Analyzing first CMIP data published on the ESGF showed that there are still many errors in the metadata and/or the actual data. Examples of such errors include, for instance, wrong units, wrong coordinates (e.g. time) and entire fields consisting of missing values. We therefore do think that running the ESMValTool on these data can be regarded as a quality control process, admittedly on a higher level than the initial quality control done by the modeling groups.

**Section 5. Font size seems to be mixed up in the Summary section - lines till 950 and after 950 are different.**
Corrected.

**Page 25, Ln 965, Sections 6 and 7 should be condensed into one section. Addressing data citations briefly in Data availability will add more value to the CMIP and ESMValTool efforts.**
We have removed the data availability section.

**Comments on Figures:**

**Adding some of the figures to a supplementary or appendix should be considered.**

Since we have one example figure for each of the recipes, we would like to keep them in the main manuscript. This also supports the general structure the reviewer is calling for.

**Verify that there is not much redundancy in the text in captions (e.g. Section 3, 5, Figures text). Avoid redundancy where possible.**
This has been cleaned up, see also our responses to Reviewer 1.

**I find the captions in figures helpful, especially the reference to the corresponding sections. The captions are mostly like IPCC-chapter and documentation paper style. A short caption in bold followed by the description is something that will make the figures stand out.**
Using bold font in the figure caption is unfortunately not supported by the journal standards.

**While specifying OBS in figures, please specify names of OBS in the figures.**
Changed as suggested.

**Name the variables/fields corresponding to the figures, example Figure 4,5,6 - zonal wind,air temperature,precipitation? respectively.**
Changed as suggested.

**How are the color palettes picked in general and what flexibility ESMVal-Tool allows w.r.t color palettes?**
Default color palettes are picked by the diagnostic authors. Some diagnostics allow for setting it as an option. Implementing such a feature, however, is also up to the diagnostic authors and therefore not supported by all diagnostics.

**Expand the acronym QBO in Figure 4, although the section covers it.**
Done.

**Better labeling on the figure itself needed for Fig 15,16 especially.**
Done.

**What is "j" in r1i1j1 in several figure captions - e.g. Fig 22.**
This was typo, thanks for spotting! Changed to r1i1p1, see https://portal.enes.org/data/enes-model-data/cmip5/datastructure

**In Fig 20, use r1i1p1 to be consistent, not r1p1i1.**
Changed as suggested.

**Is Figure 26 a reproduction of Fig 9.14 from AR5, Chapter 9. (Including chapter helped me since there is some ambiguity looking up for Fig 9.14 from AR5).**
Yes it is reproducing Fig 9.14 from Chapter 9 of AR5. We have added "Chapter 9 of" before "AR5" for clarity.

**In Figure 30, "typo" - "ether" vs either;**
Corrected

**In Figure 34- typo: predictand, not predictant. Please use long names on the figure themselves, not the short CMOR names (example Figure 35, Y axis); Units missing in some of the figures, e.g Figure 39.**
Corrected.

**In the summary section– Given the challenges of CMIP6 (and beyond) and the scientists all over the globe working on multiple research areas, this**

**manuscript should include something along the lines of the role and future of ESMValTool in the community as a whole and how it can be interoperable with overlapping efforts. The ability to cross-function using tools like ESMValTool and making them more inter-operable is a key challenge. The experience from developing ESMValTool in the form of these manuscripts is helpful to the community, and it can also be helpful for the expansion of metrics-and-recipes used in ESMValTool.**

Briefly expanded on this point. However, since inter-operability with other tools requires discussion and agreement with the developers of the other tools who are not authors on this paper, we cannot define this in detail here. There are also license issues to be considered. We are however very open for this collaboration and have put a lot of effort into actively seeking coordination with other tool developers in the past years. Having different and complementary tools might however well be desirable, see also Eyring et al. (2019).

**References**

Eyring, V., Gleckler, P. J., Heinze, C., Stouffer, R. J., Taylor, K. E., Balaji, V., Guilyardi, E., Joussaume, S., Kindermann, S., Lawrence, B. N., Meehl, G. A., Righi, M., and Williams, D. N.: Towards improved and more routine Earth system model evaluation in CMIP, Earth Syst. Dynam., 7, 813-830, doi:10.5194/esd-7-813-2016, 2016.

Eyring, V., Cox, P. M., Flato, G. M., Gleckler, P. J., Abramowitz, G., Caldwell, P., Collins, W. D., Gier, B. K., Hall, A. D., Hoffman, F. M., Hurtt, G. C., Jahn, A., Jones, C. D., Klein, S. A., Krasting, J. P., Kwiatkowski, L., Lorenz, R., Maloney, E., Meehl, G. A., Pendergrass, A. G., Pincus, R., Ruane, A. C., Russell, J. L., Sanderson, B. M., Santer, B. D., Sherwood, S. C., Simpson, I. R., Stouffer, R. J., and Williamson, M. S., Taking climate model evaluation to the next level. Nature Climate Change.

doi:10.1038/s41558-018-0355-y, 2019.

Lauer, A., Eyring, V., Bellprat, O., Bock, L., Gier, B. K., Hunter, A., Lorenz, R., Pérez-Zanón, N., Righi, M., Schlund, M., Senftleben, D., Weigel, K., and Zechlau, S.: Earth System Model Evaluation Tool (ESMValTool) v2.0 – diagnostics for emergent constraints and future projections from Earth system models in CMIP, Geosci. Model Dev. Discuss., https://doi.org/10.5194/gmd-2020-60, in review, 2020.

Righi, M., Andela, B., Eyring, V., Lauer, A., Predoi, V., Schlund, M., Vegas-Regidor, J., Bock, L., Brötz, B., de Mora, L., Diblen, F., Dreyer, L., Drost, N., Earnshaw, P., Hassler, B., Koldunov, N., Little, B., Loosveldt Tomas, S., and Zimmermann, K.: Earth System Model Evaluation Tool (ESMValTool) v2.0 – technical overview, Geosci. Model Dev., 13, 1179–1199, https://doi.org/10.5194/gmd-13-1179-2020, 2020.

Weigel, K., Eyring, V., Gier, B.K., Lauer, A., Righi, M., Schlund, M., Adeniyi, K., Andela, B., Arnone, E., Berg, P., Bock, L., Corti, S., Caron, L.-P., Cionni, I., Hunter, A., Lledó, L., Mohr, C.-M., Pérez-Zanón, N., Predoi, V., Sandstad, M., Sillmann, J., Vegas-Regidor, J. and von Hardenberg, J.: ESMValTool (v2.0) – Diagnostics for extreme events, regional model and impact evaluation and analysis of Earth system models in CMIP. Geosci. Model Dev., in prep., 2020.

---

## Author Comment (AC3) · 19 Mar 2020

**Reply to Martine Michou**

We thank Martine Michou for the helpful comments. We have now revised our manuscript in light of these and the other review comments we have received. A pointwise reply is given below.

**The text mentions that Figures 42 and 43 consist of an adaptation of di-**

**agnostics included in the CCMValDiag tool. It is very valuable to have recipe_eyring06jgr.yml in v2.0 of ESMValTool (it is not yet the case as of January 2020), as well as a recipe with all diagnostics of Eyring et al, JGR 2013, "Long-term ozone changes and associated climate impacts in CMIP5 simulations" that includes interesting diagnostics with regards to ozone evaluation.**

Thanks for your interest in the inclusion of the diagnostics from the Eyring et al., JGR, 2006 and 2013 papers on ozone evaluation. We agree that it would be good to have them available in ESMValTool v2. A subset of the diagnostics from Eyring et al., JGR, 2006 (e.g. Figures 42 and 43) are part of the official release of version 2 of the ESMValTool. These are described in the manuscript. Additional diagnostics from these two papers will be added as ressources allow.

**Second, as of January 2020, it is not easy at all to install ESMValTool v2.0, unlike v1.0. For users without conda, that is only recommended in the install web page, the installation with the "pip install ." command, as written in the install web page does not work. Documentation should be amended, in particular for the use of more "dummy" users, if ESMValTool v2.0 is to be used by a wide community. And I think it deserves it.**

Installing the ESMValTool without using conda is not possible at the moment. The installation procedure is clearly described in the ESMValTool documentation on ReadTheDocs (https://esmvaltool.readthedocs.io/en/latest/getting_started/install.html) and has been successfully applied by all users and developers so far. Specific installation issues (which might arise, for example, on some peculiar architectures) can be reported on the GitHub page of the ESMValTool and are readily addressed by the development team.

---

## Referee Report (RR1)

Re-Review of ESMValTool v2.0 - Extended set of large-scale diagnostics for quasi-operational and comprehensive evaluation of Earth system models in CMIP by Eyring et al.

General Comments

The paper is greatly improved. I appreciate the authors efforts to address the reviewer's comments. I feel they have done an excellent job. I recommend the paper be accepted. I have only two very minor comments below.

Specific Comments

1. Line 117-118 - including but not limited to those on global mean temperature or precipitation – This seems very wordy. How about "for example global mean temperature or precipitation".

2. Line 435 – Deser et al 2017 – Deser et al. investigates the issue from the obs. Wittenberg et al demonstrate the issue in a model simulation. Both seem relevant. Wittenberg et al seem like an important reference to include here.

Wittenberg, Andrew T., June 2009: **Are historical records sufficient to constrain ENSO simulations?** *Geophysical Research Letters*, **36**, L12702, DOI:10.1029/2009GL038710.